# Scanless two-photon voltage imaging

**Ruth R. Sims** [1,6], **Imane Bendifallah** [1,6], **Christiane Grimm** [1], **Aysha S. Mohamed Lafirdeen**[1], **Soledad Domínguez** [1], **Chung Yuen Chan**[1], **Xiaoyu Lu** [2], **Benoît C. Forget**[1], **François St-Pierre**[2,3,4,5], **Eirini Papagiakoumou** [1] ✉ & **Valentina Emiliani** [1] ✉

Two-photon voltage imaging has long been heralded as a transformative approach capable of answering many long-standing questions in modern neuroscience. However, exploiting its full potential requires the development of novel imaging approaches well suited to the photophysical properties of genetically encoded voltage indicators. We demonstrate that parallel excitation approaches developed for scanless two-photon photostimulation enable high-SNR two-photon voltage imaging. We use whole-cell patch-clamp electrophysiology to perform a thorough characterization of scanless two-photon voltage imaging using three parallel illumination approaches and lasers with different repetition rates and wavelengths. We demonstrate voltage recordings of high-frequency spike trains and sub-threshold depolarizations from neurons expressing the soma-targeted genetically encoded voltage indicator JEDI-2P-Kv. Using a low repetition-rate laser, we perform multi-cell recordings from up to fifteen targets simultaneously. We co-express JEDI-2P-Kv and the channelrhodopsin ChroME-ST and capitalize on their overlapping two-photon absorption spectra to simultaneously evoke and image action potentials using a single laser source. We also demonstrate in vivo scanless two-photon imaging of multiple cells simultaneously up to 250 μm deep in the barrel cortex of head-fixed, anaesthetised mice.

Deciphering the logic and syntax of neural computation is a central goal in neuroscience and requires methods capable of recording (reading-out) and manipulating (writing-in) the activity of individual neurons. Electrophysiological approaches can read and write neural activity with high fidelity. However, while extracellular probes can record activity from large populations, they have limited spatial resolution and currently lack the ability to manipulate a specific genetic class of cells. In contrast, whole-cell patch-clamp methods can manipulate and record the electrical activity of specific neurons but are invasive, hard to achieve in vivo (even for a handful of neurons simultaneously) and unsuitable for longitudinal (chronic) studies. Furthermore, electrophysiological methods currently have limited access to subcellular compartments of neurons (such as axons, distal

dendrites, spines and boutons). These limitations have stimulated the development of a plethora of photonic approaches, which combine advanced optical instrumentation with fluorescent indicators and optogenetic actuators, for recording and manipulating neural activity, respectively[1-3].

In the nervous system, calcium ions regulate a broad range of processes and generate versatile intracellular signals[4]. Since action potentials (APs) lead to large elevations of intracellular calcium, which can last an order of magnitude longer than the APs themselves[5], the developments of synthetic[6] and genetically encoded[7] fluorescent calcium indicators (GECIs) capable of reporting changes in intracellular calcium were extremely important scientific breakthroughs. GECIs can be targeted to sub-cellular compartments and specific cell types[8,9].

[1]Institut de la Vision, Sorbonne Université, INSERM, CNRS, Paris, France. [2]Systems, Synthetic, and Physical Biology Program, Rice University, Houston, TX, USA. [3]Department of Neuroscience, Baylor College of Medicine, Houston, TX, USA. [4]Department of Biochemistry and Molecular Pharmacology, Baylor College of Medicine, Houston, TX, USA. [5]Department of Electrical and Computer Engineering, Rice University, Houston, TX, USA. [6]These authors contributed equally: Ruth R. Sims, Imane Bendifallah. ✉e-mail: eirini.papagiakoumou@inserm.fr; valentina.emiliani@inserm.fr

Their long-term expression in intact tissues and organisms[2] enables the repeated observation of individual neurons. Calcium transients generally last significantly longer than the underlying voltage fluctuations, facilitating the detection of neural activity, but also limiting the quantification of spike firing rates and timing, particularly in the case of high-frequency trains of APs. Furthermore, GECIs are not well suited for detecting sub-threshold voltage changes or hyperpolarisations resulting from synaptic and neuromodulatory inputs[10].

Voltage indicators, which transduce changes in membrane potential into optical signals, promise to address many of the afore-mentioned limitations of GECIs[11]. Since the first revolutionary optical recordings of membrane potential with a synthetic dye[12], voltage-sensitive indicators have undergone continuous development, leading to the current toolbox, which includes improved synthetic dyes[13], genetically encoded voltage indicators (GEVIs)[14] and hybrid voltage indicators[15,16]. However, detecting single APs with voltage indicators typically necessitates orders of magnitude faster imaging rates than with GECIs. Similarly, the small signals associated with sub-threshold post-synaptic changes in membrane potential can only be detected using highly sensitive imaging approaches. These technical challenges have thus far limited the broad adoption of voltage indicators in optical neurophysiology experiments.

The majority of optical voltage imaging experiments so far have relied on widefield, one-photon (1P) illumination and camera detection. The major advantage of 1P approaches for voltage imaging is their relative light efficiency; 1P excitation typically requires 3–4 orders of magnitude lower average powers than two-photon (2P) excitation[17]. Additionally, 1P widefield microscopy approaches maximise the number of fluorescence photons generated and collected[18]. The resulting mesoscopic observations of population activity have enabled the investigation of the functional organisation and dynamics of cell-type specific excitatory and inhibitory cortical circuits[19,20]. To approach single-cell resolution, the lack of optical sectioning of 1P widefield microscopy has been overcome using sparse labelling strategies[21,22] and/or targeted illumination[23–26], for instance, by amplitude modulation using digital micromirror devices (DMDs)[27]. As this approach does not confer any axial sectioning, it is best suited to sparsely labelled samples, since light targeting different regions overlaps and excites fluorescence above and below the focal plane. This generates background and "crosstalk" from out-of-focus cells, neuropil and blood flow[17,28], unless differential modulation (structured illumination)[24,29] or confocal detection[30,31] strategies are employed. Targeted illumination based on 1P holography using an SLM has also been utilised for 1P voltage imaging[32], which is advantageous compared to DMD approaches, since it is possible to exploit the full numerical aperture (NA) of the microscope objective to obtain moderate axial resolution[33], sufficient for targeting single cells in sparsely labelled samples, even for laterally extended patterns. These limitations have so far limited the use of 1P voltage imaging approaches to recordings of sparsely labelled populations in superficial cortical layers (<200 μm below the dura for JEDI-2P-Kv)[17].

In principle, the optical sectioning and longer excitation wavelengths inherent to multi-photon excitation ought to be capable of recording neural activity with cellular resolution from deeper cortical layers[34]; indeed, 2P laser scanning microscopy (2P-LSM) is by far the most prevalent approach for calcium imaging of neural activity with single cell resolution in scattering tissue[35]. However, the acquisition rate of conventional 2P-LSM is limited and millisecond transients such as APs can only be detected by drastically reducing the field of view (FOV)[36–39]. Although such approaches are valuable for imaging densely packed brain regions such as the hippocampus[40], neurons in other brain regions are typically much sparser and hence require more specialised imaging approaches, based on spatial and/or temporal multiplexing to record neural activity across larger FOVs at kilohertz rates[41–47]. Since the neurons, and in particular, cell-body (somal)

membranes generally occupy a small fraction of the total imaging volume, classical raster scan trajectories use the finite photon budget inefficiently[48]. Random access microscopy based on acousto-optic holography (AO-holography)[49–52], overcomes this limitation by recording from pre-defined regions of interest (ROIs). AO-holography has demonstrated the benefits of 2P imaging with extended excitation volumes[53], achieving spectacular results such as recording the voltage dynamics of cortical neurons in layer 5 in awake behaving mice[54]. However, AO-holography imposes strict symmetry constraints on the output patterns[55] and, thus far, has only been used to perform voltage imaging of a limited number of neurons simultaneously.

Here, we propose an alternative approach for high-contrast, high-resolution voltage imaging. Our method leverages three existing scanless 2P excitation approaches: Generalised Phase Contrast (GPC)[56,57], low numerical aperture (NA) Gaussian beams and Computer-Generated Holography (CGH)[58,59], in combination with temporal focusing (TF)[60–62]. Whilst 2P-TF-Gaussian and 2P-CGH excitation have both previously been exploited for performing fast[63], volumetric Calcium imaging[64,65], here we demonstrate that scanless illumination can also be used to perform scanless 2P voltage imaging at kilohertz acquisition rates with camera detection and the recently developed soma-targeted GEVI: JEDI-2P-Kv[54]. Using simultaneous imaging and electrophysiology, we provide a thorough, quantitative, comparison of these illumination modalities using an 80 MHz laser source tuned to 940 nm (as used for conventional 2P-LSM). Next, by viral expression of JEDI-2P-Kv in mouse hippocampal organotypic slices, we show that 2P-TF-GPC enables high spatiotemporal resolution voltage imaging of neural activity in a challenging, densely labelled preparation. We demonstrate the detection of high-frequency spike trains and sub-threshold membrane depolarisations with amplitudes on the order of excitatory post-synaptic potentials (PSPs). We further demonstrate that scanless 2P voltage imaging of JEDI-2P-Kv expressing neurons can be performed at 1030 nm, using conventional 80 MHz tuneable lasers or low repetition rate, amplified, Ytterbium-doped fibre lasers which provide much higher pulse energies. We find that high SNR scanless 2P voltage imaging can be performed with amplified lasers at 1030 nm using relatively low average powers. We exploit this finding to image multiple targets simultaneously whilst mitigating light-induced temperature rises, and further, to perform scanless 2P voltage imaging in vivo at depths up to 250 μm with camera detection. Finally, we demonstrate the feasibility of performing single-beam voltage imaging and optogenetic stimulation, which ought to permit in situ characterisations of the number and timing of light-evoked APs. Collectively, these results demonstrate the feasibility of scanless 2P voltage imaging experiments, establish a set of suitable recording conditions capable of acquiring high SNR data whilst minimising physiological perturbations and hence pave the way for studying neural function with scanless 2P all-optical neurophysiology.

## Results
### Scanless two-photon voltage imaging with sculpted, temporally focused excitation
The principal optical setup built for this work (Fig. 1a, Supplementary Fig. 1 and Supplementary Table 1) was comprised of two independent excitation paths, one designed to generate temporally focused (TF) Generalised Phase Contrast (GPC) patterns[56,57] or low-NA Gaussian spots, and the second for Computer Generated Holography (CGH)[66] (Supplementary Fig. 1 and Supplementary Table 1). Temporal focusing was used to elongate the pulse duration outside of the focal plane (Fig. 1a: temporal domain inset) to eliminate spurious off-target excitation. The two excitation paths were combined prior to the microscope objective with a polarising beam splitter. Each excitation path was designed to generate circular, temporally focused, linearly polarised spots with dimensions approximating the size of a neuronal soma (12 μm lateral full width at half maximum (FWHM) and ~9 μm axial

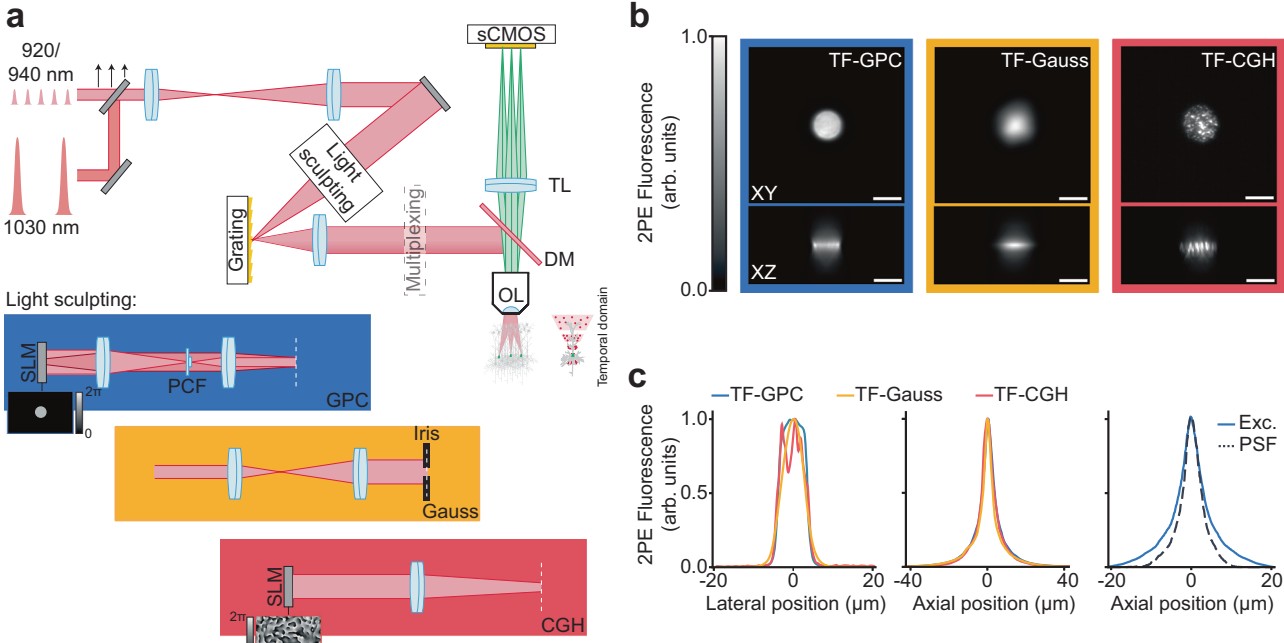

**Fig. 1 | Characterisation of the optical setup developed for scanless two-photon voltage imaging. a** Schematic diagram of a typical optical setup used to perform scanless 2P voltage imaging (refer also to Supplementary Fig. 1). High repetition rate (920 or 940 nm) sources delivering nJ pulses at 80 MHz and fixed wavelength (1030 nm) low repetition rate sources delivering μJ pulses (at variable repetition rates) were used for scanless 2P voltage imaging. The black arrows indicate an adjustable mirror used to direct light from either ultrafast source into the microscope. The light sculpting paths were designed to generate Temporally Focused (TF), Generalised Phase Contrast (GPC), Gaussian and holographic (CGH) spots with lateral FWHM (Full Width Half Maximum) diameters between 12 and 17 μm. The essential concepts of each light sculpting approach are depicted in the inset below the schematic. Diffraction gratings were positioned in conjugate image planes for temporal focusing, as indicated by the white dashed lines. Temporal focusing was used to maintain axial resolution in spite of the extended lateral spot size. 2P excited fluorescence was collected using a widefield detection axis equipped with an sCMOS camera capable of kilohertz acquisition rates. Patch-clamp electrophysiology apparatus was installed on each microscope ("Methods" section). OL objective lens, DM dichroic mirror, TL tube lens, sCMOS scientific complementary metal–oxide–semiconductor, SLM spatial light modulator, PCF phase contrast filter. **b** Lateral (XY) and axial (Z) cross-sections of representative 2P excited fluorescence generated in a thin spin-coated rhodamine layer with 12 μm TF-GPC, TF-Gaussian and TF-CGH spots, as indicated. Scale bars represent 10 μm. **c** Lateral and axial profiles of 2P excited fluorescence generated with each excitation modality, and the corresponding system response. Exc. refers to the average excitation axial profile for all modalities and PSF to the effective Point Spread Function of scanless 2P microscopy, as measured using 1 μm fluorescence microspheres excited at 940 nm with TF-GPC. Source data are provided as a Source Data file.

FWHM for all modalities (Fig. 1b, c)). The excitation volume was tailored to match the typical dimensions of cell bodies to maximise the captured neuronal signal while avoiding excitation of neighbouring neurons and the neuropil. Fluorescence was detected by a scientific Complementary Metal–Oxide–Semiconductor (sCMOS) camera incorporated into a widefield detection axis. The system resolution was estimated by recording the fluorescence from 1 μm microspheres excited with a 12 μm TF-GPC spot and was found to have an axial FWHM of 3.7 μm (Fig. 1c, right panel). We could target multiple neurons simultaneously either by generating several spots in 2D using a single SLM or by multiplexing a single spot using a second SLM positioned in a conjugate Fourier (pupil) plane[67] (Supplementary Fig. 1). As previously demonstrated, the axial resolution of temporally focused excitation is independent of both the number of targets and the lateral dimensions of the spot[66,68]. The nominal field of excitation of each of the light sculpting approaches was 300 × 300 μm². The effective imaging FOV was limited in one dimension by the number of sCMOS rows that could be readout simultaneously at a given acquisition rate ("Methods"). Our systems were equipped with laser sources with different repetition rates and wavelengths in order to characterise scanless voltage imaging using different optical configurations (see Supplementary Tables 1–3 and the "Methods" section for precise details). The experimental conditions used to acquire each dataset are summarised in Supplementary Table 4.

To compare the performance of the three excitation modalities (2P-TF-GPC, 2P-TF-Gaussian and 2P-TF-CGH) for scanless 2P voltage imaging, we used a high repetition rate (80 MHz) laser source, as typically used for conventional 2P-LSM. We transiently expressed a recently developed, negative-going, voltage indicator optimised for 2P excitation (JEDI-2P-Kv)[54], in mammalian (CHO) cells (Fig. 2a). We controlled the membrane potential of individual cells with whole-cell patch-clamp electrophysiology and simultaneously performed scanless 2P voltage imaging. We used three protocols, hereafter named 1, 2 and 3 ("Methods" section, Supplementary Table 5), to evaluate and compare the scanless approaches.

Protocol 1 (Supplementary Table 5) was used to quantify the amplitude and SNR of voltage-dependent fluorescence changes using cells expressing JEDI-2P-Kv. The responses of patched cells to three 100 ms, 100 mV voltage steps were recorded at 100 Hz under continuous illumination for 3 s. Changes in fluorescence associated with membrane potential variations were clearly observed with all scanless modalities (Fig. 2b).

All data acquired using protocol 1 (Fig. 2b) were pooled and used to establish and validate an analysis pipeline capable of automatically identifying and segmenting neurons and of detrending the optical traces (Supplementary Fig. 2a). Due to the similarities between the data obtained with scanless 2P voltage imaging and 1P voltage imaging with widefield detection, we developed an analysis pipeline based on existing open-source packages[15,28,69,70]. Compared with results obtained by calculating the unweighted mean of all pixels within segmented cells, a regression-based pixel weighting algorithm ("Methods"), which also improved segmentation, was found to

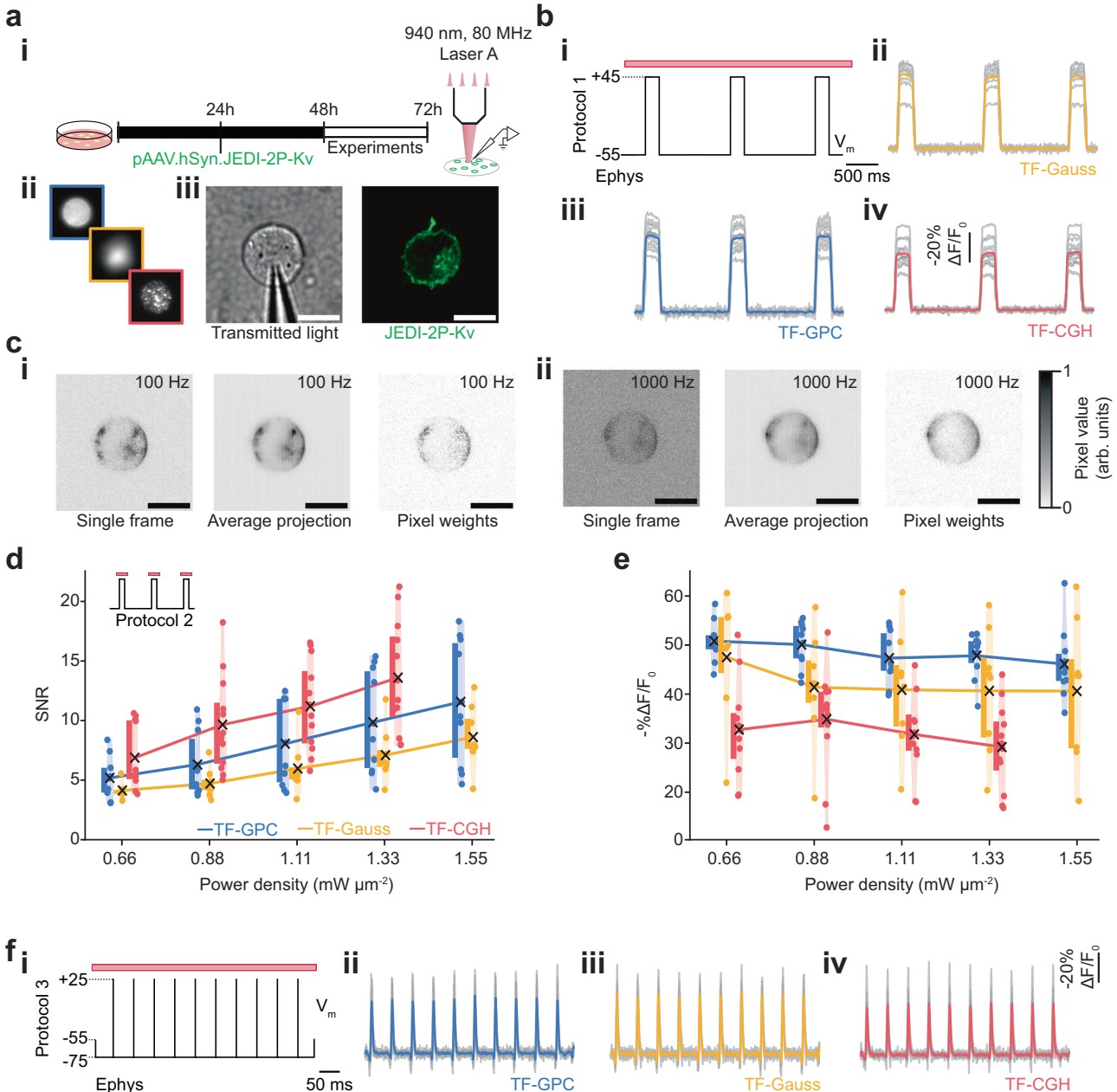

**Fig. 2 | In vitro electrophysiological characterisation of scanless two-photon voltage imaging in cultured CHO cells. a** (i) Schematic representation of the protocol used to prepare JEDI-2P-Kv expressing CHO cells ("Methods" section). (ii) Scanless 2P voltage imaging was performed using temporally focused (TF) GPC (Generalized Phase Contrast), Gaussian or holographic (CGH) spots. (iii) Transmitted light image of a patched CHO cell (left) and representative confocal image of a different JEDI-2P-Kv expressing CHO cell (right) (*n* = 41 cells, 19 independent transfections). Scale bars represent 10 μm. **b** Data from protocol 1 ("Methods" section, Supplementary Table 5) used to assess the performance of each illumination modality (100 Hz acquisition rate, power density 0.88 mW μm$^{-2}$). The electrophysiological (ephys) command voltage is plotted in black with relevant holding potentials indicated. The red bar represents the illumination epoch. (ii–iv) Average fluorescence responses acquired using each modality. Results from single trials from independent cells are also plotted in grey. Responses are reported as (-%ΔF/F$_0$ (*n* = 9 (Gauss), 15 (CGH), 17 (GPC) cells; from 2 (Gauss, CGH) or 3 (GPC) independent transfections). **c** Representative imaging data acquired during scanless 2P voltage imaging experiments on JEDI-2P-Kv expressing CHO cells at 100 Hz, (i) and

1 kHz, (ii) (power densities 0.88 and 1.11 mW μm$^{-2}$ respectively). Left: single frames, middle: temporal average of all frames, right: corresponding pixel weights (all normalised) used to generate the final fluorescence traces ("Methods"). (d-e) Violin plots (shaded) summarising the performance: -%ΔF/F$_0$ (**d**) and SNR (**e**), of each scanless 2P voltage imaging modality at different illumination power densities (0.66–1.55 mW μm$^{-2}$, *n* = 8 (Gauss), 13 (CGH), 12 (GPC) cells, 2 independent transfections per modality). Results presented in **d** and **e** were acquired using protocol 2 ("Methods" section, Supplementary Table 5). In every case, a coloured point represents a single measurement from an individual cell, a black cross is located at the population mean and the coloured bars (adjacent) depict the interquartile range. **f** (i) Electrophysiological command voltage for protocol 3 ("Methods" section and Supplementary Table 5) plotted in black with relevant holding potentials indicated. The red bar represents the illumination epoch. (ii–iv) Corresponding scanless 2P imaging data acquired using each parallel illumination modality. Results from single trials from independent cells are also plotted in grey (power density: 1.33 mW μm$^{-2}$, *n* = 8 (Gauss), 11 (CGH, GPC) cells, 2 independent transfections per modality). Source data are provided as a Source Data file.

increase -%$\Delta F/F_0$ per 100 mV for all modalities ($43 \pm 2$ vs $34 \pm 2$, mean $\pm$ S.E.M., $p < 0.00001$, Supplementary Fig. 2c), resulting in values in accordance with those previously reported for JEDI-2P-Kv[54]. Unless otherwise specified, all results are reported as mean $\pm$ S.E.M. hereafter. No significant difference in SNR (signal-to-noise ratio: signal amplitude divided by the standard deviation of the baseline signal) was found between the two analysis approaches ($59 \pm 5$ vs. $60 \pm 4$, $p = 0.9158$, Supplementary Fig. 2d), because the pixel-weighting approach approximately halved the number of analysed pixels ($8962 \pm 418$ vs $16263 \pm 340$ pixels, $p < 0.00001$, Supplementary Fig. 2e). However, the location of these remaining pixels tended to overlap with the plasma membrane (Fig. 2c and Supplementary Fig. 2e, inset). A slight reduction in photostability was observed for the fluorescence traces calculated using a weighted mean as compared with the unweighted mean ($0.80 \pm 0.005$ vs. $0.82 \pm 0.003$, $p < 0.00001$, Supplementary Fig. 2f). These results were found to be consistent for data acquired with each of the different excitation modalities when analysed separately, indicating that the analysis pipeline was not biased towards a particular modality (Supplementary Fig. 2g–o).

Having established the analysis pipeline, we then compared the data acquired using the three different excitation modalities using protocol 1 (Supplementary Table 5 and Supplementary Fig. 3a, b). Data obtained with 2P-TF-CGH consistently exhibited the highest SNR (Supplementary Fig. 3b, $82 \pm 9$), almost double that of 2P-TF-GPC ($49 \pm 5$, $p = 0.0022$) and 2P-TF-Gaussian ($44 \pm 7$, $p = 0.0061$). We attribute these differences to the speckle patterns associated with holographic spots when using CGH. Precisely, when holograms are generated using certain variants of the Gerchberg-Saxton algorithm, a random initial phase is utilised to distribute light throughout the image plane, and no subsequent constraints are imposed on the phase. Consequently, different components of the modulated beam interfere constructively and destructively in the image plane, resulting in an inhomogeneous distribution of intensity within the target pattern[71]. The resulting high spatial confinement of photons in speckle grains generates 2P excitation more efficiently, which results in an increased SNR for 2P-TF-CGH.

Since the high density of photons in speckles increases the likelihood of non-linear photophysics (for instance, photobleaching[72]), we designed and utilised a different protocol (Protocol 2, Supplementary Table 5) to investigate the extent of these non-linear effects as a function of the excitation power density. Protocol 2 consisted of three 100 ms, 100 mV voltage steps, 200 ms illumination pulses centred on each voltage step, and 2.5 s inter-pulse intervals (Supplementary Fig. 4a and Supplementary Table 5). For all modalities, the baseline fluorescence ($F_0$) increased quadratically as a function of power density as expected with 2P illumination, indicating that fluorescence excitation was not saturated at any of the powers used. Consequently, the SNR increased linearly as a function of power density (Fig. 2d, $R^2 = 0.999$ (2P-TF-GPC), $R^2 = 0.995$ (2P-TF-Gaussian), $R^2 = 0.998$ (2P-TF-CGH)), confirming that experiments were performed in a shot-noise limited regime.

The SNR of the fluorescence transients due to 100 mV changes in membrane potential (Protocol 2, Supplementary Table 5) was higher at all excitation power densities with 2P-TF-CGH than with 2P-TF-GPC or 2P-TF-Gaussian, (Fig. 2d and Supplementary Fig. 4b), though the 2P-TF-CGH fluorescent transients exhibited a systematically lower average -%$\Delta F/F_0$ than for 2P-TF-GPC (Fig. 2e and Supplementary Fig. 4c). The magnitude of this difference increased as a function of power density (Supplementary Fig. 4c). We found that JEDI-2P-Kv retained over 80% of its original fluorescence at even the highest power densities used for all scanless excitation modalities (Supplementary Fig. 4d). No significant difference in photostability between different modalities was observed. On the other hand, for 2P-TF-Gaussian and 2P-TF-GPC, at all powers, fluorescence was observed to recover to over 97% of its initial value following 2.5 s without

illumination, whilst for 2P-TF-CGH, photorecovery decreased as a function of excitation power density and was lower than the other modalities for power densities >0.88 mW $\mu m^{-2}$ (100 mW per cell, $p = 0.01$, Supplementary Fig. 4e). However, even in the case of 2P-TF-CGH, photorecovery remained above 87.5% at the highest power density used.

Finally, we used protocol 3 (Fig. 2f and Supplementary Table 5) to assess the detection of short, AP-like transients with scanless 2P voltage imaging. Cells were illuminated continuously for 500 ms and the fluorescence response to a 20 Hz train of 10 rectangular pulses (100 mV amplitude, 3 ms duration) was recorded at 1 kHz (Supplementary Table 5, "Methods" section). The transients recorded with 2P-TF-GPC, 2P-TF-Gaussian and 2P-TF-CGH, had -%$\Delta F/F_0$ values of $45 \pm 5\%$, $42 \pm 6\%$ and $26 \pm 2\%$ respectively (Supplementary Fig. 5a, b). For all modalities, the average SNR was greater than 11, indicating that action-potential-like signals can be reliably detected in single trials with scanless 2P voltage imaging. As per data presented in Supplementary Fig. 5c, the highest SNR data was acquired using 2P-TF-CGH ($21 \pm 2$ compared with $14 \pm 1$ ($p = 0.01512$, 2P-TF-GPC) and $12 \pm 1$ ($p = 0.00254$, 2P-TF-Gaussian)), in-spite of the reduced -%$\Delta F/F_0$ observed.

Overall, these results confirm that all scanless approaches can report voltage signals in vitro, albeit each with subtle advantages and limitations. Since the SNR of data acquired using 2P-TF-CGH was significantly higher than for 2P-TF-Gaussian or 2P-TF-GPC, we consider it the optimal modality for minimising the incident average power. Although no significant performance differences were found between 2P-TF-Gaussian and 2P-TF-GPC, 2P-TF-Gaussian requires higher powers at the laser output to achieve a given SNR. Uniform illumination of the plasma membrane (2P excitation profile with 12 $\mu m$ FWHM) was achieved with 2P-TF-Gaussian by expanding and subsequently cropping the beam with an iris in a conjugate image plane prior to the objective, to remove extraneous light which would contribute to background fluorescence and sample heating but not useful signal. Thus 2P-TF-Gaussian was ~3 times less power-efficient than 2P-TF-GPC[73]. However, it is perhaps the simplest approach to implement, and hence a suitable solution given a sufficiently powerful laser source. In the following sections, we demonstrate that each modality can be applied to record scanless two-photon voltage imaging of neural activity.

## Scanless two-photon voltage imaging of neural activity in hippocampal organotypic slices

The next step was to define the power densities and acquisition rates required for recording neural activity with scanless 2P voltage imaging, and to determine whether these imaging conditions perturb neural activity or otherwise impact cellular physiology. We performed simultaneous 2P-TF-GPC imaging and whole-cell patch-clamp recordings of granule cells located in the dentate gyrus (DG) of organotypic slices transduced with JEDI-2P-Kv ("Methods", Fig. 3a). Expression of JEDI-2P-Kv in the granule cells of the DG was well localised to the plasma membrane, with no evidence of intracellular aggregation (Fig. 3a, inset). Even though granule cells are extremely densely packed in DG, due to the top-hat lateral profile of the illumination spot and the optical sectioning conferred by temporally focused, targeted 2P illumination, we were able to image individual neurons with high-contrast using scanless 2P voltage imaging (Fig. 3a).

Using protocol 1 (Supplementary Table 5), we confirmed we could detect 100 mV depolarisations in densely labelled, scattering organotypic slices, and that no change to the analysis pipeline outlined previously was required (Supplementary Figs. 6a–c). Similar -%$\Delta F/F_0$ to that obtained in CHO cells was obtained ($51 \pm 3$ vs $43 \pm 2$, $p = 0.0552$, Supplementary Fig. 6d). No significant differences were observed in SNR ($69 \pm 6$ (CHO), $50 \pm 8$ (organotypic slices), $p = 0.0552$, Supplementary Fig. 6e) or photostability between hippocampal

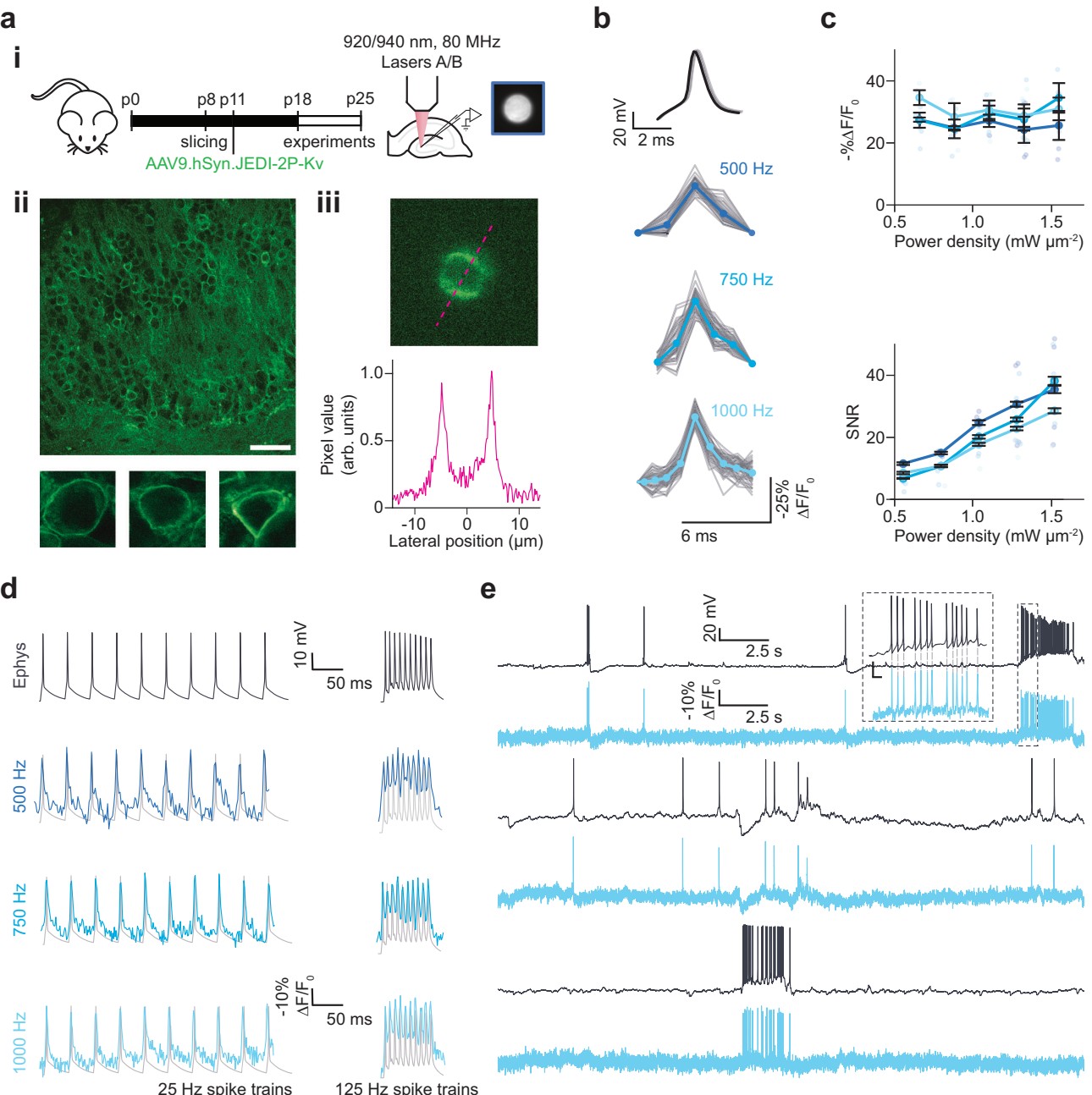

**Fig. 3 | Scanless two-photon voltage imaging of neural activity in hippocampal organotypic slices. a** (i) Schematic representation of the protocol used to prepare JEDI-2P-Kv expressing hippocampal organotypic slices (see "Methods" section and Supplementary Table 1). (ii) Upper: confocal image of a representative organotypic slice bulk-infected with JEDI-2P-Kv. Scale bar represents 50 μm. Lower: confocal images of single representative JEDI-2P-Kv expressing neurons in the dentate gyrus ($n = 29$ slices from 16 independent slice cultures). (iii) Upper: representative image acquired with 2P-TF-GPC (1 kHz acquisition rate, average temporal projection), Lower: line-profile through the image (indicated by the magenta dashed line). **b** Electrically induced and recorded APs (upper, black) and optically recorded (lower) were resolved in single trials (plotted in grey) using 2P voltage imaging at different acquisition rates. The average traces from different acquisition rates (500 Hz, 750 Hz and 1 kHz, power density: 1.1 mW μm$^{-2}$) are plotted in different shades of blue. **c** −%ΔF/$F_0$ and SNR plotted as a function of power density in different shades of blue for different acquisition rates. Error bars represent the standard error of measurements across all cells ($n = 4$−6 neurons, refer to Table 1 in

"Methods" section for precise values of n). Individual points represent the average value over 50 evoked APs for each individual cell. **d** Representative fluorescence responses recorded from an individual cell whilst 25 and 125 Hz spike trains were evoked and recorded electrically (labelled). Imaging data was acquired at three different rates (500 Hz, 750 Hz and 1 kHz, power density: 1.11 mW μm$^{-2}$). Data from different acquisition rates is plotted in different shades of blue, and corresponding electrophysiological whole-cell patch-clamp recordings are plotted in grey ($n = 2$−5 neurons from 3 different slices, from 1 slice culture). **e** Simultaneous current-clamp (upper, black) and fluorescence recordings (lower, blue) of spontaneous activity in neurons in the dentate gyrus of hippocampal organotypic slices over a continuous 30 s recording period. Inset: zoomed in portion of the electrophysiological and fluorescence traces. The dashed light grey lines indicate the correspondence between APs in the electrophysiological and fluorescence traces (average spike train rate: 17 Hz, power density: 1.33 mW μm$^{-2}$, 1 kHz acquisition rate). Source data are provided as a Source Data file.

**Table 1 | Number of cells used to generate data points at each power in Fig. 3c**

|        | 1 kHz | 750 Hz | 500 Hz |
|--------|-------|--------|--------|
| 75 mW  | 6     | 6      | 5      |
| 100 mW | 4     | 4      | 6      |
| 125 mW | 4     | 5      | 5      |
| 150 mW | 4     | 4      | 4      |
| 175 mW | 4     | 4      | 4      |

organotypic slices and CHO cells ($0.80 \pm 0.005$ (CHO), $0.81 \pm 0.005$ (organotypic slices), $p = 0.1863$, Supplementary Fig. 6f), even though the segmentation resulted in fewer voltage-responsive pixels than found for the CHO cells (Fig. 6g). The physiological lateral and axial resolution of the scanless 2P imaging system, quantified as the relative -%$\Delta F/F_0$ of an electrically evoked AP as a function of the distance between the excitation spot and the soma, was found to be approximately isotropic and of similar dimensions to the neuronal soma (14 μm lateral and 13 μm axial FWHM, Supplementary Fig. 6h), confirming that scanless 2P voltage imaging maintains isotropic cellular resolution in densely labelled preparations.

We recorded the fluorescence from patched cells while 50 APs were evoked electrically by 2 ms current injections at a rate of 1 Hz (Fig. 3b). Electrically evoked APs were imaged with 3 different acquisition rates: 500 Hz, 750 Hz, and 1 kHz (corresponding to per-frame exposure times of 2, 1.33 and 1 ms respectively) as previously used for 1P voltage imaging with camera detection[26]. In all conditions, individual APs could be clearly identified from single trials in the raw fluorescence traces (representative traces for single cells plotted in Fig. 3b). Putative APs were identified by template matching, using the most prominent peaks originally identified in each fluorescence trace[69]. The 1 kHz recordings exhibited a slightly higher -%$\Delta F/F_0$ than 500 Hz recordings across all powers ($31 \pm 2$ vs $26 \pm 2$, $p = 0.05$, Fig. 3c, upper, Table 1). However, a higher SNR was achieved for 500 Hz recordings than for 1 kHz recordings ($19 \pm 2$ vs $29 \pm 4$, $p = 0.03$, Fig. 3c, lower, Table 1), because the increase in the number of photons collected per frame more than compensated for the reduced -%$\Delta F/F_0$, consistent with previous reports[22].

Having established that it was possible to record APs with high SNR in single trials at different acquisition rates, we next tested whether we could also monitor individual spikes within high-frequency trains of APs (such as bursts) under these conditions (Fig. 3d). We observed that an acquisition rate of 500 Hz was sufficient to track individual APs in trains with frequencies up to 100 Hz (Supplementary Fig. 7a), using power densities as low as 0.66 mW μm$^{-2}$ (75 mW per cell). As a result of increased SNR recordings with lower acquisition rates, the detection probability (fraction of correctly identified APs) was higher using lower power densities (Supplementary Fig. 7b). However, an acquisition rate of 500 Hz was insufficient for robustly tracking spikes in 125 Hz trains due to a reduction in -%$\Delta F/F_0$ (Supplementary Fig. 7b, c), which led to a deterioration in detection probability and fluorescence response compared with the data acquired at 1 kHz at power densities $\geq 0.66$ mW μm$^{-2}$ (corresponding to 75 mW per cell, Supplementary Fig. 7c).

Next, we tested whether scanless 2P voltage imaging could record spontaneous neuronal activity. We performed simultaneous electrophysiological and fluorescence recordings (2P-TF-GPC, 30 s continuous illumination) of spontaneous activity from neurons in hippocampal organotypic slices which exhibited a range of different resting potentials. We were able to observe several hallmarks of spontaneously generated activity, such as large slow depolarisations, bursts of APs, rhythmic sub-threshold depolarisations and hyperpolarisations (Fig. 3e). These results confirmed the capability of scanless 2P voltage imaging for high temporal precision, single trial, recordings of APs and sub-threshold events.

We next examined whether these conditions were also suitable for imaging sub-threshold changes in membrane potential. To emulate excitatory PSPs, patched cells were clamped to −75 mV, while the membrane potential was varied in 0.5 mV steps from 0 to 2.5 mV for 20 ms (Fig. 4a). This recording was repeated 50 times (Fig. 4b). Since it was not possible to detect these transients from individual recordings, we averaged data from different trials to improve the SNR (Supplementary Fig. 7d). Averaging data from 25 repeats was sufficient to stabilise the magnitude of the fluorescence transient (-%$\Delta F/F_0$) for a given depolarisation and to increase the SNR above 1 for all depolarisations larger than 0.5 mV (Fig. 4c, d and Supplementary Fig. 7d). Collectively, these results demonstrate the potential of scanless 2P voltage imaging for recording sub-threshold events, even though the F-V curve (-%$\Delta F/F_0$ as a function of control voltage) of JEDI-2P is not optimised for such an application. Collectively, the results presented in Figs. 3 and 4 and Supplementary Fig. 7 parameterise the imaging conditions required to achieve a sufficient SNR to robustly detect neural activity using scanless 2P voltage imaging.

We next investigated whether these imaging conditions induced any observable physiological perturbations. We performed simultaneous patch-clamp recordings and scanless 2P voltage imaging using 2P-TF-CGH. We performed these experiments using 2P-TF-CGH, since we anticipated that non-linear effects would be most severe with this modality due to the high peak photon densities in the speckle grains, and hence, any conclusions would represent upper bounds for the other two modalities. Neuron properties (capacitance, rheobase, firing rate and resting potential) were measured prior to illumination and after each illumination epoch (see "Methods" section, protocol 4, Supplementary Table 5 and Supplementary Fig. 8a). Patched cells illuminated continuously for 30 s at each power density tested. At the start of each illumination epoch, 5 APs were evoked by current injection at a rate of 5 Hz. The half-width and amplitude of these evoked APs were monitored as a function of illumination power density. Control experiments were also performed using an identical protocol, without illumination. Results were pooled into two groups corresponding to neurons illuminated with power densities between 0.66–1.11 mW μm$^{-2}$ (75–125 mW, group 1), and those illuminated with power densities between 1.33–1.77 mW μm$^{-2}$ (150–200 mW, group 2). No statistically significant differences were found for any of the monitored parameters between the control group and group 1 (Supplementary Fig. 8b–g). However, we found that using higher power densities (group 2) resulted in a decrease in capacitance ($\Delta$Capacitance = $-33.8 \pm 12.4$ (group 2) vs $-5.4 \pm 2.5$ pF (control), $p = 0.0089$, Supplementary Fig. 8b) and rheobase ($\Delta$Rheobase = $-14.4 \pm 2.0$ (group 2) vs $-7.0 \pm 2.5$ pA (control), $p = 0.0235$, Supplementary Fig. 8c), in addition to an increase of the AP half-width ($\Delta$half-width= $2.2 \pm 0.6$ (group 2) vs $0.2 \pm 0.1$ ms (control), $p = 0.0002$, Supplementary Fig. 8f), resulting from a distortion of the AP waveform. To determine whether these changes were exacerbated by repeated light exposure, we repeated protocol 4 (1.11 mW μm$^{-2}$) ten times, (corresponding to five-minutes total illumination) and monitored the membrane parameters as a function of total illumination time. On average, we observed a larger and faster depolarisation of illuminated neurons compared with the control cells ($\Delta$Resting membrane potential= $10.7 \pm 1.8$ vs $5.7 \pm 3.2$ mV, Supplementary Fig. 8h). However, no statistically significant differences were observed between the control and illuminated cells.

Given that the peak intensities required for scanless 2P imaging (<2 nJ/ cell per pulse) were well below previously reported non-linear photodamage thresholds (-300 nJ/ cell per pulse)[74,75], we hypothesised that these perturbations were the result of heating due to light absorption. To verify this hypothesis and estimate temperature rises as a function of average power, recording time and number of targets, we modelled heat diffusion and temperature rise using the Green's function formalism for a homogeneous and isotropic (infinite) medium[76,77] ("Methods"). According to the model, continuously illuminating the

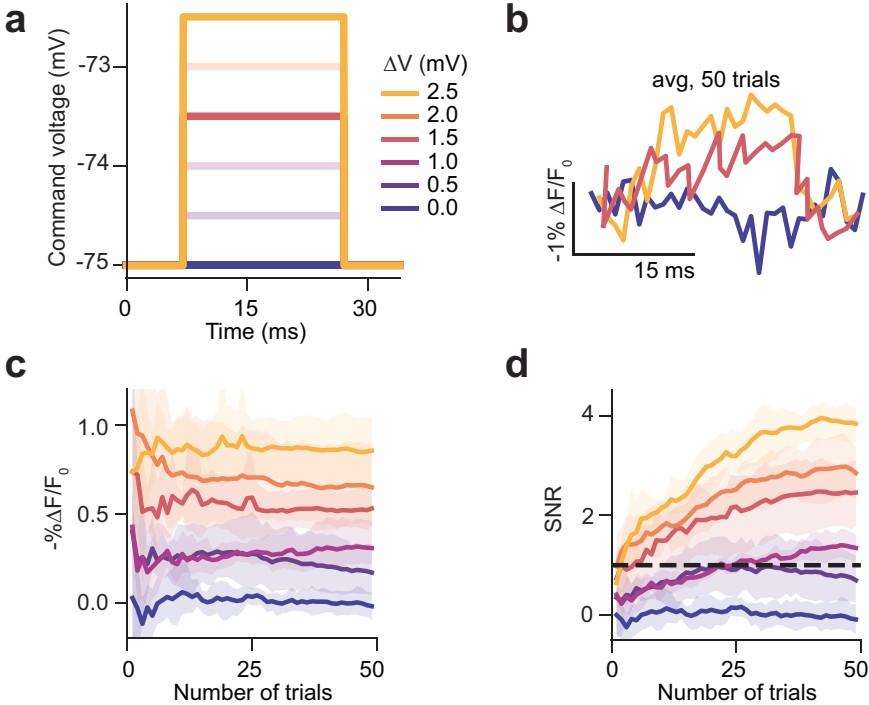

**Fig. 4 | Scanless two-photon voltage imaging of sub-threshold depolarisations in hippocampal organotypic slices.** Recording sub-threshold depolarisations in JEDI-2P-Kv expressing hippocampal organotypic slices using 2P-TF-GPC. **a** Command voltage steps used to change the membrane potential of patched neurons. **b** Average fluorescence traces recorded from neurons after 50 trials for different magnitudes of sub-threshold depolarisations ranging between 0 and 2.5 mV. Traces were recorded at an acquisition rate of 1 kHz and 1.1 mW μm⁻². **c** Average -%ΔF/F₀ and (**d**) SNR of the fluorescence response to different sub-

threshold changes of membrane potential plotted as a function of number of repeats. The 95% confidence interval is also plotted (shaded region). Sub-threshold depolarisations <2.5 mV cannot be reliably resolved in single trials using 2P-TF-GPC and JEDI-2P-Kv, however after 25 trials depolarisations greater than or equal to 1 mV can be resolved. Data was acquired from $n = 6$ neurons, from 2 slices from 1 slice culture (Refer to Supplementary Fig. 1 and Supplementary Tables 1 and 2). Source data are provided as a Source Data file.

sample with the average powers necessary to achieve high SNR recordings induces a peak temperature rise of ~4.5 K, centred on the illumination spot (Supplementary Fig. 9a, 940 nm excitation, 80 MHz repetition rate, 75 mW per target, 0.66 mW μm⁻²). The temperature increase was found to occur rapidly ($\Delta T = 3$ K after 12.5 ms, Supplementary Fig. 9b), and to extend 20–30 μm beyond the targeted neuron (Supplementary Fig. 9c). Since heating due to light absorption is a linear process, the use of higher powers results in larger, faster temperature rises (>8 K for 150 mW per target, 1.33 mW μm⁻², Supplementary Fig. 9a–c). Since temperature rises of this magnitude are much larger than those observed physiologically[78], and occur on much faster timescales[79,80], the physiological perturbations observed during scanless 2P voltage imaging at average powers exceeding 125 mW (1.11 mW μm⁻²) can reasonably be attributed to light-induced heating. However, further investigation is necessary to discern the exact mechanisms, given the large differences observed between the rise times of the temperature increases predicted by the theoretical model and the experimental observations of the onset of physiological perturbations.

Collectively, these results demonstrate that scanless 2P voltage imaging can be used to acquire high SNR recordings from single neurons in superficial layers of scattering tissue using high-repetition rate lasers tuned to the peak of the action spectrum of state-of-the-art 2P indicators. We find that an acquisition rate of 500 Hz is sufficient to capture most relevant aspects of neural activity using JEDI-2P-Kv, which permits acquiring data throughout a 2x larger FOV as compared with 1 kHz for commercially available sCMOS cameras ("Methods"), and permits the use of lower average powers. Sufficient SNR for robust AP detection was obtained using 75 mW per cell (0.66 mW μm⁻²), but achieving the same for multiple cells simultaneously, particularly

at depths beyond one scattering length would necessitate delivering higher average powers to the sample. We found that powers exceeding 125 mW per neuron (1.11 mW μm⁻²) induced physiological perturbations, most likely due to sample heating. Consequently, performing such experiments requires alternative strategies to reduce the average illumination power delivered to the sample, as explored in the following section.

## Scanless two-photon voltage imaging of multiple targets with low repetition rate 1030 nm sources

One approach to reducing the average power required for multi-target scanless 2P voltage imaging in scattering tissue is to use low repetition rate, industrial light sources, such as Ytterbium-doped fibre lasers, which provide much higher pulse energies (~μJ) and output powers than the sources used for conventional 2P imaging (~nJ). While a similar approach has proven successful in the context of 2P optogenetics[75,81–83], these lasers are commonly fixed wavelength sources (1030–1064 nm), and hence are not well suited for use with GFP-based fluorescent indicators with absorption peaks typically centred at 920 nm. However, since the excitation spectrum of JEDI-2P-Kv is slightly red-shifted as compared with previous GFP-based voltage indicators (Supplementary Fig. 10a)[54], we decided to test whether it was feasible to record activity from neurons expressing JEDI-2P-Kv using scanless 2P voltage imaging with 1030 nm excitation. We first repeated protocols 1, 2 and 3 (Supplementary Table 5), and recorded electrically evoked APs from single neurons in hippocampal organotypic slices by exciting fluorescence with 2P-TF-CGH at 940 and 1030 nm using a tuneable, 80 MHz repetition-rate laser. We found that the SNR of data obtained at 1030 nm was 40% lower than that obtained at 940 nm using the same excitation photon flux

(6.29 × 10$^{27}$ photons s$^{-1}$ μm$^{-2}$, Supplementary Fig. 10c(ii), upper, 7.4 ± 2.0 vs 12.4 ± 2.9, $p = 0.04931$), but did not find any statistical differences in the -%ΔF/F$_0$ of 100 ms, 100 mV depolarisations or APs between 1030 and 940 nm excitation (Supplementary Fig. 10c(ii), lower, 24 ± 1 vs 30 ± 6, $p = 0.40812$). We also measured the F-V curve of JEDI-2P-Kv at 1030 nm at different light intensities. We did not observe significant differences in the form of the F-V curve between 940 and 1030 nm excitation (Supplementary Fig. 10b(ii) and Fig. 10d(i)). However, we found that the F-V curve varied as a function of peak excitation intensity: higher peak intensities yielded smaller -%ΔF/F$_0$ changes for large voltage steps (Supplementary Figs. 10d and e: 28 ± 2 (0.145 GW mm$^{-2}$) vs 39 ± 2 (0.03 GW mm$^{-2}$) per 100 mV step, $p = 0.00608$). Even at the highest peak intensity tested, the reduced -%ΔF/F$_0$ was sufficient for resolving APs[41,42,53,54]. These experiments established the feasibility of performing scanless 2P voltage imaging by exciting JEDI-2P-Kv at 1030 nm using low repetition rate lasers providing higher pulse energies.

Next, we set out to identify the optimal conditions for achieving high SNR scanless 2P voltage imaging recordings using low repetition rate sources with (fixed) 1030 nm outputs. These experiments were performed using protocol 5 without electrically evoking APs. Specifically, we performed scanless 2P voltage imaging recordings from neurons in hippocampal organotypic slices (2P-TF-CGH, continuous 30 s illumination), whilst varying the repetition rate and average power. These powers were selected as corresponding to those used in the previous sections, accounting for differences in pulse energy, excitation efficiency and collection efficiency (Supplementary Tables 2-3, "Methods"). We found that the SNR increased both as a function of inter-pulse interval and average power (Supplementary Fig. 10f). Sufficient SNR for 2P voltage imaging recordings (≥ 5) was obtained for all repetition rates at average powers of 10.5 mW. We set an SNR threshold of 5, as our analysis pipeline could reliably detect most APs with SNR above this threshold. The highest SNR data was obtained using a repetition rate of 250 kHz, although this condition produced the greatest photobleaching of all cases tested (Supplementary Fig. 10f). Additionally, photobleaching increased significantly more rapidly as a function of average power for a repetition rate of 250 kHz than for the other repetition rates (Supplementary Fig. 10f). A laser repetition rate of 250 kHz also induced photodamage, observed as a sudden, irreversible increase in fluorescence similar to that described in ref. 84 (termed non-linear damage). Photodamage occured in 12% of neurons, on average 6.1 ± 2.3 s following the onset of illumination (> 15 mW per target, Supplementary Fig. 10f). We observed non-linear damage in 1 of 49 neurons (2%, average power >15 mW per cell, Supplementary Fig. 10f) when using a repetition rate of 500 kHz. Equivalent photodamage was not observed at any powers tested for repetition rates above 500 kHz. Consequently, we determined that using a repetition rate of 500 kHz provided the best compromise between maximising SNR (and hence number of neurons imaged simultaneously) at a given average power, whilst minimising any photoinduced physiological perturbations during 30 s, high-duty cycle recordings.

We then evoked APs electrically by manual current injection at random timepoints during each recording (protocol 5, Supplementary Table 5). We found that we could obtain single-cell recordings with comparable -%ΔF/F$_0$ and SNR per AP to the data obtained at 940 nm (80 MHz repetition rate) using approximately an order of magnitude lower average powers (-%ΔF/F$_0$: 30.0 ± 0.3, SNR: 11.3 ± 0.3, Fig. 5a, b).

In-silico modelling confirmed that these conditions (reduced average powers) induced much lower, localised, peak temperature rises (<0.74 K) for continuous 30 s recordings than observed previously (Supplementary Fig. 11a−c). We tested these predictions experimentally by repeating protocol 4 (Supplementary Fig. 12a and Supplementary Table 5) to detect any changes in the membrane properties of the targeted cells. In contrast with the 940 nm, 80 MHz

results, we did not find any statistically significant differences in 5 of 6 monitored properties at any powers used (Supplementary Fig. 12b−g). We observed small differences in measurements of the AP half-width between the illuminated and control cells (Supplementary Fig. 12f). The action potential half-width was decreased as a function of increasing power for the illuminated cells (ΔHalf-width = −0.037 ± 0.012 ms, group 1 and −0.131 ± 0.015 ms, group 2). However, since the magnitude of this change was smaller than the spread of the values for the control cells (inset, Supplementary Fig. 12f) and an order of magnitude smaller than what was observed in the case of the high-repetition rate laser (Supplementary Fig. 8f), we conclude that scanless 2P voltage imaging with low-repetition rate lasers induces much smaller (if any) physiological perturbations at the powers required to perform high SNR imaging of single neurons.

Next, we took advantage of high-power, low repetition-rate lasers combined with holographic multiplexing to perform scanless 2P voltage imaging of multiple targets simultaneously. We repeated protocol 5 ("Methods" section, Supplementary Table 5) and recorded fluorescence from a targeted, patched cell whilst 15 additional, randomly distributed, spots were positioned within the FOV (7.5−12.5 mW per target, Fig. 5c, d). According to simulations, these conditions limit light-induced temperature rises to ~3 K, as measured at the centre of the FOV (Supplementary Fig. 11d, e). The high SNR multi-target recordings obtained under these conditions (-%ΔF/F$_0$ per AP: 21.0 ± 0.2, SNR per AP: 9.0 ± 0.2, Fig. 5c, d and Supplementary Fig. 13) provided excellent spike detection reliability (precision = 0.939, recall = 0.961, F1 = 0.946, Fig. 5e(i-iii), see "Methods" section for definitions of precision, recall and F1 scores) using an SNR threshold of 5 (96% detection rate, 380/394 APs detected). These results demonstrate that scanless 2P voltage imaging can be used to obtain high SNR multi-cell recordings from densely labelled in vitro preparations.

## Scanless two-photon voltage imaging in vivo

Finally, we examined the feasibility of performing scanless 2P imaging in vivo. We injected a JEDI-2P-Kv encoding adeno-associated virus (AAV) into layer 2/3 of the barrel cortex of mice (7–8 weeks old, "Methods"). Experiments were performed between 3–10 weeks post-injection using head-fixed, anaesthetised mice ("Methods"). Cranial windows were installed on the day of each experiment (Fig. 6a). Stable pan-neuronal JEDI-2P-Kv expression, under a hSynapsin promoter, was observed in approximately 30% of neurons in cortical layers 1 and 2/3 of the barrel cortex (Fig. 6b). The expression of JEDI-2P-Kv was found to be well-localised to the membrane of the cell body and proximal dendrites (Fig. 6b). Expression of JEDI-2P-Kv was much sparser in layer 2/3 of the barrel cortex in vivo (783 ± 85 cells/mm$^2$) than in the dentate gyrus in hippocampal organotypic slices (4124 ± 964 cells/mm$^2$) ($n = 6$ representative FOVs). During each experimental session, reference images were acquired by collecting fluorescence excited by galvo-galvo raster-scanning of a diffraction-limited (920 nm) spot with an sCMOS camera. For these proof-of-concept experiments, this configuration eliminated the possibility of any registration mismatches between imaging modalities (for instance, due to the tilt of the coverslip used for the cranial window[85]), but also limited the maximum imaging depth from which we could obtain reference images to 250−275 μm due to a reduction of image contrast and signal-to-background ratio (SBR). We performed single and multi-cell scanless 2P voltage imaging of cells manually selected from each reference image. We used 17-μm diameter spots to minimise the disruption of small motion. Overall, we imaged 203 neurons in 43 fields of view in 7 mice (Supplementary Fig. 14a), recording spontaneous activity for 30 s with a 500-Hz acquisition rate in each case (Fig. 6c, d and Supplementary Fig. 14b).

We found that we could clearly identify and distinguish APs in spike trains of up to 50 Hz (Fig. 6d, inset) from neurons at depths up to 250 μm in cortical brain tissue (Fig. 6c−e and Supplementary Fig. 14b)

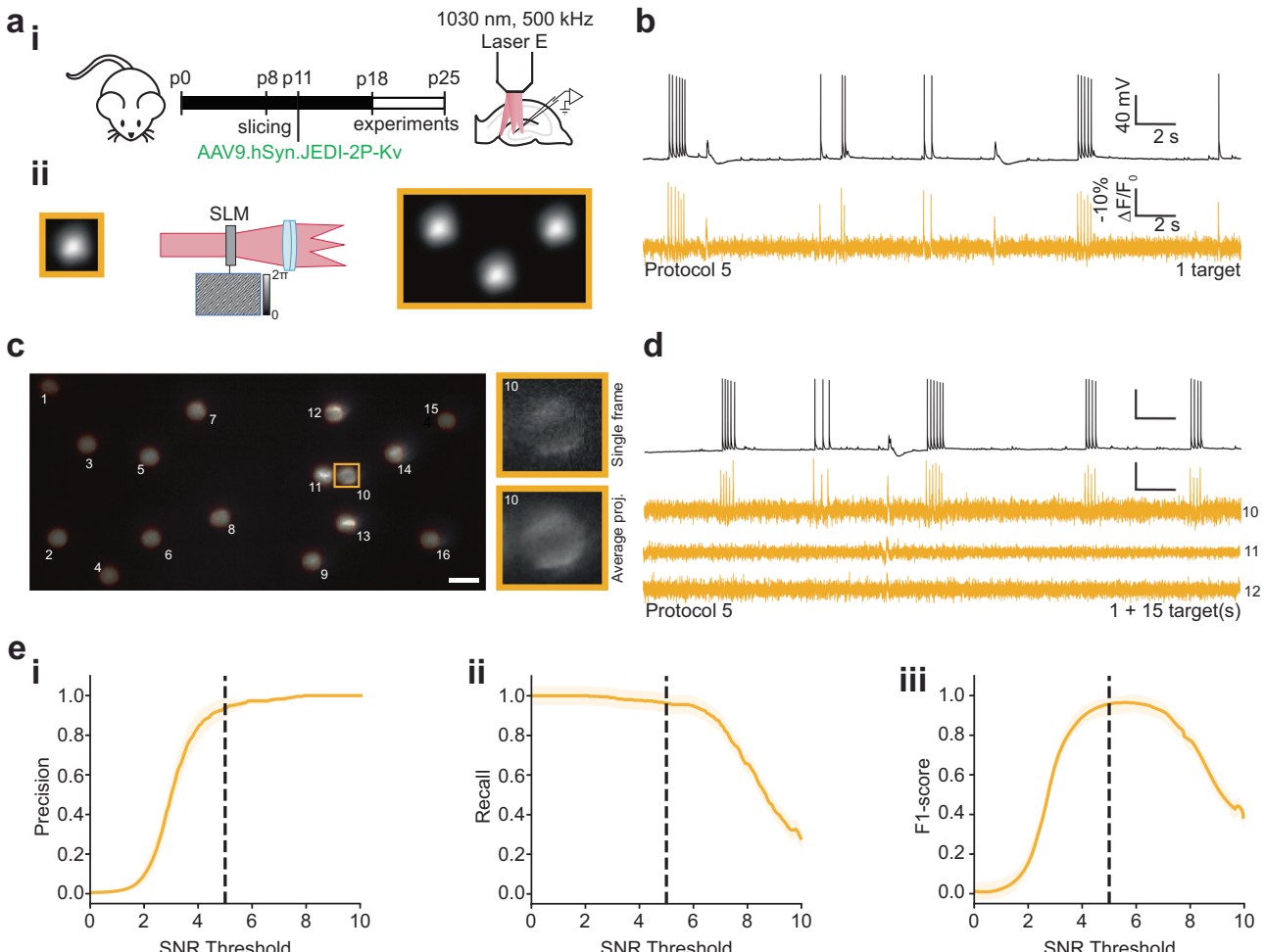

**Fig. 5 | Multi-target scanless two-photon voltage imaging using low repetition rate sources at 1030 nm. a** (i) Schematic representation of the protocol used to prepare JEDI-2P-Kv expressing hippocampal organotypic slices (see "Methods" section, Supplementary Table 2). (ii) Multiple cells were illuminated simultaneously by multiplexing a temporally focused (TF) Gaussian beam with a Spatial Light Modulator (SLM, "Methods", Supplementary Table 2). All experiments were performed using 17 μm TF-Gaussian spots at 1030 nm (laser E, 500 kHz repetition rate, power densities: 0.03–0.06 mW μm$^{-2}$). Data was acquired for 30 s at an acquisition rate of 500 Hz with camera A. **b** Simultaneous current-clamp (upper, black) and fluorescence recordings (lower, yellow) of electrically evoked activity in neurons from hippocampal organotypic slices (protocol 5, "Methods" and Supplementary Table 5). **c** Average (temporal) projection of data acquired during a representative multi-target scanless 2P voltage imaging experiment (grey lookup-table). The spot positions have been overlayed (yellow lookup-table) and numbered 1-16. The scale bar represents 20 μm. The patched neuron (cell 10) is

indicated by the yellow box. Inset: single frame and average temporal projection of a zoomed portion of the dataset containing the patched cell. **d** Simultaneous current-clamp (upper, black) and fluorescence recordings (lower, yellow) of electrically evoked activity in neurons from hippocampal organotypic slices. Data acquired by illuminating the same patched cell plus fifteen randomly distributed positions (1 + 15) as shown in **c**. The number to the left of each trace indicates the index of the targeted cell as labelled in **c**. **e** (i) Precision, (ii) recall and (iii) F1 score of action potential detection plotted as a function of SNR threshold. The 95% confidence intervals are also plotted (shaded region). Refer to the "Methods" section for definitions of these terms. Simultaneously acquired whole-cell patch-clamp electrophysiology traces were used as ground-truth datasets. Of 380 electrically evoked action potentials, 96% were detected. The SNR threshold of 5, used throughout this manuscript, is indicated by the black dashed line ($n = 15$ neurons, from 3 different slices from 2 independent slice cultures). Source data are provided as a Source Data file.

using an average power of 8.3 mW per target, corrected for loss of peak intensity due to scattering. Additionally, even in multi-target experiments performed deep in scattering tissue, individual neurons could still be identified in average projections of the dataset (inset, Fig. 6e and Supplementary Fig. 14c). On the other hand, we found that the signal-to-background ratio (SBR)[30] of scanless 2P voltage imaging data decreased sharply as a function of depth for the multi-target experiments (−0.471 per μm, Supplementary Fig. 14d) due to scattering of both the infrared targeted excitation and fluorescence (Supplementary Fig. 14a, b). Even for the deepest neurons recorded, we found that the FWHM of the correlation image was less than the average separation between targeted neurons in our experiments (<70 μm vs 86 μm, Supplementary Fig. 14a). In addition, since we found that the average -%ΔF/F$_0$ per AP (32 ± 1) only slightly decreased as a function of depth (−0.1% per μm,

Fig. 6f), we conclude that crosstalk is not currently a limiting factor for scanless 2P voltage imaging in vivo at the investigated depth (250 μm), despite the use of camera detection.

Since the amount of power required to generate sufficient 2P excitation for high SNR voltage imaging increases exponentially as a function of depth, the number of neurons that can be targeted simultaneously decreases (Supplementary Fig. 15a, b). We found that the maximum number of neurons that could be imaged simultaneously in vivo, beyond one scattering length, was limited by light-induced temperature increases rather than crosstalk between targets due to camera detection, or insufficient output laser power. Specifically, at 150 μm below the brain surface, simulations suggest that a maximum of 10–15 neurons could be targeted simultaneously using the powers currently required (Supplementary Fig. 15c, d).

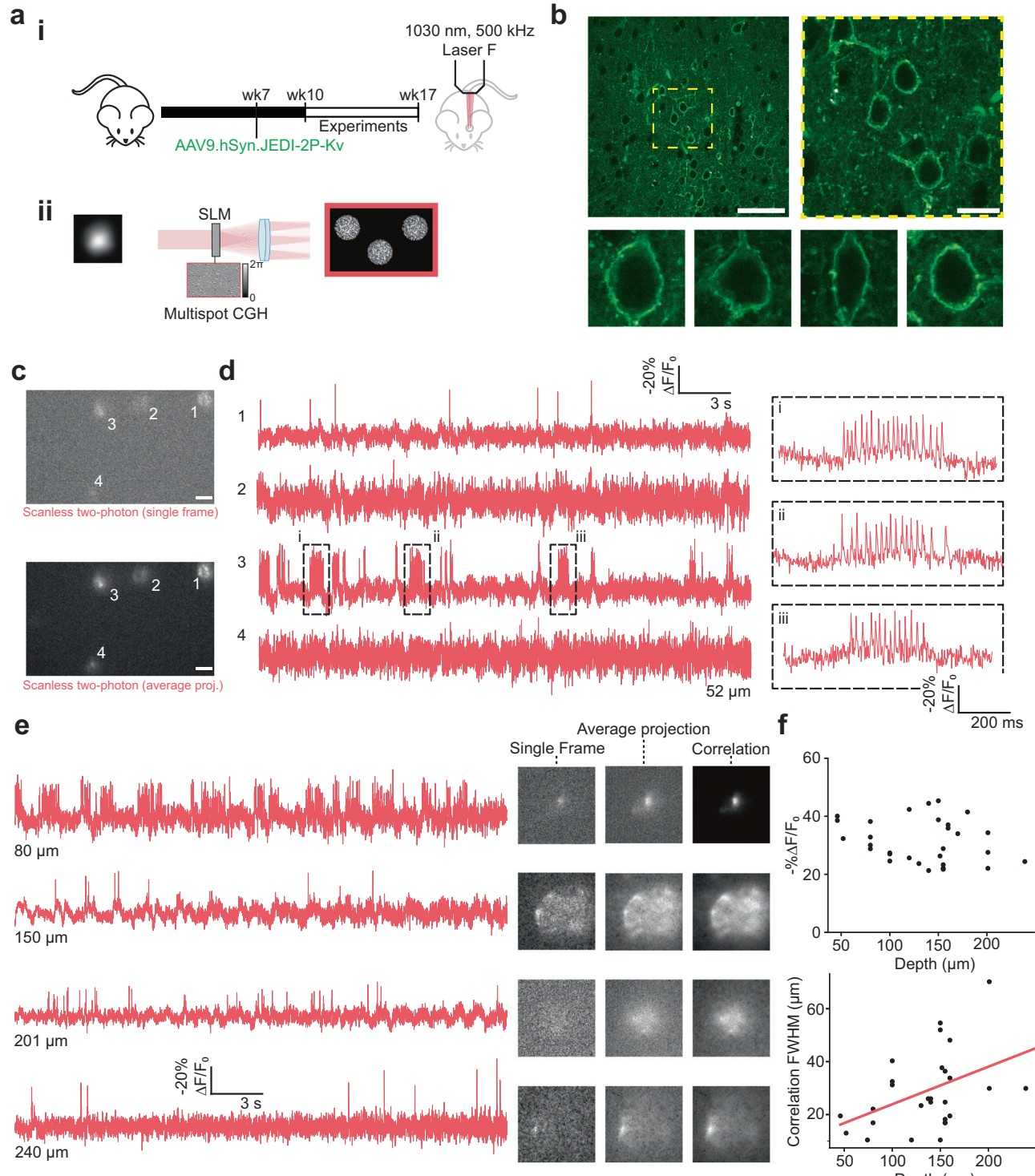

## Simultaneous scanless two-photon voltage imaging and photostimulation

Since the optical configurations used for scanless voltage imaging in this study are commonly used in 2P optogenetic photostimulation[56], we reasoned that it ought to be possible to perform simultaneous scanless 2P voltage imaging and photostimulation. Here, we present a specific configuration where this "single-beam" approach is exploited to monitor the number, timing and synchrony of light-evoked APs. We co-expressed the sensitive opsin ChroME-ST, tagged with a nuclear-localised blue fluorescent protein (H2B-BFP), and JEDI-2P-Kv in neurons in hippocampal organotypic slices by bulk

transduction of two AAV vectors (Fig. 7a, "Methods"). Co-expressing neurons were identified by membrane expression of JEDI-2P-Kv and BFP-positive nuclei (Fig. 7b). Since photostimulation protocols typically have shorter total illumination times (hundreds of ms) and lower duty-cycles (<10%) we were able to use a repetition rate of 250 kHz (940 nm, Supplementary Table 1) without observing any of the non-linear damage described previously in the context of the continuous 30 s recordings.

We capitalised on the overlapping excitation spectra of ChroME-ST and JEDI-2P-Kv to perform single-beam photostimulation and voltage imaging. We used whole-cell patch-clamp to confirm that optically

**Fig. 6 | Scanless two-photon voltage imaging in vivo. a** (i) Schematic representation of the protocol used to prepare mice expressing JEDI-2P-Kv for in vivo scanless 2P voltage imaging (see "Methods" section). (ii) Several neurons were illuminated simultaneously by generating multiple holographic spots using holography (CGH) ("Methods" section, Supplementary Table 1). All experiments were performed using 17 μm TF-CGH spots at 1030 nm (laser F, 500 kHz repetition rate, power densities: 0.02–0.07 mW μm$^{-2}$). Data was acquired for 30 s at an acquisition rate of 500 Hz with camera A. **b** Upper: representative images of JEDI-2P-Kv expression in L2/3 of the barrel cortex at different scales ($n = 4$ mice). Yellow dashed box indicates the zoomed in region, shown on the right panel. Scale bars from left to right represent 50 μm and 15 μm respectively. Lower: confocal images of single representative JEDI-2P-Kv expressing neurons in L2/3 of the barrel cortex, taken from multiple mice ($n = 4$ mice). **c** Single frame and average (temporal) projection of data acquired during a representative in vivo multi-target scanless 2P voltage imaging experiment (power density: 0.02 mW μm$^{-2}$, depth: 52 μm below the dura). Scale bar represents 20 μm. **d** Fluorescence traces from the cells identified in **b**, as numbered. Responses are reported as the fluorescence change (ΔF), expressed as a percentage of the baseline fluorescence F0 (-%ΔF/F$_0$). Inset: zoomed in portion of the bursts of APs which have an average firing rate of 50 Hz. **e** Left: fluorescence traces from cells acquired from different experiments at different depths below the dura (80–240 μm, as indicated). Right: single frames, temporal projections and correlation images (see "Methods" section) are also provided for reference. **f** Characterisation of the -%ΔF/F$_0$ and correlation FWHM (see "Methods" section), plotted as a function of target depth below the cortical surface. Individual points (black) represent the average value for a single (responsive) target during a single experiment ($n = 43$ fields of view (30 independent) from 7 mice). A linear, least-squares, fit to the experimental data is displayed in red. Source data are provided as a Source Data file.

evoked APs could be detected using scanless 2P voltage imaging (Fig. 7c, d). We found that, for 15 ms illumination times and power densities above 0.02 mW μm$^{-2}$, we were able to detect APs in single trials and measured similar latencies to those obtained using whole-cell patch-clamp recordings (4.3 ± 0.2 ms) (Supplementary Fig. 16a). The AP probability decreased as a function of stimulation frequency, although in some cases, it was possible to stimulate and image APs at 50 Hz (Supplementary Fig. 16b). For power densities between 0.02 and 0.08 mW μm$^{-2}$ (2.5–9 mW per cell), we found that the average -%ΔF/F$_0$ of optically evoked APs was 20 ± 8% (Supplementary Fig. 16c).

According to simulations, these low-duty cycle, low average power conditions induce drastically lower light-induced temperature rises (Supplementary Fig. 17). Furthermore, since the majority of heat is dissipated between consecutive illumination epochs, there is negligible inter-pulse cross-talk (Supplementary Fig. 17). These results indicate that single-beam, scanless 2P voltage imaging and photostimulation could be performed in many cells simultaneously using high-energy, low-repetition rate lasers, whilst maintaining powers well below the damage threshold.

## Discussion

In this work, we introduced scanless 2P voltage imaging with camera detection and performed high SNR voltage imaging of single and multiple neurons expressing the recently developed GEVI JEDI-2P-Kv. We performed a thorough characterisation of three, temporally focused, parallel excitation modalities (2P-TF-GPC, 2P-TF-Gaussian and 2P-TF-CGH) for scanless 2P voltage imaging in vitro and compared their performance. We found that 2P-TF-Gaussian illumination is the easiest and most cost-effective approach to implement. However, it is the least photon-efficient approach due to the necessity of cropping approximately one-third of the excitation spot to achieve a sharp lateral profile. Conversely, the photon-dense speckle grains in 2P-TF-CGH spots result in efficient 2P excitation, hence the highest SNR for a given excitation power. 2P-TF-CGH is thus the optimal modality in scenarios with limited power budgets. 2P-TF-GPC features a sharp top-hat profile and homogeneous light distribution and can be used to sculpt light into well-defined lateral shapes[56], which ought to enable precise targeting of the most responsive regions of the cell membrane or for sculpted, scanless 2P voltage imaging of thin subcellular processes.

We used 2P-TF-GPC to demonstrate many of the advantages of imaging neural activity with GEVIs by imaging APs (single-trial), subthreshold depolarisations and resolving single APs in high-frequency spike trains up to 125 Hz. We performed simultaneous imaging and electrophysiology in vitro to comprehensively characterise the performance of, and optimise scanless 2P voltage imaging with the genetically encoded indicator JEDI-2P-Kv. Consistent with other reports[42], we found that it was generally possible to reduce the imaging speed down to 500 Hz (and consequently the required power) without a critical loss in the ability to determine the number and timing of APs. We found that imaging small sub-threshold signals required averaging

data from up to 25 individual trials to reach an SNR above 1 for depolarisations larger than 0.5 mV. Whilst these findings highlight the challenges of detecting subthreshold unitary PSPs (<2.5 mV) in vivo with current state-of-the-art GEVIs under 2P excitation, they also indicate that such experiments ought to be possible using GEVIs optimised for sub-threshold voltage detection.

Due to maximized dwell times, scanless 2P voltage imaging requires much lower peak intensities than other contemporary 2P voltage imaging techniques (Supplementary Table 6). We found that the peak powers required to obtain high SNR traces were one- to two-orders of magnitude lower than previously reported non-linear damage thresholds[74,75]. However, the larger excitation areas associated with scanless 2P excitation necessitate higher average powers, which risk inducing perturbative temperature rises. Using a combination of simulations and experiments, we found that single cells could be imaged continuously for 30 s using 100 fs, 80 MHz sources, using average powers up to 125 mW per neuron, and that these conditions did not induce any other observable changes in AP parameters or membrane properties.

We demonstrated that the red-shifted absorption shoulder of JEDI-2P-Kv enables scanless 2P voltage imaging under 1030 nm excitation. The key advantage of this excitation wavelength is the availability of high-power, low-repetition rate industrial sources, already widely used for 2P optogenetics[75,81]. We characterised scanless 2P voltage imaging using such sources in vitro. We tested laser repetition rates between 250 and 2000 kHz and determined that a repetition rate of 500 kHz provided the best compromise between maximising SNR (at a given average power) and minimising photoinduced physiological perturbations. For our experiments, we found that using a 500 kHz repetition rate enabled us to reduce the average power required for scanless 2P voltage imaging by an order of magnitude as compared with conventional 80 MHz sources tuned to 940 nm, as typically used for 2P-LSM. Hence, we could target 15 neurons simultaneously in vitro.

Using the same laser source, we performed in vivo scanless 2P voltage imaging of single and multiple neurons expressing JEDI-2P-Kv at depths up to 250 μm below the cortical surface. We used a pan-neuronal promoter to achieve higher densities of fluorescently labelled cells than used in recent 1P studies[21] (783 ± 85 cells mm$^{-2}$ vs <370 cells mm$^{-2}$), we found that the -%ΔF/F$_0$ per AP only slightly decreased as a function of depth, thanks to the confinement of excitation to the target conferred by temporal focusing[57]. The maximum achievable imaging depth in these experiments was limited by the approach we used to acquire the reference images, which ought to be rectified using similar approaches as for other systems[83,86,87]. We will characterise the maximum imaging depths achievable for single-cell camera-based scanless 2P voltage imaging in future studies.

Our modelling results suggest that the primary factor limiting the number of cells addressable in scanless 2P voltage imaging in vivo beyond one scattering length is heating, since it is necessary to increase the average power delivered to the sample to compensate for

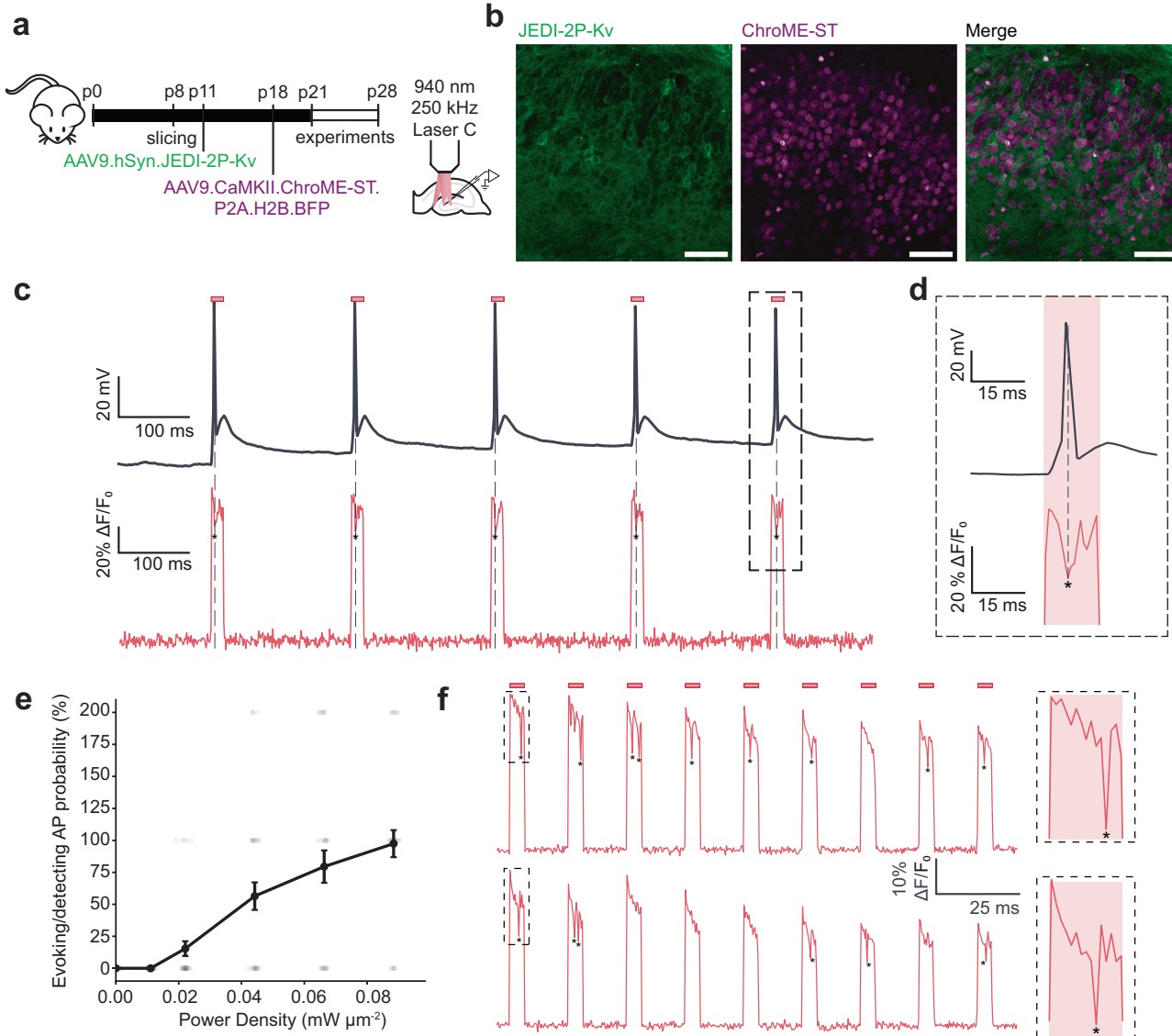

**Fig. 7 | Fluorescence recordings of photo-evoked action potentials in hippocampal organotypic slices co-expressing JEDI-2P-Kv and ChroME-ST, using 2P-TF-CGH. a** Schematic overview of the experimental protocol. JEDI-2P-Kv expressing organotypic hippocampal slices were prepared as described (for more details, refer to Fig. 3 and "Methods" section). At p18, slices were transduced with AAV9.CaMKII.ChroME-ST.P2A.H2B.BFP. Experiments were performed between p21 and p28 using a 940 nm, 250 kHz repetition rate laser source. **b** Cross-sections of hippocampal organotypic slices co-expressing the genetically encoded voltage indicator JEDI-2P-Kv and the soma-targeted channelrhodopsin ChroME-ST in the dentate gyrus ($n = 4$ slices from 2 independent slice cultures). Channelrhodopsin-expressing cells were identified according to their nuclear-localised fluorescence (see "Methods" section). Scale bar represents 50 μm. **c** Simultaneous optical and electrophysiological recordings demonstrating that APs can be evoked and imaged using a single excitation spot (12 μm diameter, power density 0.02 mW μm$^{-2}$ (2.5 mW per cell), 15 ms strobed illumination at 5 Hz. The red bar represents the illumination epoch. Asterisks indicate detected APs. **d** Zoom on simultaneous optical and electrophysiological recordings of one AP. **e** Probability of evoking and recording APs as a function of power density. AP probability is calculated as the number of APs evoked and detected at each power (power density: 0.01–0.09 mW μm$^{-2}$). Data is plotted as mean ± SEM of recordings obtained for 33 repetitions. Probabilities greater than 100 % indicate that more than one action potential was detected in a given trial. **f** Simultaneous optogenetic activation of ChroME-ST and voltage imaging of JEDI-2P-Kv. 10 cells were targeted simultaneously using 2P-TF-CGH and imaged at 500 Hz. Here we show an example trace from one cell when the 10 cells were targeted sequentially (upper panel) or simultaneously (lower panel). In both the sequential and multi-cell acquisitions, APs were only evoked/recorded in the cells co-expressing JEDI-2P-Kv and ChroME-ST. All data was acquired using laser C (940 nm, 250 kHz repetition rate) and camera A. Source data are provided as a Source Data file.

the reduction of peak excitation power due to excitation light scattering. To overcome this limitation, and target a greater number of cells simultaneously, it ought to be possible to exploit recent deep learning approaches that leverage spatiotemporal correlations in imaging data to perform denoising[42]. Furthermore, we anticipate that we will be able to reduce the average power requirements of scanless 2P voltage imaging in the future using brighter, more sensitive, red-shifted voltage indicators with action spectra which peak at 1030 nm.

Alternatively, deeper brain structures could be imaged using graded index lenses (GRIN) lenses[88,89] or emerging computation approaches capable of overcoming scattering-induced ambiguity and de-mixing the fluorescent transients emanating from different sources[90].

By co-expressing the opsins ChroME-ST and the voltage indicator JEDI-2P-Kv in neurons, we performed single-beam voltage imaging of photo-evoked APs. Real-time optical monitoring of the properties of photo-evoked APs (such as the latency and jitter) would be valuable in

cases where such data can only be obtained through lengthy electro-physiological sessions[83,91] or for characterising the properties of cell populations that are challenging to access using conventional patch-clamp techniques[92]. Additionally, this single-beam configuration could be adopted during connectivity mapping experiments to retrieve the occurrence and precise timing of evoked APs in pre-synaptic cells. While similar experiments have been performed previously using GCaMP[93,94], higher precision could be obtained by substituting GCaMP with a voltage indicator. Extending the applicability of these experiments to encompass orthogonal 2P all-optical manipulation and monitoring within a single neuronal population requires expanding the optogenetic toolbox to include efficient red-shifted GEVIs and blue-shifted channelrhodopsins.

Using an SLM for holographic light multiplexing of the temporally focused, sculpted light resulted in a nominal FOV of $300 \times 300$ μm². However, the effective FOV used for individual high-speed voltage imaging recordings was reduced in one dimension due to the maximum number of rows of pixels that could be read out at a given acquisition rate. An alternative approach would be to replace the sCMOS sensor with a detector capable of higher (full frame) readout speeds such as single-photon avalanche diode (SPAD) arrays[95], or multiple cameras[96]. The FOV of scanless 2P voltage imaging could also be trivially increased by reducing the magnification of the detection axis. However, since this approach would proportionally reduce the spatial resolution, it is not suitable for all applications, such as recording voltage fluctuations in smaller sub-compartments of neurons such as spines.

We anticipate that the description and thorough characterisation of scanless 2P voltage imaging presented in this manuscript will motivate its application to decipher the logic and syntax of neural circuits.

## Methods

### Experimental setup for performing two-photon voltage imaging with temporally focused, sculpted light

All two-photon voltage imaging presented in this manuscript was performed using 3 optical setups with excitation optical paths as shown in Supplementary Fig. 1. Each setup was equipped with one or more ultrafast laser sources (Supplementary Tables 1-3). Five different laser sources were used to acquire all data presented, referred to as Lasers A-E respectively throughout the manuscript. The source used to acquire each dataset is specified in each case. All experimental configurations used to acquire the data presented in each figure are summarised in Supplementary Tables 1–4. Laser A refers to a tuneable femtosecond source (Coherent Discovery, ~1 W, 80 MHz, 100 fs) tuned to 920, 940 or 1030 nm (as specified). Laser B refers to a femtosecond source with a fixed wavelength output (Spark Alcor, 4 W, 80 MHz, 100 fs, 920 nm). Laser C refers to a custom OPA pumped by an amplified laser, also with fixed wavelength output (Amplitude Satsuma Niji, 0.2–0.6 W, 250 kHz, 100 fs, 940 nm). Laser D refers to a femtosecond fixed wavelength source (Amplitude Goji, 40 MHz, 5 W, 150 fs, 1030 nm). Lasers E, F refer to a femtosecond fibre amplifier with fixed wavelength output, high pulse energies (μJ) and variable repetition rates (Amplitude Satsuma HP2 or HP3, 0–2 MHz, 20 or 50 W max average power respectively, 300 fs, 1030 nm). The repetition rate used was specified in each case.

Setup 1 consisted of 2 optical pathways built around laser sources A, B, C and D. Path 1 was used to generate temporally focused, multiplexed, GPC (12 μm FWHM, 2PE) according to the experimental setup described by Papagiakoumou et al.[56], or low-NA, apertured, Gaussian beams (12 μm lateral FWHM, 2PE) respectively (upper path, Supplementary Fig. 1). Path 2 (lower path, Supplementary Fig. 1) was used to generate temporally focused holographic discs (12 μm lateral FWHM, 2PE), according to the experimental setup described by Papagiakoumou et al.[66]. By virtue of the removable mirrors indicated in

Supplementary Fig. 1, light from each of the lasers could be directed through path 1 or path 2 during a given experiment as indicated by the dashed lines in Supplementary Fig. 1. The output of lasers A, B and D was modulated by a mechanical shutter (Thorlabs, SH05R/M) or using a high-speed modulator (Thorlabs, OM6NH/M) whereas the output of laser C was gated directly. In all cases the external trigger was a TTL signal generated by pCLAMP (Molecular Devices, Sunnyvale, CA) controlling an acquisition system (Molecular Devices, Axon Digi-data 1550B).

In path 1, a telescope formed of two lenses (L1, $f = 80$ mm and L2, $f = 300$ mm, for GPC and L1, $f = 80$ mm, and L2, $f = 200$ mm, for low-NA Gaussian illumination) was used to expand and project the beam onto a spatial light modulator (SLM1) (See Supplementary Table 1 for the exact reference of the optical components). In the case of GPC, SLM1 was used to apply a π phase shift to the portion of the beam overlapping with the circular spot and a phase shift of zero elsewhere. The modulated beam was Fourier transformed by L3, resulting in a spatial displacement between the high and low spatial frequency components of the field in the Fourier plane. The low spatial frequency components were selectively phase shifted by π using a phase contrast filter (PCF) with 60 μm radius (Double Helix Optics, custom design) positioned in the Fourier plane of L3. Spots with top-hat profiles were generated by constructive interference between the recombined high and low spatial frequencies in the focal plane of L4. For more details, refer to ref. 56. For parallel 2P excitation using a low-NA Gaussian beam, the corrective phase mask provided by the manufacturer was displayed on SLM1 and the PCF was displaced from the optical path as indicated in Fig. 1. A blazed diffraction grating (Richardson Gratings, 600 lines/mm) located at the focal plane of L4, a conjugate image plane, dispersed the different spectral frequency components of the ultrafast beam as required for temporal focusing. The grating was oriented at the blaze angle to maximise light throughput. The dispersed beam was collimated in one direction and Fourier transformed in the orthogonal direction by L5, resulting in an asymmetric "line" illumination of SLM2, as described previously[67]. For this work, SLM2 was used for 2-dimensional multiplexing of the beam, although 3-dimensional multiplexing is possible. Phase masks were generated using a weighted Gerchberg-Saxton algorithm as described previously[97,98]. For all data presented in Figs. 1–3, SLM2 was used to displace the sculpted light from the optical axis and the zeroth-diffraction order, which was removed using a physical beam block positioned in a conjugate image plane. Lenses 6 and 7 were used to de-magnify the beam onto the back focal plane of the objective lens which projected the light (and recombined the different spectral frequency components) onto the focal plane. A half-wave plate was included downstream of SLM2 in path 1 to convert s-polarised light to p-polarised.

In path 2, a Galilean beam expander formed of two lenses (L8 and L9), expanded the beam onto SLM3. For the single-cell experiments, 5 holograms designed to generate a single 12 μm holographic spot (located in the same position) were computed prior to each session. For each recording, one of these phase masks was randomly selected and displayed on the SLM in order to minimise any effects of the variable speckle distribution on the resulting dataset. The holograms displayed on SLM3 were designed to generate multiple 12 μm holographic spots targeted to chosen neurons throughout the field of excitation following the calibration procedure outlined in the Supplementary Information. All holograms were calculated using an iterative Gerchberg-Saxton algorithm[99]. The zeroth diffraction order was removed using a physical beam block positioned in a conjugate image plane. The modulated beam was Fourier transformed by L10 to form the holographic discs in a conjugate image plane where a blazed grating was located. A 2" diffraction grating was used to maximise the field of excitation. The diffraction grating was oriented perpendicular to the optical axis and the illumination angle was chosen such that the first diffraction order of the central wavelength propagated along the

optical axis. This orientation did not coincide with the blaze angle, and hence was not the most efficient[35] but allowed the temporal focusing plane of each of the holographic discs to coincide with the focal plane of the objective lens. A pair of telescopes comprised of L11, L12, L7 and the objective lens was used to de-magnify and relay the holographic spots on to the sample plane. In experiments where non-temporally focused spots were used (1030 nm excitation, Laser A), the diffraction grating was replaced by a mirror.

Excitation paths 1 and 2 were combined prior to the tube lens (L7) using a polarising beam splitter (PBS). The linear polarisation of the light exiting the objective was changed using a half-wave plate following the PBS. For each modality, the rotation of the half-wave plate was set to that which was found to maximise the 2P excited JEDI-2P-Kv fluorescence. SLM1 was also used to optimise the alignment of the GPC path by translating the position of the π phase disk relative to the centre of the incident Gaussian beam and adding tip/tilt/defocus phases to translate the position of the focus with respect to the PCF filter. SLMs 1, 2 and 3 were also used to correct for system aberrations by adjusting the coefficients of Zernike modes (evaluated at the centre of each SLM pixel) in order to maximise the efficiency, uniformity and contrast of 2P fluorescence excited in a thin rhodamine layer.

Setup 2 consisted of a low-NA Gaussian beam path and a second holographic path built around a commercial microscope (Zeiss Axio Examiner.Z1) and laser source E. A temporally focused low-NA, apertured, Gaussian beam of 12 μm FWHM laterally and 14 μm FWHM axially was generated similarly as described for Setup 1 (Path 1 for low-NA apertured beams) of Supplementary Fig. 1, where SLM1, L3 and L4 were omitted. The beam from laser E was truncated by a pinhole and directed to the diffraction grating G1 after being collimated by L1 and L2 (precise characteristics of optical elements are listed in Supplementary Table 2). SLM2 was used for spatial multiplexing, and the zero-order was physically blocked in this setup in a conjugate image plane of an extra relay telescope (not shown in Supplementary Fig. 1). Low-NA Gaussian beams generated in this way were used to acquire data shown in Fig. 5 and Supplementary Fig. 13. Temporally focused holographic (TF-CGH) spots were generated following the architecture shown in Path 2 of Supplementary Fig. 1, with the corresponding optical components shown in Supplementary Table 2. 12 μm TF-CGH spots of axial FWHM of ~10 μm were projected at the sample plane to perform experiments of Supplementary Fig. 12b–g.

Setup 3 was used for in vivo experiments presented in Fig. 6 and Supplementary Fig. 13, as well as in vitro experiments on organotypic slices of Supplementary Fig. 10f. Setup 3 was a standard TF-CGH setup, as described in this manuscript and in previous works[83] (See Supplementary Table 3 for the exact optical components), equipped with laser F. The setup was also equipped with a 2P-galvo-galvo scanning optical path, which we used at the beginning of each experiment (illumination at 920 nm) to identify the expression area of JEDI-2P-Kv and the target neurons. The complete description of the 2P-scanning path can be found in ref. 83.

In all cases, with the exception of Setup 2 where the average power was controlled directly by the integrated acousto-optic modulator (AOM) of laser E, the average laser power at the sample plane was controlled using a half-wave plate mounted on a motorised rotation mount (Thorlabs, PRM1Z8) in combination with a polarising beam splitter (PBS). Prior to each experiment, the efficiency of each path was measured. The power at the sample plane was recorded using a handheld power meter (Thorlabs, S121C) and the power of the s-polarised light exiting the PBS at each laser output was measured with a second power meter (Ophir, 30(150) A-BB-18, Nova II). The power of the s-polarised beam was monitored continuously during experiments and used to update the rotation of the half-wave plate to deliver the desired power at the sample plane during a given acquisition (calculated using the experimentally measured efficiency of each

path). All half-wave plates were externally triggered prior to each acquisition.

In all experiments, fluorescence was captured using a simple widefield detection axis comprised of a microscope objective, a tube lens (TL), and a scientific complementary metal-oxide semiconductor (sCMOS) camera (Photometrics Kinetix or Hamamatsu ORCAFlash 4.0, as summarised in Supplementary Tables 1-3). In the case of Setup 1, for both sCMOS detectors, the pixel size at the sample plane was 0.1625 μm. 1×1 binning was used for all experiments. The fluorescence was separated from the excitation light using a dichroic mirror. Widefield, single photon, epi-fluorescence excitation was accomplished by means of LED sources (470, 490 nm to excite JEDI-2P-Kv fluorescence and 430 nm to excite BFP fluorescence), filtered by bandpass excitation filters and focused onto the back focal plane of the objective lens by means of an achromatic lens. Long exposure times (1 s) and low excitation power densities were used to acquire all images based on single photon fluorescence. For 1P or dual-colour imaging (for instance during single-beam photostimulation and imaging experiments), fluorescence was filtered using either a quad-band filter (Chroma ZET405/488/561/640) or individual bandpass filters. Infrared light used for 2P excitation of fluorescence was blocked from the camera using a shortpass filter.

Data were acquired using a control scheme based on custom scripts written to control micro-manager 2.0 Gamma[100] from Python via Pycro-manager[101]. All experiments were controlled using desktop computers running Windows 10. During voltage imaging experiments, the micro-manager acquisition engine was bypassed. Data from the camera was streamed directly to disk on one of the acquisition computers. The first step of any experiment was to acquire 2 (JEDI-2P-Kv and transmitted light) or 3 (JEDI-2P-Kv, ChroME-ST (H2B-BFP) and transmitted light) widefield images of the sample. These images were used to select targeted cells during a given experiment. The centroids of all targeted cells were written to file for all experiments. All the voltage imaging data presented in this manuscript were acquired in "dynamic range", 16-bit mode, which meant that a maximum of 266 rows of pixels could be acquired at 1 kHz (e.g. 43 × 300 μm² FOV, Setup 1). In practice we used an exposure time of 1 ms resulting in an effective acquisition rate of 980 Hz following 0.02 ms readout (similarly for the recordings referred to as 500 and 750 Hz in the manuscript). For single cell experiments, data were only acquired for a square region with diameter less than 266 pixels centred on a given cell. For the multi-cell experiments, neurons were grouped to find the maximal number which could be imaged within the permitted FOV according to the acquisition rate. In "dynamic range" mode, the FOV is inversely proportional to the exposure time, such that we could acquire data from 532 rows (e.g. 86 × 300 μm² FOV, Setup 1) at 500 Hz. The FOV could be increased by a factor of 6 using "speed" mode, where data is read out at 8-bit. The camera was triggered using a 5 V, TTL signal generated by pCLAMP (Molecular Devices, Sunnyvale, CA) controlling an acquisition system (Molecular Devices, Axon Digidata 1550B). During experiments, widefield images were visualised using the open-source image viewer Napari and voltage imaging traces were visualised using pyqtgraph.

All fluorescence traces were analysed using the same analysis pipeline written in Python, as outlined in the main text and Supplementary Information, derived from ref. 69. When multiple cells were imaged simultaneously, the (known) centroid of the excitation spot in camera co-ordinates was used to crop an appropriately sized rectangular region of interest (ROI) surrounding each cell. In rare cases where the ROIs of independent cells overlapped, a region of each independent cell was identified manually. Individual cells were then defined according by regression of each pixel in the ROI against the average fluorescence trace of the manually segmented pixels (Supplementary Fig. 2).

## Simulation of temperature rise in tissue

Heat diffusion and temperature rise was modelled using the Green's function formalism for a homogeneous and isotropic (infinite) medium[76,77]. The heat was obtained by calculating the propagation of the field throughout the medium. In all cases, the absorption coefficient was taken as that for 1030 nm. Temperature rises are reported as those at the centre of the FOV. In the single target case, this also corresponds to the position of the target. For the multi-target case, targets were randomly distributed throughout the FOV. Although the inter-target distance was maximised for the simulations, in practice, due to the relatively long (30 s) illumination times, we found the results to be independent of the distribution of targets. For short illumination times (10 ms) the distribution of targets impacts the peak temperature rise.

For illumination times τ such that heat can diffuse over the distance $\ell = 150\ \mu m$ from the focal plane ($\tau \approx 30\ ms$) the brain can no longer be considered as an infinite medium and a boundary condition must be introduced. Here we use the so-called 'method of images' which is based on linear superposition[77,102] and consider the exposed surface of the brain as a perfect heat sink with no contact resistance. Although this is a simplified approximation, it is coherent with the observed significant drop in temperature at the surface of the brain[79]. It should be noted that ΔT calculated here is the increase to the initial (or steady-state) temperature distribution due to exposed brain surface.

Scattering is modelled using the beam propagation method which has been shown to be particularly effective for modelling propagation in biological media, in which scattering is mainly forward. This method is based on successive random phase masks whose statistical properties can be related to the macroscopic scattering properties (scattering length $\ell_s$ (166 μm) and anisotropy (0.9))[103,104]. When scattering is considered, the power at the objective needs to be increased by a factor of $\exp(\ell_s/\ell)$ in order to reach the required power density in the spot. This excess power forms distributed heat sources which also contribute to the local temperature rise.

## Preparation of CHO cells

CHO cells were acquired from Sigma (Sigma, 85050302) and cultured in T25 flasks (Falcon, 353107) in a medium consisting of DMEM-F12 + Glutamax (Fisher, Gibco™ 10565018), supplemented with 10% SBF (Fisher, Gibco™ 10500064) and 1% penicillin/streptomycin (5000 U ml⁻¹). Cells were passaged every 2–3 days, until P20 to avoid genetic drift. Prior to each experiment, cells were seeded on coverslips (Fisher, 10252961) in 24-well plates (25,000 cells/ml). After 24 hours, cells were transiently transfected with the plasmid pAAV_hSyn_JEDI-2P_GSS3_Kv2.1 using the Jet prime kit (Ozyme, POL101000015). The medium was then replaced after 4 hours. Experiments were performed 48–72 hours post transfection.

## Electrophysiology for scanless two-photon voltage imaging in CHO cells

48–72 hours post transfection, whole-cell voltage-clamp recordings of JEDI-2P-Kv-expressing CHO cells were performed at room temperature (21–23 °C). An upright microscope (Scientifica, SliceScope) was equipped with a far-red LED (Thorlabs, M660L4), oblique condenser, microscope objective (Nikon, CFI APO NIR, 40X, 0.8 NA), tube lens (Thorlabs, TTL200-A), and an sCMOS camera (Photometrics, Kinetix, or Hamamatsu, Flash4.0) to collect light transmitted through the sample. Patch-clamp recordings were performed using an amplifier (Molecular Devices, Multiclamp 700B), a digitizer (Molecular Devices, Digidata 1550B) at a sampling rate of 10 kHz and controlled using pCLAMP11 (Molecular Devices). Cells were continuously perfused with artificial cerebrospinal fluid (ACSF) comprised of 125 mM NaCl, 2.5 mM KCl, 1.5 mM CaCl₂, 1 mM MgCl₂, 26 mM NaHCO₃, 0.3 mM ascorbic acid, 25 mM D-glucose, 1.25 mM NaH₂PO₄. Continuous aeration of the

recording solution with 95% O₂ and 5% CO₂, resulted in a final pH of 7.4 (measured). Borosilicate pipettes (with filament, OD: 1.5 mm, ID: 0.86 mm, 10 cm length, fire polished, WPI) were pulled using a Sutter Instruments P1000 puller, to a tip resistance of 3.5–6 MΩ. Pipettes were filled with an intracellular solution consisting of 135 mM K-gluconate, 4 mM KCl, 4 mM Mg-ATP, 0.3 mM Na₂-GTP, 10 mM Na₂-phosphocreatine, and 10 mM HEPES (pH 7.35). All membrane potentials reported in this manuscript are Liquid Junction Potential (LJP) corrected by −15 mV (measured). Recordings were compensated for capacitance (Cm) and series resistance (Rs) to 70% (Cm = 11.6 ± 0.6 pF; Rs = 16.1 ± 1.1 MΩ). Only recordings with an access resistance below 35 MΩ were included in subsequent analysis.

All experiments were performed with laser A, except data presented in Supplementary Fig. 10 where laser D was also used.

For protocols 1 and 2 (Fig. 2), JEDI-2P-Kv-expressing CHO cells were patched and clamped at −55 mV and 3, 100 mV steps were applied under either continuous (3 s, power density: 0.88 mW μm⁻², corresponding to 100 mW per cell) or strobed illumination (200 ms every 2.5 s, power densities ranging from 0.66 to 1.55 mW μm⁻², corresponding to 75–175 mW per cell). The fluorescent responses to the depolarisation steps were simultaneously recorded at 100 Hz. For protocol 3, JEDI-2P-Kv-expressing CHO cells were patched and clamped at −75 mV to mimic the resting potential of neurons in the dentate gyrus of hippocampal organotypic slices. A 20 Hz train of 10, 3 ms, 100 mV steps was electrically induced, and the fluorescent response to different scanless illumination methods was recorded simultaneously (500 ms, power density: 1.33 mW μm⁻², corresponding to 150 mW per cell, 1 kHz acquisition).

To measure the F-V curve of the voltage indicator JEDI-2P-Kv under scanless 2P illumination at 940 nm (Supplementary Fig. 10b), JEDI-2P-Kv-expressing CHO cells were patched and clamped at −55 mV or −75 mV, and 200 ms voltage steps were applied (ranging from +35 mV to −95 mV) under strobed illumination (400 ms, centred on the voltage steps). The fluorescence response was recorded at 100 Hz.

The F-V curve of JEDI-2P-Kv was also measured under scanless 2P illumination at 1030 nm, at different peak intensities (0.03, 0.0625 and 0.145 GW mm⁻², corresponding to 20, 43 and 98 mW per spot). Cells were clamped at −75 mV and, 2 s voltage steps ranging from +35 mV to −95 mV were applied under strobed illumination (6 s, centred on the voltage steps). The fluorescence response was recorded at 5 Hz (Supplementary Fig. 10d, e).

## Preparation of hippocampal organotypic slice cultures for validating scanless two-photon voltage imaging of neuronal activity using JEDI-2P-Kv

All experimental procedures were conducted in accordance with guidelines from the European Union and institutional guidelines on the care and use of laboratory animals (Council Directive 2010/63/EU of the European Union). Hippocampal organotypic slices were prepared from mice (Janvier Labs, C57Bl6J) at postnatal day 8 (p8). Hippocampi were sliced with a tissue Chopper (McIlwain type 10180, Ted Pella) into 300 μm thick sections in a cold dissecting medium consisting of GBSS (Sigma, G9779) supplemented with 25 mM D-glucose, 10 mM HEPES, 1 mM Na-Pyruvate, 0.5 mM α-tocopherol, 20 nM ascorbic acid, and 0.4% penicillin/streptomycin (5000 U ml⁻¹).

After 30–45 min of incubation at 4 °C in the dissecting medium, slices were placed onto a porous membrane (Millipore, Millicell CM PICM03050) and cultured at 37 °C, 5% CO₂ in a medium consisting of 50% Opti-MEM (Fisher 15392402), 25% heat-inactivated horse serum (Fisher 10368902), 24% HBSS, and 1% penicillin/streptomycin (5000 U ml⁻¹). This medium was supplemented with 25 mM D-glucose, 1 mM Na-Pyruvate, 20 nM ascorbic acid, and 0.5 mM α-tocopherol. After three days in vitro (DIV), the medium was replaced with one containing 82% neurobasal-A (Fisher 11570426), 15% heat-inactivated horse serum (Fisher 11570426), 2% B27 supplement (Fisher, 11530536), 1% penicillin/

**Table 2 | List and final titres of viruses**

| Virus | Final titre (vg ml⁻¹) |
|---|---|
| AAV9.hSyn.JEDI-2P.Kv2.1 | $3.12 \times 10^{13}$ |
| AAV9.CaMKII.ChroME-ST.P2A.H2B.BFP | $4.46 \times 10^{12}$ |
| AAV1.EF1a.DIO.JEDI-2P.Kv2.1.WPRE | $2.36 \times 10^{13}$ |
| AAV9.hSyn.Cre.WPRE.hGH | $2.3 \times 10^{11}$ |

streptomycin (5000 U ml⁻¹), which was supplemented with 0.8 mM L-glutamine, 0.8 mM Na-Pyruvate, 10 nM ascorbic acid and 0.5 mM α-tocopherol. This medium was removed and replaced once every 2-3 days.

Slices were transduced with AAV9_hSyn_JEDI-2P_Kv2.1 at DIV 3 by bulk application of 1 µl of virus per slice (see Table 2). Experiments were performed between DIV 10 and 17.

### Electrophysiology for validating scanless two-photon voltage imaging of neuronal activity using JEDI-2P-Kv in hippocampal organotypic slices

At DIV 10-17, whole-cell patch-clamp recordings of JEDI-2P-Kv expressing granule cells in DG were performed at temperatures varying between 31-35 °C. During experiments, slices were perfused with ACSF as previously described. This extracellular solution was supplemented with 1 µM AP5 (Abcam, ab120003) and 1 µM NBQX (Abcam, ab120046) in all experiments except for the spontaneous activity recordings (Fig. 3e). Patch pipettes were filled with intracellular solution, as previously described.

Neurons were held at −75 mV in voltage-clamp. Recordings were compensated for capacitance (Cm) and series resistance (Rs) to 70% (Cm = 21 ± 6.3 pF; Rs = 19.2 ± 8.5 MΩ; mean ± s.d.). In current-clamp, neurons were injected with some current (less than 100 pA) if necessary to maintain their resting membrane potential to −75 mV. In the latter configuration, bridge potential was corrected (Bridge potential = 13.9 ± 4.2 MΩ; mean ± s.d.).

Neurons were first patched in whole-cell voltage-clamp configuration. Protocol 1 was then performed to confirm that the fluorescence of the patched cell was voltage responsive (Supplementary Fig. 6).

The ability to record single APs in neurons (Fig. 3b, c) was assessed by electrically triggering a 1 Hz train of 50 APs with short latency and jitter under strobed illumination (10 ms, power densities ranging from 0.66 to 1.55 mW µm⁻², corresponding to 75 to 175 mW per cell) while recording at three different acquisition rates (500 Hz, 750 Hz and 1 kHz). APs were triggered by injecting 700-900 pA currents for 2 ms.

The axial resolution of JEDI-2P-Kv (Supplementary Fig. 6) was measured by electrically triggering an AP and measuring the fluorescence response while displacing the objective in the z axis (from +50 to −50 µm, in 5 µm steps). The lateral resolution was measured (from +20 to −20 µm, in 2 µm steps) by mechanically moving the sample in the x-y axis.

Then, to assess the ability to record fast spike trains in neurons, trains of 10 APs from 25 to 125 Hz were electrically induced under illumination at different power densities (from 0.66 to 1.55 mW µm⁻², corresponding to 75 to 175 mW per cell) and recorded at different acquisition rates (500 Hz, 750 Hz and 1 kHz). The amount of current injected was one that was sufficient to evoke 10 APs at each of the different spike trains. Recordings with APs missing in the electrophysiological trace were dismissed. 125 Hz was found to be the limit at which the granule cells could spike in our conditions (Fig. 3d and Supplementary Fig. 7a–c).

To record spontaneous activity (Fig. 3e), JEDI-2P-Kv expressing neurons were patched and their membrane potential was monitored under continuous illumination for 30 s at a power density of 1.33 mW µm⁻² (150 mW per cell) while recording the fluorescent response at 1 kHz.

To investigate the possibility of recording subthreshold depolarisations (Fig. 4 and Supplementary Fig. 7d), 6, 20 ms steps separated by 30 ms and ranging from 0 to 2.5 mV (in 0.5 mV steps) were induced in JEDI-2P-Kv expressing neurons under strobed illumination (40 ms centred around the steps, power density: 1.11 mW µm⁻², corresponding to 125 mW per cell) while recording the fluorescence response at 1 kHz. This was repeated 50−75 times.

To investigate the potential effects of the laser illumination on neuronal health (Supplementary Figs. 8b−g, i and 12b−g), JEDI-2P-Kv neurons in DG were patched in whole-cell configuration and protocol 4 was performed at different powers. Protocol 4 consisted of three different recordings, to extract relevant membrane characteristics (Supplementary Figs. 8 and 12). For the first recording, a 5 Hz train of 5 APs was electrically triggered, while cells were illuminated for 30 s, to extract the half-width and amplitude of the APs. The second recording was performed in voltage-clamp configuration and consisted of a 10 mV, 100 ms voltage step, repeated 5 times, to estimate the capacitance. The third recording was performed in current-clamp configuration to measure the rheobase, the firing rate and the resting membrane potential of the patched cell. Here, increasing steps of current were injected for 1 second, starting from −20 pA in 20 pA steps. The amount of current necessary to evoke a spike was measured as the rheobase. The firing rate and the resting membrane potential were extracted from the fifth trace following the first spike (rheobase + 60 pA). Protocol 4 was also used at given powers for repeated voltage imaging recordings (Supplementary Figs. 8h).

The performances of JEDI-2P-Kv under 1030 nm illumination (Supplementary Fig. 10c) were investigated in hippocampal organotypic slices infected with a mixture of AAV1.EF1a.DIO.JEDI-2P.Kv2.1.WPRE and AAV9.hSyn.Cre.WPRE.hGH (refer to Table 2) at DIV 3, in order to get a sparser expression. Isolated expressing cells in the dentate gyrus were then patched in whole-cell current-clamp configuration and illuminated with a holographic spot (12 µm diameter, no temporal focusing) at 1030 nm (power density: 1.21 mW µm⁻², corresponding to 137 mW per cell) or 940 nm (power density: 1.33 mW µm⁻², corresponding to 150 mW per cell). Single APs were obtained as described previously.

To characterise the feasibility of using low-repetition rate, high energy pulsed lasers for performing multicell scanless 2P voltage imaging (Fig. 5), protocol 5 was performed. JEDI-2P-Kv expressing neurons were patched in whole-cell current-clamp configuration. Patched neurons were illuminated with a gaussian spot for 30 s, and APs were triggered by random injections of sufficient current to elicit an AP (200−550 pA). The fluorescence response was recorded using an acquisition rate of 500 Hz, at power densities ranging from 0.03 to 0.06 mW µm⁻² (corresponding to 7.5−12.5 mW per spot). This protocol was repeated while either 2 or 15 additional spots were randomly positioned in the surrounding, densely expressing, FOV.

### Viral vector injections and surgical procedures for in vivo experiments

Male and female C57BL/6 J mice (Janvier Labs) were used for experiments, which were performed in accordance with EU Directive 2010/63. Protocols were reviewed by the local Animal Experimentation Ethics Committee (CETEA n.44) and approved by the French Ministry of Research and Education (APAFIS#14267-201803261541580). Advice on procedures, refinement of animal experimentation standards, and pain and distress management were provided by the Local Animal Welfare Office. Animals were housed in groups of 2−5 per cage on a 12 h/12 h light/dark cycle with food and water ad libitum under controlled temperatures (19−22 °C) and humidity (45−55%).

High-titre adeno-associated viral vector was delivered via stereotaxic injections into cortical neurons of 7−8-week-old mice to express JEDI-2P-Kv in layers 1 and 2/3 of the barrel cortex. The surgical procedure was performed in sterile conditions on mice anaesthetised with

intraperitoneal injection of a mixture of ketamine (80 mg/Kg) and xylazine (8 mg/Kg). Body temperature was constantly monitored and kept constant at 37 °C. Before surgery, buprenorphine (0.05 mg/Kg) was injected subcutaneously and eyes were protected from light mechanically and with ophthalmic gel. Before piercing the skin, lidocaine 2% was injected subcutaneously. Then, a craniotomy of 0.5–0.7 mm was performed on the skull above the barrel cortex (1 mm caudal and 3 mm lateral from bregma). 1 μl of the viral solution (Table 2) was delivered at a depth of 200 ± 50 μm from the brain surface. Next, the skin was sutured, and the mouse recovered from anaesthesia on a heating pad. Buprenorphine was injected 12- and 36-h post-surgery.

After 3–10 weeks, cranial windows were performed under isoflurane anaesthesia (5% induction, 1–2% maintenance). 24- and 2-h before surgery, dexamethasone sodium phosphate (0.7 mg/Kg) was injected subcutaneously. Similar to viral injections, buprenorphine was injected subcutaneously, eyes were protected from light mechanically and with ophthalmic gel, and lidocaine was injected subcutaneously before piercing the skin. A cranial window of 3–3.5 mm was performed, centred on the viral injection site. Dura was removed while keeping the brain surface moist with sterile saline solution. A 3 mm cover glass (#0, 0.085 to 0.13 mm thickness; Multichannel System) was placed on top of the craniotomy and sealed with dental cement (Tetric EvoFlow A2, Ivoclar). A custom-made headplate was then fixed with dental cement. After that, mice were either maintained head-fixed and placed under the objective of the microscope (for acute preparations) or left to recover for at least one week before recording (chronic preparations). With the latter preparation, we were able to obtain clear windows for at least one month. No notable differences were observed between results from chronic or acute preparations.

At the beginning of each experimental session, the expressing region was identified via 2P-LSM imaging and experiments were performed.

## In vivo motion correction

Following each acquisition, we estimated whether or not any significant motion had occurred during the recording. Briefly, each dataset was convolved with a 3D Gaussian kernel (sigmas: 1(t), 5(x), 5(y)). We then calculated the centre of mass of each image (X, Y) in the filtered dataset. For these proof of principle experiments, any datasets where any changes in fluorescent intensity were correlated with a displacement in the centre of mass were rejected.

One of the benefits of scanless 2P imaging is that a 2D image is acquired at each timepoint. Hence it ought to be possible to use common approaches for detecting, measuring and ultimately, correcting for motion artefacts or the presence of neighbouring cells within each illumination spot[105].

## Preparation of hippocampal organotypic slices for two-photon actuation and imaging of neural activity using ChroME-ST and JEDI-2P-Kv

All animal procedures followed national and European animal care guidelines (Directive 2010/63/EU) and institutional guidelines on animals used for research purposes. Hippocampal organotypic slice preparations were prepared as described in ref. 106 with a few modifications, as follows. Hippocampi were extracted from P5-P8 C56Bl/6 J mouse pups sacrificed by decapitation. The dissection was carried out in filter sterilised (0.2 μm pore size) ice-cold medium containing: 248 mM sucrose, 26 mM NaHCO3, 10 mM glucose, 4 mM KCl, 5 mM MgCl2, 1 mM CaCl2, 2 mM kynurenic acid and 0.001% phenol red saturated with 95% O2 / 5% CO2. Transverse slices of 300–400 μm thickness were cut with a McIlwain Tissue Chopper using double-edge stainless steel razor blades. Undamaged slices were transferred individually onto the small pieces of PTFE membrane (Millipore FHLP04700) placed on membrane inserts (Millicell PICM0RG50) in the 6-well-plate containing 1 mL pre-warmed culture medium, using a plastic transfer pipette. The slices were cultured at 37 °C and 5% CO2 in antibiotic-free culture medium consisting of 80% MEM and 20% Heat-inactivated horse serum supplemented with 1 mM L-glutamine, 0.01 mg/ml Insulin, 14.5 mM NaCl, 2 mM MgSO4, 1.44 mM CaCl2, 0.00125% Ascorbic acid and 13 mM D-glucose. The culture medium was partially replaced with fresh, 37 °C warmed culture medium every 3 days.

Various titrations were tested to achieve sufficient levels of expression of both sensor and actuator. When slices were transduced with two viruses simultaneously, we observed a reduction in the expression of JEDI-2P-Kv. Furthermore, overexpression-mediated apoptosis was observed in some cases when slices were transduced with two viruses simultaneously. The best results were obtained by transducing slices with JEDI-2P-Kv first, followed a week later by ChroME-ST which resulted in strong co-expression of both proteins (Fig. 7b). However, in general we found that the expression levels of both proteins were more variable when the two constructs were co-expressed than when either construct was expressed independently.

Slices were transduced firstly with AAV9.hSyn.JEDI-2P.Kv2.1 at DIV 3 and secondly with AAV9.CaMKII.ChroME-ST.P2A.H2B.BFP (provided by H. Adesnik, University of California, Berkeley, USA) at DIV 10 by bulk application of 1 μl of virus per slice (Table 2). Channelrhodopsin-expressing cells were visualised using stable expression of an H2B–BFP fusion, which resulted in nuclear-localised BFP fluorescence. Experiments were performed between DIV 13 and 20.

To characterise the performances of ChroME-ST (Fig. 7 and Supplementary Fig. 16), ChroME-ST and JEDI-2P-Kv co-expressing granule cells in DG were patched in whole-cell current-clamp configuration. 5, 17.5 ms pulses of light at different spike rates (5–50 Hz) were applied at different power densities ranging from 0 to 0.09 mW μm$^{-2}$ (0–10 mW per cell) to photo-evoke APs. The fluorescent responses were recorded at an acquisition rate of 1 kHz. The AP probability was calculated as the percentage of APs evoked and recorded over five trials. Then, simultaneous optogenetic activation of ChroME-ST and voltage imaging of JEDI-2P-Kv in sets of 10 cells were performed simultaneously using 2P-TF-CGH and imaged at 500 Hz.

## Confocal imaging

Confocal imaging was performed on both organotypic hippocampal slices transduced with JEDI-2P-Kv and viral injected mouse brain slices. For the organotypic hippocampal slices, immersion in 4% paraformaldehyde (PFA) was conducted for 30 minutes, followed by three washes with phosphate-buffered saline (PBS). For the brain slices, the mouse brain was extracted immediately after in vivo imaging experiments, immersed in 4% PFA overnight, and subsequently washed with PBS. The fixed brain was then sectioned into 100-μm-thick coronal sections in cold PBS, using a Leica VT1100S vibratome. These fixed sections were mounted on glass slides using Fluoromount-g mounting medium and sealed with a coverslip using nail polish. Confocal images were acquired using an Olympus FV1000 confocal laser scanning microscope. Imaging was performed using both a 20X NA:0.85 oil immersion objective lens and a ×60 NA:1.4 objective lens, with an excitation wavelength of 488 nm.

## Definitions of precision, recall and F1-score

Simultaneous scanless 2P voltage imaging and whole-cell patch-clamp electrophysiology was performed in the dentate gyrus of hippocampal organotypic slices as described previously. APs were evoked by manual current injection at random timepoints during each 30 s recording. The APs in the electrophysiology traces were taken as the so-called "ground truth". The imaging data was upsampled using sinc interpolation to match the sampling rate of the electrophysiology data. We then measured the quality of spike identification in the scanless 2P voltage imaging datasets as a function of SNR threshold as follows[107]:

*True positives* (TP): number of APs at matching timepoints in the electrophysiology and optical traces

*False positives* (FP): number of APs identified in the optical trace not in the electrophysiology traces

*False negatives* (FN): number of APs identified in the ground truth traces not identified in the optical traces

*True negatives* (TN): 0

The precision, recall and F1 score were then calculated as:

Precision = TP / (TP + FP)

Recall = TP / (TP + FN)

F1 = (2 × Precision × Recall)/(Precision + Recall)

## Statistics and reproducibility

All experiments were repeated for at least two (and generally many more) independent passages of cells, transfections or infections. The Shapiro test (scipy.stats.shapiro) was used to test whether data were normally distributed. For normally distributed data, the paired or unpaired two-tailed students t-test was used to compare two independent samples. The non-parametric Mann-Whitney *U*-test (scipy.stats.mannwhitneyu) was used to compare two samples in the case when either or both samples were found not to be normally distributed. 'n' refers to the number of independent biological replicates, as stated in each figure caption and summarised in Supplementary Table 2. A statistical comparison was deemed significant if the p-value was less than 0.05. For all figures $*p < 0.05$, $**p < 0.01$ and $***p < 0.0001$. All results reported in the manuscript are communicated as the mean ± S.E.M. (standard error of the mean) of at least three technical replicates unless otherwise stated. As specified, error bars in plots denote either the standard deviation or the S.E.M. All biological replicates were included in each estimate. Estimation stats were performed using the Python package dabestr[87].

## Reporting summary

Further information on research design is available in the Nature Portfolio Reporting Summary linked to this article.

## Data availability

Due to the quantity of data acquired in total, the data that support the findings of this study are available from the corresponding author on reasonable request. The data generated in this study are provided in the Source Data files. Source data are provided with this paper.

## Code availability

A representative minimal working example of the data analysis pipeline used to analyse scanless two-photon voltage imaging data may be found at: https://zenodo.org/records/11507381. All other relevant code is available upon reasonable request.

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

## Acknowledgements

The project was supported by the 'Agence Nationale de la Recherche' through the projects SLALLOM (ANR-17-CE16-0021), HOLOPTOGEN (ANR-19-CE16-0026), LabEx LIFESENSES (ANR-10-LABX-65) and IHU FOReSIGHT (ANR-18-IAHU-01) (V.E., E.P. and R.S.), the AXA research foundation (V.E.) and the European Research Council ("HOLOVIS-AdG" ERC2019-ADG-885090 to V.E.), the Deutsche Forschungsgemeinschaft (DFG, German Research Foundation, Postdoc fellowship 442616457 to C.G.), the European Union's Horizon 2020 research and innovation programme under the Marie Skłodowska-Curie grant agreement #813457 and The Foundation for Medical Research FDT- End of thesis FDT202304016818 (C-Y.C.) and FDT202204015069 (I.B.), the Klingenstein-Simons Fellowship Award in Neuroscience (F.S.-P.); the McNair Medical Foundation (F.S.-P.); the John S. Dunn Foundation (F.S.-P.); Welch Foundation grants Q-2016-20190330 and Q-2016-20220331 (F.S.-P.); NIH grants R01EB027145 (F.S.-P.), U01NS113294 (F.S.-P.), U01NS118288 (F.S.-P.), and R01EB032854 (F.S.-P.), 1RF1NS128901 (F.S.-P.), U01NS133971 (F.S.-P.) and U01NS133657 (F.S.-P.); and NSF grants 1707359 (F.S.-P) and 1935265 (F.S.-P.). We thank Yoann Zaouter and Alexandre Thai from Amplitude Laser Group for the generous loan and installation of the Satsuma Niji laser, Christophe Tourain for the fabrication of custom electronic and mechanical components, Vincent de Sars for continued software development, Dimitrii Tanese for assistance on optical setup 2 and discussions, and Hillel Adesnik for providing the ChroME-ST virus. We thank the viral vector facility at the Vision Institute for producing AAVs, the Vision Institute animal house facility for the handling of laboratory animals and Stéphane Fouquet from the Vision Institute imaging facility for technical assistance with confocal microscopy. We thank Dr. B. Arenkiel, J. Ortiz-Guzman and Z. Chen at the TCH Neuro-connectivity Core for AAV packaging; this Core is supported by NIH grant P50HD103555 and the Charif Souki Fund.

## Author contributions

R.R.S. and E.P. designed the principal optical setup. R.R.S. built and characterised the principal optical setup, performed experiments,

developed the acquisition and analysis pipelines, and analysed data. I.B. prepared biological samples (CHO cells, organotypic slices, animal injections and surgeries), performed experiments and analysed data. I.B., A.S.M.L. and S.D. prepared organotypic slices. I.B., C.G., A.S.M.L. and S.D. performed electrophysiological recordings and analysed electrophysiology data for Supplementary Figs. 8 and 12. A.S.M.L acquired the confocal images. C.Y.C. characterised the microscopes used for the low-repetition rate experiments in Figs. 5 and 6 with contributions from C.G. for Supplementary Fig. 12. X.L. and F.S-P. provided the JEDI-2P-Kv construct, reviewed and proof-read the manuscript. B.C.F. performed heat propagation simulations. V.E. conceived of and V.E. and E.P. supervised the project. F.S-P., E.P. and V.E. acquired funding for the project. R.R.S., I.B., E.P. and V.E. wrote the manuscript, with contributions from all authors.

## Competing interests

The authors declare no competing interests.
