## [Peer Review File · Nature Communications]

REVIEWER COMMENTS

Reviewer #1 (Remarks to the Author):

Sims et al propose to use two-photon holographic excitation in combination with high-speed camera imaging, as a tool for voltage imaging in brain tissue. One-photon Voltage imaging using GEVIs has made tremendous progress in the past few years and proved useful for voltage recordings from sparsely labeled ensembles of superficial neurons. Two-photon approaches are now catching up, holding promise for voltage imaging from densely labeled neurons from deeper layers. In line with this effort, the authors used the recently published green GEVI Jedi-2P, carefully compared three common light-sculpting approaches in cultured cells, and provided some proof-of-principle validation in organotypic brain slices. Technically, it's overall an excellent paper. The authors provide a very thorough and careful characterization of their optical system, its capabilities, and its limitations. However, the actual demonstration of the utility of this system is quite limited.

Major comments:

1) One-photon holographic excitation with DMD or SLM has been recently demonstrated as a way to increase the SNR for voltage imaging from sparsely labeled cells in many publications from multiple groups (Adam Cohen, Eric Schreier, Xue Han, Karl Deisseroth). Of note, all these papers provided convincing data from awake-behaving animals and used relatively simple and cheap hardware. The expectation from the much more complex and expensive 2P approach is that it will provide a significant advantage, i.e. voltage imaging from denser and deeper cells. However, as the authors use conventional camera imaging, the emission from deeper cells mix, and deep imaging is not possible. All the data from organotypic slices is indeed from superficial cells, at the depth of ~50 μm , and it questions the advantage of the 2P approach. The actual depth used is mentioned only in the discussion and it should have been clearly stated in the introduction and results that imaging was done only in superficial cells.

2) The claim that 2P excitation will allow high SNR imaging from dense samples is also not fulfilled. The presented single cells data from dense slices is quite nice, but it required enormous laser intensities and is thus not applicable for imaging more than one cell at a time. As a solution, the authors turned to the use of a low rep-rate laser and multiplexed it over ~10 cells. This data (figure 6) shows poor SNR compared with the single-cell data in figures 3-5, suggesting that this approach is not so successful. This could be the result of mixed signals from adjacent cells. Alternatively, it could be the result of the low rep-rate laser which provides low SNR compared with the high rep-rate laser. While the authors carefully characterized the performance of the high rep-rate system they didn't do so for the low rep-rate laser. To reveal this point the authors will need to provide a proper comparison of both systems. Specifically, they should repeat the experiments described in figures 2-3 with the low rep-rate laser to find the optimal conditions to achieve high SNR from multiple cells with this laser. I hope that the authors could end up with a nice demonstration for high SNR recordings from many densely labeled cells imaged simultaneously, as with the current

data (figure 6) it seems that this approach is inferior to 1P holography or even to high-speed scanning 2P.

3) As said, using the high rep-rate laser the authors needed to use extremely high laser power which likely led to significant bleaching and heating. In accordance, the claim for 20 minutes of continuous recording without a decrease in SNR is not well supported by the data presented in Supp figure 15. If 20 minutes of continuous recording is possible, please show examples of clear spikes recorded at high SNR before and after 20 minutes of continuous illumination (with zoom-in on the spiking part of the trace). Please also quantify the spike SNR before and after.

4) I don't understand the rationale of the all-optical approach presented in figures 7-8. In this configuration, the imaging SNR and the number of excited spikes are correlated. Furthermore, it's impossible to image an ensemble and photoactivate specific cells. I thus don't see what scientific questions could be studied using this approach. A useful all-optical experiment should be based on an optically orthogonal indicator and actuator and this couldn't be achieved with the current setup. Please clarify this point, omit this data, or convincingly explain what could be the potential application for this tool.

Minor comments:

- Lines 150-152 – The difference in bleaching between the 3 approaches is marginal, it's not an important point for the main text and could be moved to the supplement.
- Line 301 – I don't think that sample stability over 20 minutes of imaging should be a concern when imaging a brain slice, this is a potential issue only for in vivo imaging.
- In the introduction, please mention and properly cite the many papers that used 1P holography for voltage imaging.

Reviewer #2 (Remarks to the Author):

Summary:

This manuscript describes scanless two-photon imaging of voltage signals in cultured CHO cells and in organotypic slices using JEDI-2P-kv. The authors further attempted to combine it with two-photon optogenetics in organotypic slices (however, with significant optical crosstalk. essentially the imaging and stimulation had so much crosstalk that they could use one laser). Overall, simultaneous optical imaging and stimulation is an exciting direction. However, the current manuscript has many issues (see below) and many claims that are not substantiated by the data.

Major comments:

1. Organotypic slices vs in vivo preparations:

The introduction motivates the work with the ability ‘for multitarget voltage imaging in 71 densely labelled scattering samples with cellular resolution, such as mammalian in-vivo preparations’ (Line 71, 72).

However, the entire paper contained data only in organotypic slices, no data in acute slices, let alone any data in ‘mammalian in-vivo preparations’. Ideally, the authors should provide data in ‘mammalian in-vivo preparations’. If this is not feasible, they should prominently explain what the problem is and delete the strong motivation of ‘mammalian in-vivo preparations’.

Meanwhile, the introductions are not precise, for example, in Line 71, 72, the author claimed that ‘sparse labelling strategies or sculpted illumination’ are ‘not suitable for multitarget voltage imaging in 71 densely labelled scattering samples with cellular resolution, such as 5.’ The cited paper successfully achieved voltage imaging in ‘mammalian in-vivo preparations’.

2. Illumination:

The light sculpting techniques are within the abilities of many neural imaging labs. However, the illumination power density is extremely high. For instance,

Line 130: ‘power density: 0.88 mW μm^{-2} , 100 mW per cell) for 3 seconds.’

Line 169: ‘75 – 175 mW per cell’

Line 197: ‘power 196 density: 1.33 mW μm^{-2} , corresponding to 150 mW per cell’

Line 1175: ‘(Power density: 1.33 mW μm^{-2} , 150 mW per cell, 1 kHz acquisition rate).’

Broad near IR illumination is reported to increase brain temperature (Podgorski 2016).

1) How does the heat capacity of the brain limit the number of neurons simultaneously imaged?

2) For how long is it possible to record a given number of cells? 3 seconds are rarely enough for imaging any physiological interesting dynamics in vivo.

a. Supplementary figure 15 claimed that SNR did not decay with 20 min 150 mW imaging, it is unclear if the cell is still alive at all. Ephys data as in figure 5 would be ideal.

b. In supplementary figure 15, it is also unclear how SNR was measured because the traces did not have any indication of single action potentials as shown in figure 5.

c. How many cells are these quantifications are measured over? At what temperature these measurements were made? In vivo preparation will likely have higher temperatures than organotypic slices. Therefore, it is recommended that the measurements should be done at in vivo temperature, eg, > 30C.

3) Does illumination with this extreme intensity increase the firing rate due to local heating?

4) Does illumination with this extreme intensity change the physiological properties of neurons?

a. Supplementary figure 16 provided immunostaining results to address this question. However, in this measurement, only one condition was tested '(10 ms strobed illumination, 50 cycles, 1 Hz, total illumination time 500 ms,'. 10 ms strobed illumination, 50 cycles, 1 Hz, and a total illumination time of 500 ms are rarely enough for imaging any physiological interesting dynamics. Please test properly using the claimed imaging condition in figure 2, figure 5, and supplementary figure 15.

b. In supplementary figure 16b, please provide the raw trace besides the extracted amplitude, half-width, and latency. Please also provide the membrane properties including the membrane resistance, resting potential, membrane capacitance, and rheobase under the illumination conditions.

3. Unphysiological dynamics in "spontaneous" traces:

Supplementary figure 14 provided 'spontaneous' traces. However, the traces and dynamics look *dramatically* different from the traces provided in figure 5 with ground-truth patch clamp recordings. There is rarely any single action potential in the traces; most are huge unphysiological dynamics, for instance, in a, c, j, k. The authors should provide patch clamp data to support the claim. Please also provide a zoomed view, for instance, in f, to show it is an action potential.

These unphysiological dynamics make it even more necessary to provide evidence that cells did not die under extreme illumination conditions.

4. Single trial trace and SNR:

Figure 2: please provide single-trial traces to give the audience a rough idea of the signal and noise level.

5. Photobleaching:

Figure 2b protocol 2 showed clear photobleaching. This is not so much clear in protocols 1 and 3. Please explain.

Figure 2 Protocol 3: please provide the whole trace when the laser is turned on so the audience can see baseline fluorescence.

6. Distortion of AP waveform:

Figures 3c and e clearly showed the distortion of the AP waveform even at 1 kHz recording speed. It is recommended that the authors should plot the ephys data and imaging trace together so that the audience can see the partial match and partial distortion. The author should include the kinetics of the JEDI under this imaging condition.

7. Detection of 1 mV is not supported by the data:

Line 1160: 'however after 25 trials 1159 depolarisations greater than or equal to 1 mV can be resolved'.

This claim is not supported by the data. In figure 4a, with either 25 or 50 trials of average, the blue vs the pink lines (1 mV), and the pink vs the orange lines (1 mV) are not separable; it is pretty similar to the baseline noise (SNR ~ 1).

Figure 4b should move before 4a.

8. Low rep rate laser power and high peak power:

1) The claim that 'using the low rep rate laser, the average illumination power would be less to resolve the action potential' is not substantiated. The authors should provide simultaneous imaging data and patch clamp data like that in figure 2. In supplementary 19, *45 mW* per cell power was used, as high as that using a high rep rate laser. This is inconsistent with the authors' claim.

2) In supplementary figure 17, it is important to show the baseline activity under the imaging condition with very high peak power.

3) While the average power using a low rep rate laser is lower (up to 10 mW), the peak power is *much* higher and this could potentially kill cells and could be a problem for long time illumination. The author should provide the quantification of how long it is possible to record a given number of cells, provide the quantification of SNR of well-defined voltage signals like that in figure 2, and provide the characterization of the physiological properties of neurons during imaging.

4) Meanwhile, the imaging condition should be explicitly provided in both the main text and figure legend. It seems that '10 ms strobed illumination, 50 cycles, 1 Hz, and total illumination time 500 ms' were used in figure 6. 500 ms are rarely enough for imaging any physiological interesting dynamics.' The authors should provide imaging and characterization under continuous illumination. If this is not feasible, they should prominently explain what the problem is.

9. Dynamics in figure 6 and supplementary figure 18 with low rep rate laser vs figure 5:

1) The dynamics in both figure 6 and supplementary figure 18 look *dramatically* different from the traces provided in figure 5 using the same preparation (organotypic slices) with ground-truth patch clamp recordings. It is unclear if real physiological dynamics are recorded, or if it is a motion or

heating artifact from the very high peak power. The author should provide simultaneous patch clamp data and imaging data using a low rep rate laser like that in figure 5.

2) Supplementary figure 18: many big fluctuations (for instance, e and f) are significantly higher amplitude than action potentials. This suggests that the recorded signal is more likely an artifact than real voltage depolarizations. The author should explain why this happens.

10. Power under low rep rate laser:

Although supplementary figure 19 provided patch clamp data, *45 mW* per cell power was used, as high as that using a high rep rate laser and 10X higher than the data in figure 6, supplementary figures 17 and 18. The author should explain why.

11. Optical crosstalk between stimulation and recording:

The authors demonstrated simultaneous optical stimulation and recording. However, there is significant crosstalk. The light used for imaging activates the opsin and even saturates the opsin, making it *only* possible to image the cellular activity during stimulation and only during low-frequency stimulation. In the meantime, the light used for stimulation activates the fluorescence, providing an almost equally large signal to the camera (Figure 8, d).

Therefore, it is unclear how useful the approach would be. The author should acknowledge the crosstalk and limitation upfront in both the introduction and main text.

For example, in Line 539-41, 'it would be possible to 539 record sensory-evoked activity patterns using voltage imaging, replay this activity using optogenetic 540 stimulation and, also tune the excitation parameters in order to explore the logic and syntax of neural 541 computation.'. How is it possible to record the activity without activating the opsins and perturbing the system?

12. Lateral resolution:

The authors emphasized the high lateral resolution (for instance, Line 436). However, this potentially would be a big problem for 'mammalian in-vivo preparations' with breathing and motion. The author should either provide data for in vivo recording or acknowledge the limitations.

Minor:

1. Line 1101: GCH should be CGH

2. Figure 1 light path:

If figure 1 is going to stand on its own, the complete light path (supplementary figure 1) should be included in the main figure 1.

3. Line 335 and 336: unclear what 'with long times' means.

4. control area in figure 6b:

Please provide the simultaneously recorded trace in the control area using the same time window. In the current C1 trace, both the start time point and the end time point are not aligned with the other cellular area.

Reviewer #3 (Remarks to the Author):

In this manuscript, Sims et al undergo an elegant and thorough proof of concept of using scanless temporally focused illumination strategies for voltage imaging. They develop a strategy to reliably read out the activity of multiple cells at kHz speeds. The authors should be commended for their technical achievements (the head-to-head comparison of three distinct strategies) and the clarity and skill of their writing. This work is timely, as voltage indicators are being improved regularly, while existing imaging approaches are still insufficient. However, I believe a few domains of the paper still need improvement. First, I am unconvinced that combining this imaging modality with an opsin is an effective strategy yet given the constraints. Second, some imaging questions remain that will help it be useful in vivo. I'll expand on these concerns below. All in all, I thought it was a good paper, and I look forward to its publication.

Concerns:

1. While I appreciate using imaging and 2p photostimulation together, these findings appear to prove that the techniques are incompatible at this time. It appears that powers needed to image cells would necessarily activate them if they were ChroME positive. Without being able to independently read out and stimulate cells, I'm not sure what the utility is of including ChroME.

To be convinced that the two could be used at the same time, I would need to see a figure performing whole cell physiology on chrome positive cells showing that effective imaging parameters (powers, illumination periods) do not, or minimally, depolarize cells. And that those same parameters are sufficient to resolve endogenous (or electrically induced) spiking. Furthermore, I'd need to be convinced that photostimulation artifacts (aka increased light intensity) was resolvable from normal imaging.

I don't think this is possible. But even in its absence I think this is an interesting paper. I'd recommend (although I believe this should be the authors prerogative) that the ChroME figures be moved to supplement and downplayed in the narrative.

2. All of the TF scanless illumination techniques have been developed for and/or used in vivo, and this imaging approach would be improved by relating it to in vivo use. While performing some proof-of-concept recordings in vivo would be ideal, it's possible they are out of scope of this manuscript.

Instead, TF is generally thought to be relatively immune to scatter. Does that hold true with imaging?

At least some discussion about how one would overcome motion with this approach.

3. While there is a description of a novel cell segmentation process, additional explanation and validations are necessary. Fig 6a Highlights a potential problem where the bottom right of cell 2 appears to actually be a different cell than the one circled. More validations that the cell segmentation process is accurate are needed to confirm when this is or is not an issue. Perhaps confirming with a somatic signal would make it clear.

Minor Points

4. The introduction should cover more of the random access imaging literature, not just the highspeed literature.
5. Line 134 Seems to reference data that is not shown. Should show data that demonstrates voltage responses were random and varied by spatial location and not simply read noise.
6. Fig 2c its very hard to parse the individual points, overlaid violin plots, or darker colors dots would be more legible. Errorbars on the mean could also improve readability.
7. Fig 2c would be easier to follow if it was broken into more panels (fig 2c could be fig 2c-g)
8. Supplementary note 2 while very thorough is very long and at points hard to follow. Can it be reorganized to highlight most important points? Maybe certain points can be moved to the main text, and keep the supplement crisp.
9. Several figures would benefit from having more quantification. (e.g.. 3e, S15, S16a etc).
10. The authors should ensure that the N is stated for each figure (including number of biological replicates for S16, S17), and that the unit is described (e.g. trial vs cell vs animal)

11. Cell Health does look debatable, is S16B a change in action potential threshold? AP threshold is a common cell health metric, along with input resistance, and resting membrane potential. Is this change just due to heating?

12. Being able to interleave holograms with ms timing (as suggested in line 356) would be a significant improvement, and maybe nontrivial to implement. If the authors are able to demonstrate this it would be an improvement, otherwise this line should be moved to the discussion.

13. S19 1030nm compatibility is really excellent, do you have data showing if higher laser intensities will create more SNR? What sets the upper limit of power used?

Rebuttal Table 1: Response to Reviewers' comments

Comment	Response
Reviewer #1	
Sims et al propose to use two-photon holographic excitation in combination with high-speed camera imaging, as a tool for voltage imaging in brain tissue. One-photon Voltage imaging using GEVIs has made tremendous progress in the past few years and proved useful for voltage recordings from sparsely labeled ensembles of superficial neurons. Two-photon approaches are now catching up, holding promise for voltage imaging from densely labeled neurons from deeper layers. In line with this effort, the authors used the recently published green GEVI Jedi-2P, carefully compared three common light-sculpting approaches in cultured cells, and provided some proof-of-principle validation in organotypic brain slices. Technically, it's overall an excellent paper. The authors provide a very thorough and careful characterization of their optical system, its capabilities, and its limitations. However, the actual demonstration of the utility of this system is quite limited.	We are glad the reviewer appreciated the thorough characterization of the optical system we performed. Taking note of the reviewers' doubts about the utility of our approach, we have incorporated a series of new characterization experiments using low repetition rate sources, and demonstrated in vivo voltage imaging. We hope this additional data addresses the referee's concerns.
Major Comments	
1) One-photon holographic excitation with DMD or SLM has been recently demonstrated as a way to increase the SNR for voltage imaging from sparsely labeled cells in many publications from multiple groups (Adam Cohen, Eric Schreiter, Xue Han, Karl Deisseroth). Of note, all these papers provided convincing data from awake-behaving animals and used relatively simple and cheap hardware. The expectation from the much more complex and expensive 2P approach is that it will provide a significant advantage, i.e. voltage imaging from denser and deeper cells. However, as the authors use conventional camera imaging, the emission from deeper cells mix, and deep imaging is not possible. All the data from organotypic slices is indeed from superficial cells, at the depth of ~50 um, and it questions the advantage of the 2P approach. The actual depth used is mentioned only in the discussion and it should have been clearly stated in the introduction and results that imaging was done only in superficial cells.	We agree with the reviewer that a number of groups have demonstrated in vivo single-photon voltage imaging from sparsely labelled cells, and that the expectation from a 2P approach is voltage imaging from denser and deeper cells. Indeed, in the case of single photon excitation, targeted illumination becomes less effective with increasing target density, resulting in crosstalk and reduced SNR (even at superficial depths) as nicely summarized in the recent work by Weber et al.³ We previously demonstrated that temporally-focused patterns are extremely robust to scattering and maintain their original shape following propagation through hundreds of micrometers of mouse brain⁴. In this work, we demonstrated high SNR in vitro scanless voltage imaging of much more highly densely-labelled preparations than previously used for 1P voltage imaging (4124 ± 964 cells/mm² vs 370 cells/mm²)⁵. We have made this comparison clearer in the revised manuscript.

	To demonstrate the major advantage of scanless 2P excitation which is to record from multiple cells simultaneously in depth, we have now included new in vivo experiments. Specifically, in vivo experiments were performed in mice expressing JEDI-2P under a hSynapsin promoter, resulting in dense fluorescence expression throughout layers 2/3 of the barrel cortex (783 ± 85 cells/mm², Figure 6b; i.e twice denser than in experiments performed using 1P excitation⁵). We were routinely able to perform scanless two-photon imaging at depths up to 250 μm (Figure 6 and Supplementary Figure 14). Moreover, it has been shown that the probability of crosstalk between points separated by 30 μm is less than 2% at depths up to 500 μm when using camera detection⁶ (Supplementary Figure 8d in reference⁶). Since the minimum separation between recorded neurons was 32 μm (average: 86 μm), we can conclude that our in vivo recordings were not limited by crosstalk due to camera detection. We have updated the manuscript so that the imaging depths of data acquired for both the in vitro and in vivo experiments have been clearly stated in the main text and Supplementary Table 4.
2) The claim that 2P excitation will allow high SNR imaging from dense samples is also not fulfilled. The presented single cells data from dense slices is quite nice, but it required enormous laser intensities and is thus not applicable for imaging more than one cell at a time. As a solution, the authors turned to the use of a low rep-rate laser and multiplexed it over ~10 cells. This data (figure 6) shows poor SNR compared with the single-cell data in figures 3-5, suggesting that this approach is not so successful. This could be the result of mixed signals from adjacent cells. Alternatively, it could be the result of the low rep-rate laser which provides low SNR compared with the high rep-rate laser. While the authors carefully characterized the performance of the high rep-rate system they didn't do so for the low rep-rate laser. To reveal this point the authors will need to provide a proper comparison of both systems. Specifically, they should repeat the experiments described in figures 2-3 with the low rep-rate laser to find the optimal conditions to achieve high SNR from multiple cells with this laser. I hope that the authors could end up with a nice demonstration for high SNR recordings from	To improve the characterization of data acquired using low repetition rate sources, we have replaced the low repetition-rate data shown in the previous manuscript with a series of new experiments performed using a fiber amplifier at 1030 nm, which provided high energy pulses with tunable low repetition rates (between 0.25 and 2 MHz). We first characterized the performance of JEDI-2P under 940 and 1030 nm (at 80 MHz) using simultaneous imaging and electrophysiology experiments in CHO cells and organotypic slices (Supplementary Figure 10) to assess the difference related to the use of different wavelengths. We did not observe a statistically significant difference in $\% \Delta F/F_0$ per action potential (46 ± 16 [940 nm], 37 ± 8 [1030 nm]), but on average the SNR of data acquired under 1030 nm illumination was 40% lower than that obtained under 940 nm (28 ± 16 [940 nm], 16 ± 3 [1030 nm]), due to the reduced absorption at this wavelength in agreement with previous studies⁷. Having established the feasibility of using JEDI-2P with 1030 nm excitation, we performed

many densely labeled cells imaged simultaneously, as with the current data (figure 6) it seems that this approach is inferior to 1P holography or even to high-speed scanning 2P.	experiments to identify the optimal imaging parameters using a fixed-wavelength high-pulse energy source (Amplitude Satsuma HP3). As outlined in the revised manuscript, we found that using a repetition rate of 500 kHz provided the best balance between SNR and number of targeted cells at a given average power, whilst minimizing any photoinduced physiological perturbations. Under 1030 nm excitation at a repetition rate of 500 kHz, we found that average powers of 7.5 mW (CGH)/ 12.5 mW (Gaussian) per neuron generated sufficient SNR (>5) to resolve all relevant aspects of neural activity in hippocampal organotypic slices, corresponding approximately to a 10x reduction in average power with respect to the high repetition rate laser. We have included high SNR recordings from a patched neuron (to provide the ground truth membrane potential), whilst 15 additional spots were randomly positioned in the surrounding, densely labelled field of view (Figure 5). In these experiments, we achieved a 96% (true-positive) detection rate of action potentials (as confirmed using the ground truth electrophysiology), with an F1-value of 0.946. This value is on the order of those referred to as “high-SNR” data in recent work published in Nature Methods⁸. We wish to clarify that the poor quality of the data in Figure 6 of the original manuscript was due to the expression levels and health of the organotypic slices during the short loan period of the NIJI OPA rather than the approach and laser themselves. To avoid any confusion, we have now removed this data from the updated version of the manuscript.
3) As said, using the high rep-rate laser the authors needed to use extremely high laser power which likely led to significant bleaching and heating. In accordance, the claim for 20 minutes of continuous recording without a decrease in SNR is not well supported by the data presented in Supp figure 15. If 20 minutes of continuous recording is possible, please show examples of clear spikes recorded at high SNR before and after 20 minutes of continuous illumination (with zoom-in on the spiking part of the trace). Please also quantify the spike SNR before and after.	We agree with the referee's assessment that the primary potential photodamage effects induced by 2P scanless illumination are thermal damage and photobleaching, given that the typically employed peak power levels are well below the threshold for nonlinear photodamage^{2,9}. To comprehensively investigate these effects, we have therefore monitored the threshold for these effects using both high repetition and low repetition laser sources. i) Light induced temperature rises with high-repetition rate excitation: We found that using high rep-laser, single cells could be imaged continuously for 30 s using 100 fs, 80 MHz sources, and average powers up to 125 mW per neuron, and that these conditions did not induce any other

observable changes in AP properties (Supplementary Figures 8a – f). We observed statistically significant changes in the membrane properties using powers greater than 150 mW. This threshold power is twice higher than what we found necessary for sufficient SNR recordings of all aspects of neural activity (Figures 2 – 4). By modelling heat diffusion and temperature rises using the Green's function formalism we found that the use of higher powers resulted in larger, faster temperature rises (> 8 K for 150 mW per target, Supplementary Figures 9a-c) and concluded that the physiological perturbations observed at average powers exceeding 150 mW can reasonably be attributed to light-induced heating.

ii) **Light induced temperature rises with low-repetition rate excitation:** We extended this investigation to include low-repetition rate lasers. In this case <15 mW per target was required with all modalities (7.5 mW per target using holography) to achieve an SNR > 5 and an action potential detection rate of over 96 % (percentage of true positives). We did not observe any physiological perturbations (other than a slight reduction in action potential half width) at these light doses.

iii) **Photobleaching:** We further characterized the photobleaching and photostability of JEDI-2P-Kv under parallel illumination at different powers and with different modalities (Supplementary Figures 4 – 6 and 10).

iv) **Long recordings:** We agree with the referee that the claims of 20 minutes continuous recordings (at 80 MHz) were not well supported by the data provided in the original submission. We have removed all references to 20 minutes of continuous recording from the manuscript, and instead have performed new experiments in order to characterize the capacity of long-term scanless two-photon voltage imaging using a high repetition rate laser. In these experiments, we patched neurons and evoked and recorded 5 action potentials at the beginning of each trial (protocol 4: 30 seconds continuous illumination; see Methods of the revised manuscript). We monitored the evolution of membrane potential properties as a function of number of repeats by observing the neuronal

	responses to injected current between trials (see Methods). We repeated this 10 times, corresponding to 5 minutes total illumination. We observed that, on average, illuminated cells depolarized sooner and with larger amplitudes than the control cells (10.7 ± 1.8 vs 5.7 ± 3.2 mV, mean \pm SEM, Supplementary Figure 8), whilst no significant differences were observed for the other properties for the control and illuminated cells. We have included representative electrophysiology traces and imaging data (zoomed in on the APs obtained at different powers) in Supplementary Figure 8 of the revised manuscript. We found that monitoring longer term effects of light exposure with electrophysiology is experimentally challenging because of the concomitant effect induced on the cell from light exposure and the patch electrode. In future studies we intend to monitor the effect of light illumination by imaging spontaneous or sensory evoked responses in vivo as done in reference¹⁰.
4) I don't understand the rationale of the all-optical approach presented in figures 7-8. In this configuration, the imaging SNR and the number of excited spikes are correlated. Furthermore, it's impossible to image an ensemble and photoactivate specific cells. I thus don't see what scientific questions could be studied using this approach. A useful all-optical experiment should be based on an optically orthogonal indicator and actuator and this couldn't be achieved with the current setup. Please clarify this point, omit this data, or convincingly explain what could be the potential application for this tool.	We agree with the reviewer regarding the necessity for an all-optical experiment to be based on optically orthogonal indicators and actuators. Unfortunately, achieving this requirement is currently challenging due to substantial cross-talk between the GEVIs and opsins commonly used for two-photon all-optical experiments. To address this limitation, the expansion of the optogenetic toolbox incorporating efficient red-shifted GEVIs and blue-shifted channelrhodopsins becomes necessary. However, the application suggested by the referee is just one among various possibilities benefitting from the combination of optogenetic stimulation and voltage imaging. Notably, the proposed single-beam excitation of cells co-expressing the spectrally overlapping opsin ChroME-ST and indicator JEDI-2P-Kv facilitates real-time optical monitoring of photo-evoked action potentials, providing a quick characterization of their properties such as latency and jitter. This is particularly valuable in scenarios where obtaining such data traditionally involves lengthy electrophysiological sessions or patching difficult accessible cells. Moreover, the single-beam configuration holds promise for use in connectivity mapping

	experiments, allowing the retrieval of occurrence and precise timing of evoked action potentials (APs) in pre-synaptic cells. As highlighted in a recent review¹¹ “the inability to reliably detect single action potentials through Ca²⁺ imaging has been a key challenge in all-optical studies. Achieving single-spike sensitivity with voltage indicators would significantly enhance the precision of optogenetic experiments, enabling detailed investigations into the impact of subtle differences in spike numbers, timing, and synchrony on downstream brain areas and animal behavior”. Although similar experiments have been conducted using GCaMP, this did not give the precision necessary to retrieve single spike timing, latency and jittering distribution which we demonstrate is possible by substituting GCaMP with a voltage indicator. We have simplified the presentation of these experiments, avoiding the use of the term 'all-optical investigation' to prevent creating misleading expectations. We have outlined the potential applications of single-beam photostimulation and imaging in the discussion section.
Minor Comments	
 • Lines 150-152 – The difference in bleaching between the 3 approaches is marginal, it’s not an important point for the main text and could be moved to the supplement. 	In response to reviewer comments, we have changed Figure 2 to improve clarity. We have moved the portion of the figure concerning photobleaching and photostability to the supplement of the revised manuscript (New Supplementary Figure 4).
 • Line 301 – I don’t think that sample stability over 20 minutes of imaging should be a concern when imaging a brain slice, this is a potential issue only for in vivo imaging. 	We have removed this sentence from the revised manuscript.
 • In the introduction, please mention and properly cite the many papers that used 1P holography for voltage imaging. 	To the best of our knowledge, we have now cited all of these papers in the revised manuscript. We have also added citations for articles that have either been published or uploaded to bioRxiv in the interim.
Reviewer #2	
Summary: This manuscript describes scanless two-photon imaging of voltage signals in cultured CHO cells	We thank the reviewer for the feedback and time dedicated to reviewing our work. In response to the comments made, we have substantially

and in organotypic slices using JEDI-2P-kv. The authors further attempted to combine it with two-photon optogenetics in organotypic slices (however, with significant optical crosstalk. essentially the imaging and stimulation had so much crosstalk that they could use one laser). Overall, simultaneous optical imaging and stimulation is an exciting direction. However, the current manuscript has many issues (see below) and many claims that are not substantiated by the data.	modified the manuscript and included additional experimental data to support all the claims made.
Major Comments:	
1. Organotypic slices vs in vivo preparations: The introduction motivates the work with the ability ‘for multitarget voltage imaging in densely labelled scattering samples with cellular resolution, such as mammalian in-vivo preparations’ (Line 71, 72). However, the entire paper contained data only in organotypic slices, no data in acute slices, let alone any data in ‘mammalian in-vivo preparations’. Ideally, the authors should provide data in ‘mammalian in-vivo preparations’. If this is not feasible, they should prominently explain what the problem is and delete the strong motivation of ‘mammalian in-vivo preparations’. Meanwhile, the introductions are not precise, for example, in Line 71, 72, the author claimed that ‘sparse labelling strategies or sculpted illumination’ are ‘not suitable for multitarget voltage imaging in 71 densely labelled scattering samples with cellular resolution, such as 5.’ The cited paper successfully achieved voltage imaging in ‘mammalian in-vivo preparations’.	In the revised manuscript we have included scanless two-photon imaging data acquired in vivo in head-fixed mice pan-neuronally expressing JEDI-2P-Kv in Layer 2/3 of the barrel cortex. We have modified the manuscript to reflect this change. We have substantially edited the introduction to make sure all statements and citations are more precise.
2. Illumination: The light sculpting techniques are within the abilities of many neural imaging labs. However, the illumination power density is extremely high. For instance:  • Line 130: ‘power density: 0.88 mW μm^{-2}, 100 mW per cell) for 3 seconds.’ • Line 169: ‘75 – 175 mW per cell’ • Line 197: ‘power 196 density: 1.33 mW μm^{-2}, corresponding to 150 mW per cell’ • Line 1175: ‘(Power density: 1.33 mW μm^{-2}, 150 mW per cell, 1 kHz acquisition rate).’ 	We would like to emphasize that due to the large lateral dimensions of the excitation spots used in this work, the power densities ($\text{mW}/\mu\text{m}^2$) employed, both with high and low repetition rate lasers, are significantly below the levels currently quoted in the literature for 2P voltage imaging (Rebuttal Table 2). Furthermore, the power densities used are lower than previously reported thresholds for non-linear damage^{2,9}. On the other hand, we agree that generating sufficient two-photon excitation in laterally extended areas requires high average powers. We performed additional experiments to monitor the effect of these powers on cell health using patch-clamp electrophysiology. Please refer to responses

Broad near IR illumination is reported to increase brain temperature (Podgorski 2016).	between 1 and 4 to reviewer #1 for a detailed description of these experiments and their results. We agree with the sentiment that in general efforts ought to be made to minimize the powers used to record neural activity in order to reduce any perturbations to the system under investigation. The aim of this work was to determine what powers are necessary and whether such powers perturb neural activity. The next steps of the project will be to identify approaches capable of further reducing the average power necessary in order to be able to image a larger number of cells simultaneously.
1) How does the heat capacity of the brain limit the number of neurons simultaneously imaged?	Light-induced heating is a linear function of average power delivered¹. In the case of the data acquired using a low repetition rate laser (500 kHz), we did not observe any light-induced perturbations below 30 mW per cell. According to simulations, this corresponds to a local temperature increase of <0.5K (Supplementary Figures 11a-b). These powers are approximately 3x higher than the power we found necessary to perform high SNR imaging of single neurons. We also estimated how the temperature rise scales as a function of the number of neurons imaged using simulations. We found that approximately 15 neurons in superficial layers could be imaged simultaneously (7.5 mW per target) whilst maintaining the temperature rises below 3K. The reduction of peak intensity due to scattering in vivo requires a corresponding increase in the average power delivered to the sample. As a result, the number of cells that could be imaged simultaneously in vivo reduces to 10 in layer 2/3 and to a single cell in layer 5 (Supplementary Figure 15). Please note that these numbers represent an initial benchmark for scanless two-photon voltage imaging, and clear avenues for increasing the number of neurons which can be simultaneously imaged exist - for instance, using brighter indicators excited at the peak of their excitation spectrum, or using more sophisticated light sculpting to maximize the photon density at the cell membrane to increase the efficiency of two-photon excitation.
2) For how long is it possible to record a given number of cells? 3 seconds are rarely enough for imaging any physiological interesting dynamics in vivo.	We have now added a new dataset using longer recording time. Specifically, we analyzed the possible damage induced using high and low-repetition rate lasers as a function of power

	following 30 s continuous illumination. We found a threshold power of 150 mW for the 80 MHz source tuned to 940 nm. We did not observe any statistically significant damage using low-repetition rate lasers below 30 mW. All in vivo recordings provided in the updated manuscript were performed for 30 s. Comparable recording times have been used to validate complementary 2P voltage imaging approaches (refer to Rebuttal Table 2). Furthermore, the use of low-duty cycle recordings is a commonly used strategy, proven to mitigate photo-induced damage¹² and one recommended by the Podgorski 2016 referred to by the reviewer. In general, the optimal recording length depends on the specific biological question the study intends to address. In each experiment it is necessary to find the optimal balance between number of targeted cells (total power delivered) and the length of the recordings. In all cases, we recommend that researchers perform control experiments to ensure that the imaging conditions are not perturbing the phenomenon under investigation. We performed these experiments and have emphasized this point in the revised manuscript.
a. Supplementary figure 15 claimed that SNR did not decay with 20 min 150 mW imaging, it is unclear if the cell is still alive at all. Ephys data as in figure 5 would be ideal. b. In supplementary figure 15, it is also unclear how SNR was measured because the traces did not have any indication of single action potentials as shown in figure 5. c. How many cells are these quantifications are measured over? At what temperature these measurements were made? In vivo preparation will likely have higher temperatures than organotypic slices. Therefore, it is recommended that the measurements should be done at in vivo temperature, eg, > 30C.	a. We agree that the original manuscript did not provide sufficient evidence to support this claim. We have removed this figure from the revised manuscript and performed a new set of experiments (Protocol 4, Methods, Supplementary Table 5). Please refer to response 3) of reviewer number 1. b. In the revised manuscript, we used Protocol 4 (Methods, Supplementary Table 5) in which five action potentials are evoked electrically during each recording. c. i. For all experimental data presented in the manuscript, we have summarized the number of trials, cells and biological replicates in Supplementary Table 4. We have also ensured that this information is summarized in all figure captions in each case. c. ii. For all setups equipped with a heater, experiments in organotypic slices were performed

	at temperatures similar to those in in vivo experiments (>30°C). The temperature at which each experiment was performed has been summarized in Supplementary Table 4.
3) Does illumination with this extreme intensity increase the firing rate due to local heating?	To address this question, we performed additional simultaneous imaging and electrophysiology experiments to measure the firing rate as a function of incident power for scanless two-photon voltage imaging. We performed these experiments with both high and low repetition rate sources, and both as a function of illumination power and number of repeats (total illumination time). Full details of the experimental protocol are provided in the Methods section of the updated manuscript (Protocol 4, Supplementary Table 5). We did not observe any statistically significant changes in firing rate at the powers we found necessary to perform scanless two-photon voltage imaging in combination with JEDI-2P-Kv. The results of these experiments are presented in Supplementary Figures 8 and 12.
4) Does illumination with this extreme intensity change the physiological properties of neurons? a. Supplementary figure 16 provided immunostaining results to address this question. However, in this measurement, only one condition was tested '(10 ms strobed illumination, 50 cycles, 1 Hz, total illumination time 500 ms, . 10 ms strobed illumination, 50 cycles, 1 Hz, and a total illumination time of 500 ms are rarely enough for imaging any physiological interesting dynamics. Please test properly using the claimed imaging condition in figure 2, figure 5, and supplementary figure 15. b. In supplementary figure 16b, please provide the raw trace besides the extracted amplitude, half-width, and latency. Please also provide the membrane properties including the membrane resistance, resting potential, membrane capacitance, and rheobase under the illumination conditions.	To address this question, we performed new experiments using protocol 4 (see Supplementary Table 5 and the Methods section of the revised manuscript for full details). These experiments were performed using 30 s continuous illumination, as used to acquire spontaneous activity recordings both in vitro and in vivo. We recorded the membrane properties recommended by the reviewer (amplitude, half-width, and latency), in addition to the membrane resistance, resting potential, membrane capacitance, and rheobase. We measured these properties both as a function illumination power and number of repeats (total illumination time) for both high and low repetition rate sources. The results obtained using the 80 MHz source are summarized in Supplementary Figure 8. As outlined in the updated manuscript, we did not detect any changes in any of the aforementioned membrane properties using powers up to 125 mW (Supplementary Figures 8b – g). However, use of powers greater than or equal to 150 mW induced statistically significant changes in capacitance, rheobase and action potential half width. Since the peak intensity used for these experiments is well below previously reported damage threshold^{2,9}, as

also mentioned in response 3) to Reviewer #1, we attribute these perturbations to light-induced heating. We found that it was possible to record many relevant aspects of neural activity using average powers less than or equal to 125 mW (Figures 3 –5).

We repeated this protocol, this time keeping the power constant (125 mW) and monitoring the membrane properties as a function of number of repeats (total illumination time). We observed that illuminated cells depolarized sooner (180 [illuminated] vs 420 s [control]) and with larger average amplitude (10.7 ± 1.8 mV [illuminated] vs 5.7 ± 3.2 mV [control]), but did not observe any other statistically significant differences in membrane properties between the illuminated and control cells (Supplementary Figure 8h).

We also repeated this protocol using a low repetition rate source at 1030 nm. We monitored the membrane properties as a function of excitation power. We did not observe any statistically significant differences in capacitance, firing rate, resting potential, rheobase or AP amplitude at any of the powers tested (Supplementary Figures 12b-g). The action potential half width was observed to decrease as a function of increasing power for the illuminated cells (-0.037 ± 0.012 mV, group 1 and -0.131 ± 0.015 mV, group 2). We observed differences in the AP half width between the illuminated and control cells (Supplementary Figure 12f). However, since, the magnitude of these changes was smaller than the spread of the values for the control cells (inset, Supplementary Figure 12f) and an order of magnitude smaller than what was observed in the case of the high-repetition rate laser (Supplementary Figure 8f), we conclude that scanless two-photon voltage imaging with low-repetition rate lasers induces much smaller (if any) physiological perturbations at the powers required to perform high SNR imaging of single neurons.

During the revision process, we repeated the immunohistochemistry protocol following 30s illumination at different powers with hippocampal organotypic slices imaged using the high and low repetition rate lasers, and additionally, fixed brain slices following in vivo experiments in order to

	provide more quantification of this data. However, we have decided to remove all of the immunohistochemistry data from the revised manuscript as the results were inconclusive, even in cases where we increased the laser power in order to induce visible damage. We believe that the extensive simultaneous imaging and electrophysiology characterization performed (both as a function of power and total illumination duration), in addition to the new heat diffusion simulation results provide a clearer summary of the damage threshold limits for scanless two-photon voltage imaging.
3. Unphysiological dynamics in “spontaneous” traces: Supplementary figure 14 provided ‘spontaneous’ traces. However, the traces and dynamics look *dramatically* different from the traces provided in figure 5 with ground-truth patch clamp recordings. There is rarely any single action potential in the traces; most are huge unphysiological dynamics, for instance, in a, c, j, k. The authors should provide patch clamp data to support the claim. Please also provide a zoomed view, for instance, in f, to show it is an action potential. These unphysiological dynamics make it even more necessary to provide evidence that cells did not die under extreme illumination conditions.	Organotypic slices are an extremely useful preparation to test and optimize new tools since a large number of slices can be obtained from a single animal, and these slices can be kept for several weeks. One problem however, is that these cultured slices do not mature physiologically, resulting in different activity patterns than typically observed for instance in acute brain slices. We have also seen these motifs in electrophysiological data during other experiments. However, in light of the reviewer comments, we have replaced these traces with those from patched neurons in in vitro preparations (Figure 5 and Supplementary Figure 13) and in vivo recordings which exhibit more familiar physiological dynamics (Figure 6 and Supplementary Figure 14). As described in the previous sections in this response, we performed additional experiments to determine at which conditions the incident light detectably perturbs physiology (measured using whole-cell patch clamp electrophysiology). Whilst we observed light-induced perturbations (as outlined above), we only observed cells dying in when using conditions beyond those necessary for recording high SNR traces using scanless two-photon voltage imaging (for instance > 15 mW using a 250 kHz repetition rate).
4. Single trial trace and SNR: Figure 2: please provide single-trial traces to give the audience a rough idea of the signal and noise level.	Figure 2 has been changed to include single-trial traces and single camera frames to provide readers with an idea of the signal and noise level at different recording speeds.
5. Photobleaching: Figure 2b protocol 2 showed clear	Raw traces from protocols 1 and 3 were detrended to remove the photobleaching, which otherwise

photobleaching. This is not so much clear in protocols 1 and 3. Please explain. Figure 2 Protocol 3: please provide the whole trace when the laser is turned on so the audience can see baseline fluorescence.	distorts the $\% \Delta F / F_0$ estimates. This was stated in the methods section of the original manuscript, and the procedure outlined in Supplementary Figure 2. We have amended the revised manuscript to include raw traces from protocols 1, 2 and 3, along with their detrended counterparts (Supplementary Figures 3-5 and 8).
6. Distortion of AP waveform: Figures 3c and e clearly showed the distortion of the AP waveform even at 1 kHz recording speed. It is recommended that the authors should plot the ephys data and imaging trace together so that the audience can see the partial match and partial distortion. The author should include the kinetics of the JEDI under this imaging condition.	Electrophysiological recordings remain the gold standard for precisely recording action potential waveforms. On the contrary, optical voltage imaging provides a low-pass filtered representation of the underlying membrane potential as thoroughly characterized in the original work⁷ on the performance of JEDI-2P-Kv. Precisely, the distortion of the AP waveform is a result of the recording speed, the indicator kinetics (0.54 ms for depolarization and 1.21 ms repolarization¹³) and sensitivity curve (Supplementary Figures 10b and d). Electrophysiology and imaging traces are plotted side-by-side throughout the manuscript. We also included the kinetics of JEDI-2P-Kv in the manuscript (Supplementary Figure 10a).
7. Detection of 1 mV is not supported by the data: Line 1160: 'however after 25 trials 1159 depolarisations greater than or equal to 1 mV can be resolved'. This claim is not supported by the data. In figure 4a, with either 25 or 50 trials of average, the blue vs the pink lines (1 mV), and the pink vs the orange lines (1 mV) are not separable; it is pretty similar to the baseline noise (SNR ~ 1). Figure 4b should move before 4a.	We have substantially revised this figure (Figure 4, revised manuscript) such that the individual traces can be seen more clearly (see also Supplementary Figure 7d). The difference between the 0 mV and $\Delta V > 1$ mV steps can be resolved by eye in this figure. We have revised the manuscript to clarify that whilst we demonstrated that it is possible to detect sub-threshold changes in membrane potential (> 1 mV) using scanless two-photon imaging and JEDI-2P-Kv, it would not be possible to accurately report the magnitude of the depolarization. This is a result of the sensitivity curve of JEDI-2P-Kv which is not an indicator optimized for imaging sub-threshold depolarizations.
8. Low rep rate laser power and high peak power: 1) The claim that 'using the low rep rate laser, the average illumination power would be less to resolve the action potential' is not substantiated. The authors should provide simultaneous imaging data and patch clamp data like that in figure 2. In supplementary 19, *45 mW* per cell power was used, as high as that using a high rep rate laser. This is inconsistent with the authors' claim.	1) There appears to have been a misunderstanding. The data the reviewer is referring to was obtained using the 80 MHz high-repetition rate laser tuned to 1030 nm which explains the use of comparable power levels to those used at 940 nm with an 80 MHz frequency. We aimed to use this first set of data only to demonstrate that scanless two-photon imaging could be used to record neural activity at 1030 nm with JEDI-2P-Kv. To better characterize the advantages and limitations of the use of low repetition rate lasers,

2) In supplementary figure 17, it is important to show the baseline activity under the imaging condition with very high peak power.

3) While the average power using a low rep rate laser is lower (up to 10 mW), the peak power is *much* higher and this could potentially kill cells and could be a problem for long time illumination. The author should provide the quantification of how long it is possible to record a given number of cells, provide the quantification of SNR of well-defined voltage signals like that in figure 2, and provide the characterization of the physiological properties of neurons during imaging.

4) Meanwhile, the imaging condition should be explicitly provided in both the main text and figure legend. It seems that '10 ms strobed illumination, 50 cycles, 1 Hz, and total illumination time 500 ms' were used in figure 6. 500 ms are rarely enough for imaging any physiological interesting dynamics.' The authors should provide imaging and characterization under continuous illumination. If this is not feasible, they should prominently explain what the problem is.

we added a new set of experimental data (Figure 5, Supplementary Figures 10 and 13) which summarize the feasibility of using JEDI-2P-Kv with 1030 nm excitation. This includes data acquired in vivo in Layer 2/3 of the barrel cortex in head-fixed mice.

For all experiments employing low repetition rate lasers we used substantially lower average power than for the 80 MHz source. Collectively we believe that this data substantiates the claim that low repetition rate lasers can be used to record action potentials with lower average powers (high peak energy) than for high repetition rate (low pulse energy) sources. In particular, using the low-repetition rate source tuned to 1030 nm at powers below 15 mW per cell, we found that we achieved an F1 score of 0.946 (96% true-positive detection rate of action potentials), which is on the order of the values achieved for "high-SNR" voltage imaging data in recent work published in Nature Methods⁸ (please also refer to the answer to comment No 2) of Reviewer #1).

2) We have removed supplementary figure 17 from the updated manuscript. We have provided simultaneous scanless two-photon voltage imaging traces for the low-repetition rate laser (Figure 5 and Supplementary Figure 13) which show the membrane potential under the highest peak-intensity conditions used for imaging in this study.

3) To address this question, we have performed a much more comprehensive characterization of scanless two-photon voltage imaging with the low repetition rate laser, as summarized with data presented in Figure 5 and Supplementary Figures 10 and 12 of the revised manuscript. We found that using a repetition rate of 500 kHz provided the best balance of SNR and number of target cells, whilst minimizing any photoinduced physiological perturbations. An average power of 7.5 mW per neuron generated sufficient SNR (>5) to resolve all relevant aspects of neural activity in hippocampal organotypic slices (Supplementary Figure 10). In general, we observed that the probability of non-linear damage was strongly dependent on the level of JEDI-2P-Kv expression in the target neurons, but that in each case the power required to achieve sufficient SNR to observe relevant aspects of neural activity was lower than that which resulted in non-linear damage (Supplementary Figure 10f vi).

	We found that the optimal strategy was to begin imaging using a low average power (around 2.5 mW per target, corrected for depth) and increase in 0.5 mW increments until sufficient SNR was achieved. As described at point 2.4), we have provided a thorough characterization of the physiological properties of neurons during imaging using the low-repetition rate, high pulse energy source (Supplementary Figure 12). 4) We have provided imaging and electrophysiology data under continuous illumination (Figure 5 and Supplementary Figure 12). Furthermore, all relevant experimental parameters (including repetition rate and average power) are specified in each of the figure captions, in the main text and have also been summarized in Supplementary Table 4.
9. Dynamics in figure 6 and supplementary figure 18 with low rep rate laser vs figure 5: 1) The dynamics in both figure 6 and supplementary figure 18 look *dramatically* different from the traces provided in figure 5 using the same preparation (organotypic slices) with ground-truth patch clamp recordings. It is unclear if real physiological dynamics are recorded, or if it is a motion or heating artifact from the very high peak power. The author should provide simultaneous patch clamp data and imaging data using a low rep rate laser like that in figure 5. 2) Supplementary figure 18: many big fluctuations (for instance, e and f) are significantly higher amplitude than action potentials. This suggests that the recorded signal is more likely an artifact than real voltage depolarizations. The author should explain why this happens.	1) We wish to reiterate the point made to Reviewer 1 that the poor quality of the data in Figure 6 of the original manuscript was due to the expression levels and health of the organotypic slices during the short loan period of the NIJI OPA rather than the approach and laser themselves. To avoid any confusion, we have removed all of the multi-cell data acquired using the NIJI OPA from the updated version of the manuscript. We have included single-cell data in the revised manuscript (Figure 5), both from organotypic slices (with ground truth electrophysiology data) and in vivo which exhibit clear action potentials. This data conclusively demonstrates that low repetition rate lasers can be used to record neural activity with scanless two-photon voltage imaging. We additionally acquired data whilst illuminating multiple spots simultaneously, in both organotypic slices and in vivo recordings from the layer 2/3 of the barrel cortex of head-fixed mice. 2) These are motifs of immature neurons we have seen in other experiments performed using hippocampal organotypic slices. This data has now been removed.
10. Power under low rep rate laser: Although supplementary figure 19 provided patch clamp data, *45 mW* per cell power was used, as high as that using a high rep rate laser and 10X higher than the data in figure 6, supplementary figures 17 and 18. The author should explain why.	This comment is related to the misunderstanding we outlined in response 8 to Reviewer 2. The data the reviewer is referring to was obtained using the 80 MHz high-repetition rate laser tuned to 1030 nm. All relevant experimental parameters have been reported in the figure captions, the main text and in Supplementary Table 4.
11. Optical crosstalk between stimulation and recording:	As stated in the original manuscript, simultaneous optical imaging and stimulation using spectrally

The authors demonstrated simultaneous optical stimulation and recording. However, there is significant crosstalk. The light used for imaging activates the opsin and even saturates the opsin, making it *only* possible to image the cellular activity during stimulation and only during low-frequency stimulation. In the meantime, the light used for stimulation activates the fluorescence, providing an almost equally large signal to the camera (Figure 8, d). Therefore, it is unclear how useful the approach would be. The author should acknowledge the crosstalk and limitation upfront in both the introduction and main text. For example, in Line 539-41, 'it would be possible to 539 record sensory-evoked activity patterns using voltage imaging, replay this activity using optogenetic 540 stimulation and, also tune the excitation parameters in order to explore the logic and syntax of neural 541 computation.'. How is it possible to record the activity without activating the opsins and perturbing the system?	orthogonal indicators and channelrhodopsins is an extremely exciting direction. Unfortunately, crosstalk free all-optical experiments are beyond the capabilities of the current optogenetic toolbox. We believe that single-beam optogenetic stimulation and voltage imaging is an extremely useful technique for assessing whether and, if so, precisely when action potentials were induced by photostimulation. The utility of a similar approach was demonstrated in recently published work⁹, and feedback from colleagues in the field suggests that this approach would be particularly useful for connectivity experiments. In response to comments from the reviewers and editors we have toned down the discussion in the main text as recommended by the reviewer.
12. Lateral resolution: The authors emphasized the high lateral resolution (for instance, Line 436). However, this potentially would be a big problem for 'mammalian in-vivo preparations' with breathing and motion. The author should either provide data for in vivo recording or acknowledge the limitations.	We would like to highlight the fact that the lateral resolution we referred to in the original submission is that of the detection axis (pixel size divided by magnification). To avoid any confusion, we have removed this sentence from the revised manuscript. However, we would like to highlight that scanless two-photon imaging is relatively robust to lateral motion artifacts because a 2D image is acquired at each timepoint and therefore conventional approaches for rigid motion correction of time series of 2D images can be applied (for instance, tools commonly used for 2P-LSM calcium imaging data). However, relatively large lateral motion can result in a loss of signal to noise ratio, due to portions of the membrane being displaced from beneath the excitation spot. To mitigate this problem, larger spot sizes (17 μm) were used for in vivo recordings to account for larger neuron sizes and to minimize the disruption of small motion.
Minor:	
1. Line 1101: GCH should be CGH	We thank the reviewer for noticing this typo which has now been corrected.
2. Figure 1 light path: If figure 1 is going to stand on its own, the complete light path (supplementary figure 1) should be included in the main figure 1.	We have changed Figure 1 in the updated manuscript. Part A is more detailed and includes new schematic diagrams to introduce the light sculpting methods used for scanless two-photon imaging. The "complete light path", has been

	adapted to become a supplementary figure (new Supplementary Figure 1).
3. Line 335 and 336: unclear what 'with long times' means	We meant "for long periods". We thank the reviewer for bringing this to our attention. This sentence has now been changed.
4. control area in figure 6b: Please provide the simultaneously recorded trace in the control area using the same time window. In the current C1 trace, both the start time point and the end time point are not aligned with the other cellular area.	These recordings have been removed from the revised manuscript. Instead refer to Figure 5 where multi-spot experiments were performed with simultaneous whole-cell patch clamp electrophysiology.
Reviewer #3	
In this manuscript, Sims et al undergo an elegant and thorough proof of concept of using scanless temporally focused illumination strategies for voltage imaging. They develop a strategy to reliably read out the activity of multiple cells at kHz speeds. The authors should be commended for their technical achievements (the head-to-head comparison of three distinct strategies) and the clarity and skill of their writing. This work is timely, as voltage indicators are being improved regularly, while existing imaging approaches are still insufficient. However, I believe a few domains of the paper still need improvement. First, I am unconvinced that combining this imaging modality with an opsin is an effective strategy yet given the constraints. Second, some imaging questions remain that will help it be useful in vivo. I'll expand on these concerns below. All in all, I thought it was a good paper, and I look forward to its publication.	We wish to thank the reviewer for their constructive feedback of our work and appreciation of the technical aspects of the project. We hope that the extensive changes we have made to the manuscript, including adding in vivo experiments, have sufficiently improved the manuscript to alleviate their concerns.
Concerns:	
1. While I appreciate using imaging and 2p photostimulation together, these findings appear to prove that the techniques are incompatible at this time. It appears that powers needed to image cells would necessarily activate them if they were ChromE positive. Without being able to independently read out and stimulate cells, I'm not sure what the utility is of including ChromE. To be convinced that the two could be used at the same time, I would need to see a figure performing whole cell physiology on chromE positive cells showing that effective imaging parameters (powers, illumination periods) do not, or minimally, depolarize cells. And that	We have simplified the presentation of these experiments, avoiding the use of the term 'all-optical investigation' to prevent creating misleading expectations. Please, refer to response to point 4) of referee #1. We would like to keep this reduced version in the main manuscript.

those same parameters are sufficient to resolve endogenous (or electrically induced) spiking. Furthermore, I'd need to be convinced that photostimulation artifacts (aka increased light intensity) was resolvable from normal imaging. I don't think this is possible. But even in its absence I think this is an interesting paper. I'd recommend (although I believe this should be the authors prerogative) that the ChroME figures be moved to supplement and downplayed in the narrative.	
2. All of the TF scanless illumination techniques have been developed for and/or used in vivo, and this imaging approach would be improved by relating it to in vivo use. While performing some proof-of-concept recordings in vivo would be ideal, it's possible they are out of scope of this manuscript. Instead, TF is generally thought to be relatively immune to scatter. Does that hold true with imaging? At least some discussion about how one would overcome motion with this approach.	In order to showcase the utility of scanless two-photon imaging, we have incorporated new experimental data into the manuscript. We acquired scanless two-photon voltage imaging recordings from multiple cells in Layer 2/3 of the barrel cortex (Figure 6, Supplementary Figure 14). As highlighted by the reviewer, temporally focused, sculpted, light has been demonstrated to exhibit remarkable robustness to scattering¹⁴, which means that extraneous excitation of neuropil fluorescence is avoided, and the light shaping and axial resolution of the imaging spot are well conserved even deep in scattering tissue. Regarding the evoked fluorescence, this is scattered as it propagates through the tissue independently by the fact of having used TF, but recent studies have demonstrated that the crosstalk between points separated by 30 μm is less than 2% even at depths approximately 500 μm below the cortical surface (Supplementary Figure 8d of reference⁶). In agreement with this prediction, we could perform in vivo imaging at depths up to 250 μm (with a minimum separation between neurons recorded simultaneously in-vivo of $\sim 30 \mu\text{m}$ (average 86 μm)). We have added some discussion about how scanless two-photon voltage imaging is relatively robust against motion artefacts to the main text.
3. While there is a description of a novel cell segmentation process, additional explanation and validations are necessary. Fig 6a Highlights a potential problem where the bottom right of cell 2 appears to actually be a different cell than the one circled. More validations that the cell segmentation process is accurate are needed to confirm when this is or is not an issue. Perhaps	The approach for cell segmentation was adapted from existing approaches developed for single photon voltage imaging, such as VolPy¹⁵ and TreFiDe¹⁶. We originally validated the approach with data from 41 CHO cells using different excitation modalities and then on data from neurons in organotypic slices with ground truth

confirming with a somatic signal would make it clear.	electrophysiology (Supplementary Figures 3-7). We found the algorithm to be very capable of assigning low weights to pixels which do not co-vary with the average somatic signal (defined as the average of that within a given radial distance from the centre of the target spot). We have added additional data to Figure 2c in the main text which highlights this point. Both the single and average frames contain (relatively) bright pixels of non-voltage sensitive fluorescence (most likely protein in the ER), which is much less prominent in the pixel weights. Consequently, the fluorescent trace calculated as the weighted average exhibits higher $\% \Delta F / F_0$ than the unweighted average (Supplementary Figure 2, revised manuscript). Similarly, we found that the algorithm was able to distinguish between signals from adjacent cells with sufficiently different activity patterns. We have updated the description of the analysis method and included a new schematic (Supplementary Figure 2) to clarify the method. We anticipate that this will become more of an issue upon imaging multiple cells simultaneously in deeper brain regions and are actively working on improvements to existing de-mixing approaches to handle these cases.
Minor points	
4. The introduction should cover more of the random-access imaging literature, not just the highspeed literature.	We have substantially changed the introduction in response to reviewer comments and included representative references from the field of random-access literature, as requested (Reddy & Saggau 2005, Salomé et al. 2006, Katona et al. 2012, Nadella et al. 2016).
5. Line 134 Seems to reference data that is not shown. Should show data that demonstrates voltage responses were random and varied by spatial location and not simply read noise.	We have removed these comments from the revised manuscript.
6. Fig 2c its very hard to parse the individual points, overlayed violin plots, or darker colours dots would be more legible. Error bars on the mean could also improve readability.	We have substantially revamped Figure 2 to improve readability and to emphasize the main points of the text. The individual points in Figures 2e and f are now darker, violin plots have been overlayed and the interquartile range is also displayed.
7. Fig 2c would be easier to follow if it was broken into more panels (fig 2c could be fig 2c-g)	We have broken Fig 2c of the original manuscript into two panels to emphasize the main findings of these experiments in the updated manuscript.

8. Supplementary note 2 while very thorough is very long and at points hard to follow. Can it be reorganized to highlight most important points? Maybe certain points can be moved to the main text, and keep the supplement crisp.	We have substantially re-organised the supplementary information. We have moved the most important points to the main text and removed Supplementary Note 2 from the revised manuscript. We intend to use this work in a follow-up paper.
9. Several figures would benefit from having more quantification. (e.g., 3e, S15, S16a etc).	Since we have acquired simultaneous electrophysiological and imaging data, we have removed Supplementary Fig. 15 from the revised manuscript. For more quantification of the perturbations induced by repeated light exposure please refer to Supplementary Figures 8 and 12 of the revised manuscript. During the revision process, we repeated the immunohistochemistry protocol following 30s illumination at different powers with hippocampal organotypic slices imaged using the high and low repetition rate lasers, and additionally, fixed brain slices following in-vivo experiments in order to provide more quantification of this data. However, we have decided to remove all of the immunohistochemistry data from the revised manuscript as the results were inconclusive, even in cases where we increased the laser power in order to induce visible damage. We believe that the extensive simultaneous imaging and electrophysiology characterization performed (both as a function of power and total illumination duration), in addition to the new heat diffusion simulation results provide a clearer summary of the damage threshold limits for scanless two-photon voltage imaging. In general, we have revised the manuscript to provide more quantitative information.
10. The authors should ensure that the N is stated for each figure (including number of biological replicates for S16, S17), and that the unit is described (e.g., trial vs cell vs animal)	We thank the reviewer for pointing this out. For all experimental data presented in the manuscript, we have summarized the number of trials, cells and biological replicates in Supplementary Table 4. We have also ensured that this information is summarized in all figure captions.
11. Cell Health does look debatable, is S16B a change in action potential threshold? AP threshold is a common cell health metric, along with input resistance, and resting membrane potential. Is this change just due to heating?	We performed additional simultaneous imaging and electrophysiology experiments to measure the change in action potential threshold as a function of incident power for scanless two-photon voltage imaging. We performed these experiments with

	both the high and low repetition rate lasers. Full details of the experimental protocol are provided in the Methods section of the updated manuscript (Protocol 4, Supplementary Table 5). The results of these experiments are presented in Supplementary Figures 8 and 12 and described in the results section of the revised manuscript. In summary, we observed light-induced physiological perturbations when using the 80 MHz source at powers greater than or equal to 150 mW. We confirmed that the likely source of these perturbations is heating using heat-propagation simulations (Supplementary Figures 9 and 11). We did not observe such perturbations (other than a slight reduction in action potential half width) when performing experiments using the low repetition rate 1030 nm sources. In the latter experiments <15 mW per target was required with all modalities and <7.5 mW per target using holography to achieve an SNR > 5 and an action potential detection rate of over 96 % (true positive rate).
12. Being able to interleave holograms with ms timing (as suggested in line 356) would be a significant improvement, and maybe non-trivial to implement. If the authors are able to demonstrate this it would be an improvement, otherwise this line should be moved to the discussion.	We agree that interleaving holograms with ms timing would be a significant improvement. This ought to be possible to implement², and this is an avenue we hope to explore in future. We have moved this line to the discussion of the revised manuscript.
13. S19 1030nm compatibility is really excellent, do you have data showing if higher laser intensities will create more SNR? What sets the upper limit of power used?	We have performed a thorough characterization of the compatibility of scanless two-photon voltage imaging of JEDI-2P at 1030 nm. Since these lasers are widely used for two-photon optogenetics experiments, this data greatly expands the scope of scanless two-photon imaging with respect to the original version of the manuscript. We have included new figures in the manuscript (Figure 5 and Supplementary Figures 10 and 13) which demonstrate that higher laser intensities result in higher SNR at different repetition rates and demonstrated that the upper limit of tolerable power is non-linear damage in the case of very low repetition rates. In general, however, we found that the onset of non-linear damage occurred at powers greater than those necessary to record relevant aspects of neural activity (Supplementary Figure 10 f vi).

Rebuttal Table 2: Comparative Table between different methods*2P excitation methods are highlighted in grey.*

Illumination method	Indicator	Max depth (μm)	Laser power	Laser Peak Intensity (GW/mm^2)	Longest imaging recording	Reference
1P-widefield	VSFP-butterfly	77	4-30 mW/mm^2 (@ sample)		7 seconds	17
1P-Light Sheet Microscopy	Voltron	148	<50 mW/mm^2		15 min	18
1P-DMD targeted	somArchon/ Optopatch	20-150 / 100-230	3 mW/cell		15 min	19
1P-DMD targeted	paQuasAr3	130	5–30 mW/cell		10 min	20
1P-DMD targeted	SomArchon	50-150	3–5 W/mm^2 (0.7–1.1 mW/cell)		5 min	21
1P-widefield	Ace2N-mNeon	150	20 mW/mm^2		20 seconds	22
1P-widefield	somArchon	50-150	1.6 - 4 W/mm^2		25 seconds	23
1P-widefield	pACE	200-250	<25 mW/mm^2		30 seconds	24
2P microscope in linescan mode	ANNINE-6plus	50	60 mW after the objective (45 mW if corrected for scattering)	7,1	4 min	25
2P raster scanning on a rectangular field-of-view around the cell	ASAP1, CAESR, ArcLight	-	4 mW	0,5	25 seconds	26
2P scanning	di-3-ANEPPDHQ	150	10–20 mW at laser focus	1,6	1 second	27
FACED (2P)	ASAP3	345	10-85 mW after the objective depending on depth (e.g. ~26 mW at 200 μm depth)	4,2	6 seconds	28
ULoVE (2P)	ASAP3	440	20 mW/cell corrected for scattering (120 mW at 440 μm)	2,2 (3 foci considered for ULoVE)	150 seconds	13
RAMP (AODs) (2P)	ASAP2s	130	5-30 mW	4,8		29
Spatiotemporal Multiplexing	SpikeyGi/2	300	30 $\text{mW}/\text{beamlet}$ (240 mW for 8 beamlets)	12,2	1 h	10

ULoVE (2P)	JEDI-2P	430	max 30 mW after the objective corrected for scattering (376 mW for $l_s=170 \mu\text{m}$ and $430 \mu\text{m}$ depth)	4,8 (3 foci considered for ULoVE)	40 min	7
2P-scanning	ASAP4	185	18-31 mW after the objective, 10.5 mW corrected for scattering	1,7	100 s, 10's of minutes	30
SLAP - Scanned line angular projection microscopy	yGluSnFR, jRGECO1a	250	96 mW in Vivo after the objective, around 50 mW at target in $110 \mu\text{m}$	846	>3,5 s	31
Spatial multiplexing (lens array+optical chopper)	Calbryte-590, GCaMP6f, R-CaMP2	300	0,3 -2,9 mW per beamlet @ 100 kHz - 1 MHz, 400 beamlets, e.g. 2,9 mW @ 200 kHz	66	~40 s	6
Parallel illumination with TF	JEDI-2P	240	7,5 mW corrected for scattering (31,8 mW after the objective for the max depth)	0,22	30 s	This work

Rebuttal Table 3: Major revisions to manuscript

As a result of the major revisions made to the original manuscript and supplementary information, highlighting all these changes obscured the text. We have instead summarized these changes in Rebuttal Table 3 (see below). In addition, we would like to highlight the following content which has been added to the manuscript in light of the new experiments and simulations performed:

- Section regarding in-vivo experiments (Results: Scanless two-photon voltage imaging in vivo)
- Methods (Simulation of temperature rise in tissue; Viral vector injections and surgical procedures for in vivo experiments; Definitions of Precision, Recall and F1-score)
- Figures 5 and 6 (Multi-target scanless two-photon voltage imaging using low repetition rate sources at 1030 nm, Scanless two-photon voltage imaging in vivo)
- Supplementary Tables 2, 3, 5 and 6
- Supplementary Figures 8, 9, 10, 11, 12, 13, 14, 15, 16, 17

Main text (original submission)	Description of changes
Abstract	We modified the abstract to incorporate the updated characterization of the low repetition rate source and the in-vivo data. We have also toned down the claims of the simultaneous voltage imaging and photostimulation data.
Introduction	We have modified the introduction to include more discussion on the random-access techniques as requested by reviewer #3. We have also updated the references to include papers published in the interim and single-photon techniques, as requested by reviewer #1.
Results: Scanless two-photon voltage imaging with sculpted, temporally focused excitation	Mostly unchanged. Minor changes have been made in response to reviewer comments.
Results: Scanless voltage imaging of neural activity in hippocampal organotypic slices with two-photon, temporally focused Generalised Phase Contrast	Mostly unchanged. We have amended the paragraphs containing the new characterisation results for continuous 30 s recordings, in addition to the heat diffusion simulations.
Results: Scanless two-photon voltage imaging of multiple targets with low repetition rate lasers	Major revisions. This section has become: Scanless two-photon voltage imaging of multiple targets with low repetition rate 1030 nm sources. This section contains new experimental and simulation data.
Results: Scanless two-photon voltage imaging and photostimulation of multiple targets with a single beam	Major revisions. As requested by the reviewers and editor, we have reformatted this section and toned down the claims made. This section has become: Simultaneous scanless two-photon voltage imaging and photostimulation.
Discussion	We have substantially modified the text of the discussion.
Methods: Experimental setup for performing two-photon voltage imaging with temporally focused, sculpted light	Mostly unchanged. Minor changes have been made in response to reviewer comments.
Methods: Preparation of CHO cells	Mostly unchanged. Minor changes have been made in response to reviewer comments.
Methods: Electrophysiology for scanless two-photon voltage imaging in CHO cells	Mostly unchanged. Minor changes have been made in response to reviewer comments.

Methods: Preparation of hippocampal organotypic slice cultures for validating scanless two-photon voltage imaging of neuronal activity using JEDI-2P-kv	Mostly unchanged. Minor changes have been made in response to reviewer comments.
Methods: Electrophysiology for validating scanless two-photon voltage imaging of neuronal activity using JEDI-2P-kv in hippocampal organotypic slices	Mostly unchanged. Minor changes have been made in response to reviewer comments.
Methods: Preparation of hippocampal organotypic slices for two-photon actuation and imaging of neural activity using ChromE-ST and JEDI-2P-kv	Mostly unchanged. Minor changes have been made in response to reviewer comments.
Methods: Immunostaining	Major revisions. This section has been removed from the revised manuscript.
Methods: Statistics	Mostly unchanged. Minor changes have been made in response to reviewer comments.
References	We added the 1P and random-access scanning references requested by reviewers 1 and 3. We have also added references for relevant works published during the interim.
Figures	
1	Major revisions. We updated the schematic overview of scanless two-photon voltage imaging (part (a)) in response to the comments of reviewer #2.
2	Minor revisions. We have added a schematic to illustrate the experimental protocol and re-arranged the figure. All traces have been plotted as $-\% \Delta F / F_0$ rather than as $\% \Delta F / F_0$ (as in the original manuscript) in order to be consistent throughout the entire results section. We have plotted the traces of individual cells in grey as requested by reviewer #2. We have added representative data as requested by reviewer #2. We have changed the format of the plots in (d), (part (c) in the original submission) to add violin plots as requested by reviewer #3. We have also moved the photostability and photobleaching to the supplementary information as requested by reviewer #1.
3	Minor revisions. We have added a schematic to illustrate the experimental protocol and re-arranged the figure. We have added new confocal data and zoomed in regions of single cells. We have changed the colour of the dashed blue line in part (b) of the original submission to magenta for clarity. We merged figure 5 of the original manuscript with this figure.
4	Minor revisions. We have re-arranged this figure and zoomed into the “50 trial” data for clarity in response to comments by reviewer #2.

5	Major revisions. Two of the traces shown in this figure have been moved to Figure 3 of the updated manuscript.
6	Major revisions. Data from this figure has been removed from the revised manuscript.
7	Major revisions. Data from figures 7 and 8 of the original manuscript have been combined to create a single figure (Figure 7 in the revised manuscript).
8	Major revisions. See above.
Supplementary Information	
Supplementary Note 1: Experimental setup used for widefield 2P-voltage imaging	This is now Supplementary Figure 1.
Supplementary Note 2: Optimal excitation and detection for widefield voltage imaging	Major revisions. This has been removed from the updated manuscript in response to comments from reviewer #3.
Supplementary Note 3: Extracting fluorescence time series from widefield voltage imaging data	Major revisions. This is now Supplementary Method 1. Additional details have been added in response to comments from reviewer #3.
Supplementary Methods	Major revisions. We have removed the “simulations” section from the revised manuscript as these results have now been removed.
Supplementary Figures	
1	Major revisions. We have updated the schematic diagram of the optical setup in light of the new experimental data added to the manuscript.
2	Major revisions. We have removed this figure from the revised manuscript.
3	Major revisions. We have removed this figure from the revised manuscript.
4	Major revisions. We have removed this figure from the revised manuscript.
5	Major revisions. We have removed this figure from the revised manuscript.
6	Major revisions. We have updated this figure in response to comments from reviewer #3 and added additional information about how data from multi-target experiments was analysed. This is Supplementary Figure 2 of the updated manuscript.
7	No major changes. This is now part of Supplementary Figure 2 of the updated manuscript.
8	No major changes. This is now part of Supplementary Figure 2 of the updated manuscript.
9	Major revisions. We have added the raw and processed traces in response to comments from

	reviewer #2. This is now part of Supplementary Figure 4 of the updated manuscript.
10	Major revisions. We have added the raw and processed traces in response to comments from reviewer #2. This is now part of Supplementary Figure 5 of the updated manuscript.
11	Major revisions. We have removed this figure from the updated manuscript. Part (c) is now part b(ii) of Supplementary Figure 10 of the revised manuscript.
12	No major changes. This is now part of Supplementary Figure 6 of the updated manuscript.
13	Major revisions. We have added data from an additional power density ($1.11 \text{ mW}\mu\text{m}^{-2}$) to this figure and re-arranged the panels. This is now part of Supplementary Figure 7 of the updated manuscript. We have added some sub-threshold data to part (d) of the updated figure.
14	Major revisions. We have removed this figure from the updated manuscript.
15	Major revisions. We have removed this figure from the updated manuscript.
16	Major revisions. We have removed this figure from the updated manuscript. The new data acquired under continuous 30 s illumination under a greater number of conditions, and monitoring a larger number of membrane parameters is Supplementary Figure 8 of the revised manuscript.
17	Major revisions. We have removed this figure from the updated manuscript. The new data acquired under continuous 30 s illumination under a greater number of conditions, and monitoring a larger number of membrane parameters is Supplementary Figure 12 of the revised manuscript.
18	Major revisions. We have removed this figure from the updated manuscript.
19	Major revisions. We have removed this figure from the updated manuscript. The characterization of scanless two-photon voltage imaging with 1030 nm, amplified laser sources forms part of Supplementary Figure 10 of the revised manuscript.
20	Major revisions. We have removed this figure from the updated manuscript.

1. Picot, A. *et al.* Temperature Rise under Two-Photon Optogenetic Brain Stimulation. *Cell Rep* **24**, 1243–1253 (2018).
2. Faini, G. *et al.* Ultrafast light targeting for high-throughput precise control of neuronal networks. *Nat Commun* **14**, 1888 (2023).

3. Weber, T. D., Moya, M. V., Kılıç, K., Mertz, J. & Economo, M. N. High-speed multiplane confocal microscopy for voltage imaging in densely labeled neuronal populations. *Nat Neurosci* **26**, 1642–1650 (2023).
4. Papagiakoumou, E. *et al.* Functional patterned multiphoton excitation deep inside scattering tissue. *Nat Photonics* **7**, 274–278 (2013).
5. Kannan, M. *et al.* Dual-polarity voltage imaging of the concurrent dynamics of multiple neuron types. *Science (1979)* **378**, (2022).
6. Zhang, T. *et al.* Kilohertz two-photon brain imaging in awake mice. *Nat Methods* **16**, 1119–1122 (2019).
7. Liu, Z. *et al.* Sustained deep-tissue voltage recording using a fast indicator evolved for two-photon microscopy. *Cell* **185**, 3408–3425.e29 (2022).
8. Cai, C. *et al.* FIOLA: an accelerated pipeline for fluorescence imaging online analysis. *Nat Methods* **20**, 1417–1425 (2023).
9. Mardinly, A. R. *et al.* Precise multimodal optical control of neural ensemble activity. *Nat Neurosci* **21**, 881–893 (2018).
10. Platasa, J. *et al.* High-speed low-light in vivo two-photon voltage imaging of large neuronal populations. *Nat Methods* (2023) doi:10.1038/s41592-023-01820-3.
11. Kim, T. H. & Schnitzer, M. J. Fluorescence imaging of large-scale neural ensemble dynamics. *Cell* **185**, 9–41 (2022).
12. Podgorski, K. & Ranganathan, G. Brain heating induced by near infrared lasers during multi-photon microscopy. *J Neurophysiol* **116**, 1012–1023 (2016).
13. Villette, V. *et al.* Ultrafast Two-Photon Imaging of a High-Gain Voltage Indicator in Awake Behaving Mice. *Cell* **179**, 1590–1608.e23 (2019).
14. Papagiakoumou, E. *et al.* Functional patterned multiphoton excitation deep inside scattering tissue. *Nat Photonics* **7**, 274–278 (2013).
15. Cai, C. *et al.* VolPy: Automated and scalable analysis pipelines for voltage imaging datasets. *PLoS Comput Biol* **17**, 1–27 (2021).
16. Buchanan, E. K. *et al.* Penalized matrix decomposition for denoising, compression, and improved demixing of functional imaging data. 1–36 (2019).
17. Quicke, P. *et al.* Single-Neuron Level One-Photon Voltage Imaging With Sparsely Targeted Genetically Encoded Voltage Indicators. *Front Cell Neurosci* **13**, 1–12 (2019).
18. Abdelfattah, A. S. *et al.* Bright and photostable chemigenetic indicators for extended in vivo voltage imaging. *Science (1979)* **365**, 699–704 (2019).
19. Fan, L. Z. *et al.* All-Optical Electrophysiology Reveals the Role of Lateral Inhibition in Sensory Processing in Cortical Layer 1. *Cell* **180**, 521–535.e18 (2020).
20. Adam, Y. *et al.* Voltage imaging and optogenetics reveal behaviour-dependent changes in hippocampal dynamics. *Nature* **569**, 413–417 (2019).
21. Xiao, S. *et al.* Large-scale voltage imaging in behaving mice using targeted illumination. *iScience* **24**, (2021).
22. Gong, Y. *et al.* High-speed recording of neural spikes in awake mice and flies with a fluorescent voltage sensor. *Science (1979)* **350**, 1361–1366 (2015).
23. Piatkevich, K. D. *et al.* Population imaging of neural activity in awake behaving mice. *Nature* **574**, 413–417 (2019).
24. Kannan, M. *et al.* Fast, in vivo voltage imaging using a red fluorescent indicator. *Nat Methods* (2018) doi:10.1038/s41592-018-0188-7.
25. Roome, C. J. & Kuhn, B. Simultaneous dendritic voltage and calcium imaging and somatic recording from Purkinje neurons in awake mice. *Nat Commun* **9**, 1–14 (2018).
26. Brinks, D., Klein, A. J. & Cohen, A. E. Two-Photon Lifetime Imaging of Voltage Indicating Proteins as a Probe of Absolute Membrane Voltage. *Biophys J* **109**, 914–921 (2015).
27. Fisher, J. a N. *et al.* Two-photon excitation of potentiometric probes enables optical recording of action potentials from mammalian nerve terminals in situ. *J Neurophysiol* **99**, 1545–1553 (2008).
28. Wu, J. *et al.* Kilohertz two-photon fluorescence microscopy imaging of neural activity in vivo. *Nat Methods* **17**, 287–290 (2020).

29. Chamberland, S. *et al.* Fast two-photon imaging of subcellular voltage dynamics in neuronal tissue with genetically encoded indicators. *Elife* **6**, 1–35 (2017).
30. Evans, S. W. *et al.* A positively tuned voltage indicator for extended electrical recordings in the brain. *Nat Methods* **20**, 1104–1113 (2023).
31. Kazemipour, A. *et al.* Kiloherz frame-rate two-photon tomography. *Nat Methods* **16**, 778–786 (2019).

REVIEWERS' COMMENTS

Reviewer #1 (Remarks to the Author):

The authors adequately responded to all comments and thoroughly revised the paper. I have no further requests and I support the publication of the current version.

Reviewer #3 (Remarks to the Author):

This is a re-review of Simms et al.

I generally felt positively about the original manuscript, and my overall opinion remains positive.

I am still skeptical of the utility of using opsin and voltage imaging in this confounded way. I believe the use case of examining the timing of an optogenetically evoked voltage sensor signal, is worthwhile, but is an incredibly niche application.

More consideration needs to be placed on how motion will impact this approach:

- The only discussion I can find about motion artifacts is a line that says a 17 μ m spot minimizes artifacts. This argument should be matched with data. It seems that artifacts will occur anytime a portion of a fluorescent cell reaches the edge of the holographic disk. Therefore, in the absence of active motion correction, the critical question isn't the size of the disk but the distance from the edge of the disk to the nearest cell membrane. Brain motion can be variable, but 3-5 μ m would not be unreasonable. This however poses a tradeoff with cell density, if a spot is too big, you'll get a second source of contamination where additional cells entering your holographic spot. These caveats and how/if they're overcome are important for a practical user of the technology.

The authors liken their low NA Gaussian approach to 3D-SHOT, but this seems imprecise. Rereading the 3D-SHOT papers show substantial differences. For example, there isn't a beam expander before the diffraction grating in any of the published 3D-SHOT approaches that I'm aware of. Furthermore, given the diffuser in 3D-SHOT its debatable whether it is best described as a low NA gaussian beam. Perhaps simple text changes that make it clear that this is not precisely 3D-SHOT would allow readers to appreciate the differences.

Additional reviewer comments:

I've finished rereading the rebuttal and looking into the Author's response to each of Reviewer 2's concerns. Reviewer 2 clearly had many detailed concerns and how well the authors addressed them would be best answered by the original reviewer. That said, I agree with many of Reviewer 2's original concerns, I don't believe that the authors have neglected any of the reviewer's questions but could see the original reviewer not being completely satisfied by the findings. My net assessment is that the final product is technically viable, but this paper doesn't prove to me that I want to adopt it as an approach for biological experiments. In my opinion, that's ok - and it still deserves to be published. I have tried similar experiments in preliminary attempts (although its not part of my current research strategy) so to see what worked and how well is important step in the scientific process and adds value.

Some specific points:

Main Point 1.

The first concern was that this approach is supposed to be used in vivo, but they hadn't shown it. The Authors added a figure devoted to in vivo recordings and this made their paper much stronger. While this figure could be better (showing longevity of recordings, more comprehensive study of the number of cells recorded at a time, etc), it at the very least shows in vivo recordings are achievable. If there was one thing to change it would be to expand upon these in vivo experiments.

Additional Thoughts while rereading the in vivo figures:

There are some ambiguities in how the manuscript is written, it says that the headplate and window were implanted on the day of recording, but presumably the authors mean on the first day of recording. Recording on the same day as surgery is a small deviation from the state of the art, and will give artificially good window clarity – but surgical drugs can take days to leave the system leaving the biological findings suspect. As the authors record for 7 weeks, showing that the window remains clear and useable past 72hours shouldn't be challenging – but would be necessary to show that it can be used for biological experiments. While i'm sure the Author's followed their universities guidelines, the two universities i've performed cranial windows in required 72 hours post-surgical recovery before any awake imaging experiments.

Potential users are going to want to know, how many cells they can resolve simultaneously in vivo. Especially, how many is too many. How deep can it go, and how does motion disrupt the ability to see these responses.

Details like the number of mice that these recordings came from, and the number of independent recordings are required. This doesn't look like many cells, given that the technique should be scalable to many spots simultaneously.

How easy or hard is it to resolve multiple spots at different z planes in scattering media/given heat constraints? Exactly how many spots can be read simultaneously in vivo is a critical parameter to assess how viable this is as a practical approach. Understanding how and when it fails is key to its use. It appears that in L2/3 it can only activate ~10 cells and maybe none in L5? This cuts into its useability.

Fig 6b lower do not appear to be the same cells as above, the legend would imply they don't need to be, but this could be more clear in the presentation.

Main Point 2.

How does heat/power density/long recordings affect cell health. (this question seemed to be asked in a few different ways across several 'concerns')

The authors provide many lines of evidence evaluating how well individual cells tolerated the experiment, however i feel that the final manuscript would have benefited from a main figure exploring this critical parameter. In particular the change in the action potential width could indicate a variety of problems in cell health, or simply changes in local temperature. Either way, any user of this technique would need to be aware of the potential confounds that this would have.

The changes in the latency to respond to depolarization is more concerning as its much larger in amplitude. This indicates that either cells are being chronically depolarized, some leak current is changing, or ion channel conductance/kinetics has changed. Any of these could be sufficient to disrupt a biological experiment. It didn't seem apparent to me if the authors had fully explored how long of an experiment they could perform without damage, but perhaps i missed it.

For any user, information about the limitations and side effects of an approach need to be very apparent and shouldn't be 'hidden' in a supplementary figure. But caveats are not in themselves a reason to reject the paper. All techniques have caveats, and the key to doing science well is to know and account for these side effects.

Similarly, the low number of neurons that can be imaged in vivo simultaneously seriously constrains possible experiments - but is also just one caveat to be considered.

Main Point 3.

At the risk of being blunt, Reviewer 2 seems to believe that a lot of the figures and plots were somewhat sloppy. I think the revised manuscript is sufficiently 'clean' - but this is a subjective decision. I am very much a fan of presenting raw data, and single trial data, as is featured in several of the main figures. The authors seems to have addressed all the issues raised, and nothing strikes me as 'incorrect'.

Overall Opinion:

I remain in favor of publication of this manuscript. I think the approach they present is not perfect, there may be heat induced changes in cell biology, and it can neither image as many cells or as deeply as other approaches can. However, it still manages to hit a niche: imaging moderate numbers of cells with good temporal resolution. It may be a steppingstone to future more revolutionary approaches. So, despite some shortcomings I think this manuscript aids the scientific discourse, and could be beneficial to future scientists.

Rebuttal Table: Response to Reviewer’s Comments

Comment	Response
Reviewer #1	
The authors adequately responded to all comments and thoroughly revised the paper. I have no further requests and I support the publication of the current version.	We thank Reviewer #1 for the time they dedicated to reviewing our manuscript, and we are glad we were able to adequately respond to all of their comments.
Reviewer #3	
This is a re-review of Sims et al. I generally felt positively about the original manuscript, and my overall opinion remains positive.	We thank Reviewer #3 for their positive view of our work and we appreciate the extra time they spent to provide an input on the comments of Reviewer #2.
I am still skeptical of the utility of using opsin and voltage imaging in this confounded way. I believe the use case of examining the timing of an optogenetically evoked voltage sensor signal, is worthwhile, but is an incredibly niche application.	We believe simultaneous optogenetic stimulation and voltage imaging in the same cells is critical for confirming the occurrence of optogenetic activation of the target cell(s). It is also important to characterize (and optimize) how individual cells are activated, e.g., whether single spikes or extended bursts are elicited. These results are enabled by the fast temporal resolution of voltage imaging over calcium imaging. We believe this method will become more widely used as more groups adopt approaches for optogenetic stimulation of groups of individual target cells.
More consideration needs to be placed on how motion will impact this approach:  - The only discussion I can find about motion artifacts is a line that says a 17um spot minimizes artifacts. This argument should be matched with data. It seems that artifacts will occur anytime a portion of a fluorescent cell reaches the edge of the holographic disk. Therefore, in the absence of active motion correction, the critical question isn't the size of the disk but the distance from the edge of the disk to the nearest cell 	One of the benefits of our approach is that a 2D image is acquired at each timepoint (in contrast to cases where, for instance, all fluorescence is integrated using a point detector). Hence it ought to be possible to use classic registration techniques to detect, measure and, ultimately, correct for motion artefacts or the presence of neighboring cells within each illumination spot. However, as described in the manuscript, we found that expression in our in vivo preparation was relatively sparse, and we did not encounter the problem highlighted by Reviewer #3. Moreover, because mice were both head-fixed and deeply anaesthetized we did not observe any motion artefacts in the majority of experiments. In rare cases when motion was observed, we found that the resulting traces did not resemble neural activity and largely correlated with changes in the centre of mass (X, Y) of the fluorescence, for instance:

membrane. Brain motion can be variable, but 3-5um would not be unreasonable. This however poses a tradeoff with cell density, if a spot is too big, you'll get a second source of contamination where additional cells entering your holographic spot. These caveats and how/if they're overcome are important for a practical user of the technology.

Whereas, in the majority of cases we did not observe that the centre of mass of the fluorescence changed as a function of time, for instance (representative data presented in Figure 6e, upper):

For these experiments, any datasets where any changes in fluorescent intensity were correlated with a displacement in the center of mass (as above) were rejected. We have updated the methods section of the revised manuscript to include this (unintentionally) omitted information.

However, of course, as pointed out by Reviewer #3, in order for the method to be utilized in a broader context (and in more densely populated brain regions such as the hippocampus) it will be necessary to implement "active" motion correction. e.g. co-expressing a fluorescent label in the cell nucleus and performing real-time closed-loop correction of target spot positions.

The authors liken their low NA Gaussian approach to 3D-SHOT, but this seems

To avoid confusion, we removed all references to 3D-SHOT in the main text (lines 116, 145).

imprecise. Rereading the 3D-SHOT papers show substantial differences. For example, there isn't a beam expander before the diffraction grating in any of the published 3D-SHOT approaches that I'm aware of. Furthermore, given the diffuser in 3D-SHOT it's debatable whether it is best described as a low NA gaussian beam. Perhaps simple text changes that make it clear that this is not precisely 3D-SHOT would allow readers to appreciate the differences.	
---	--

Reviewer #3 on behalf of #2	
--

I've finished rereading the rebuttal and looking into the Author's response to each of Reviewer 2's concerns. Reviewer 2 clearly had many detailed concerns and how well the authors addressed them would be best answered by the original reviewer. That said, I agree with many of Reviewer 2's original concerns, I don't believe that the authors have neglected any of the reviewer's questions but could see the original reviewer not being completely satisfied by the findings. My net assessment is that the final product is technically viable, but this paper doesn't prove to me that I want to adopt it as an approach for biological experiments. In my opinion, that's ok - and it still deserves to be published. I have tried	We are grateful that Reviewer #3 believes that we have adequately responded to all of reviewer #2's questions and that the work merits publication.
---	--

similar experiments in preliminary attempts (although it's not part of my current research strategy) so to see what worked and how well is important step in the scientific process and adds value.	
Main Point 1.	
The first concern was that this approach is supposed to be used in vivo, but they hadn't shown it. The Authors added a figure devoted to in vivo recordings and this made their paper much stronger. While this figure could be better (showing longevity of recordings, more comprehensive study of the number of cells recorded at a time, etc), it at the very least shows in vivo recordings are achievable. If there was one thing to change it would be to expand upon these in vivo experiments.	We thank the Reviewer for recognizing that the new in-vivo data have made the paper much stronger. We agree that further work is necessary to optimize and characterize this approach, which we have already begun and intend to continue.
Additional Thoughts while rereading the in vivo figures:	
There are some ambiguities in how the manuscript is written, it says that the headplate and window were implanted on the day of recording, but presumably the authors mean on the first day of recording. Recording on the same day as surgery is a small deviation from the state of the art, and will give artificially good window clarity – but surgical drugs can take days to leave the system leaving the biological findings suspect. As the	All in vivo data was acquired on anaesthetized animals. Most in vivo data presented was acquired using acute preparations. In this configuration, animals were injected with a JEDI-2P-Kv virus. Recordings were performed between 3 to 7 weeks following injection. On the day of recording, the animal was anaesthetized and the headplate and cranial window were implanted. The mouse was placed under the microscope and experiments were performed under anesthesia. At the end of the recording session, the animal was sacrificed and the brain was fixed. We also conducted experiments using chronic preparations. In this case, animals were injected with a JEDI-2P-Kv virus and the headplate and cranial windows were implanted 3 weeks after AAV injection. The animal was then left to recover for at least one week before recording. We generally obtained clear cranial windows for a month and scanless two-photon voltage imaging experiments were performed during this period. No notable differences were observed between results obtained from chronic or acute preparations.

authors record for 7 weeks, showing that the window remains clear and useable past 72hours shouldn't be challenging – but would be necessary to show that it can be used for biological experiments. While i'm sure the Author's followed their universities guidelines, the two universities i've performed cranial windows in required 72 hours post-surgical recovery before any awake imaging experiments.	To avoid any ambiguities, we have clarified these points in the methods section of the revised manuscript.
Potential users are going to want to know, how many cells they can resolve simultaneously in vivo. Especially, how many is too many. How deep can it go, and how does motion disrupt the ability to see these responses.	We have shown that the number of cells that can be resolved in vivo using our approach will depend on depth, cell distribution and thermal heating. For now, we can estimate a safe limit to 10 to 15 cells in layer 2/3 for long (>1s) illumination times. After a second, the temperature rise stabilizes to an equilibrium value (as shown in Supplementary Figures 11, 14 and 15). We performed 30s recordings; hence more studies will be necessary to determine whether prolonged exposure to higher temperatures induce any deleterious impacts in the case of longer acquisition times. Regarding motion, please refer to the answer above.
Details like the number of mice that these recordings came from, and the number of independent recordings are required. This doesn't look like many cells, given that the technique should be scalable to many spots simultaneously.	All relevant details are summarized in Supplementary Table 4. We have also added these details to the caption of Figure 6.
How easy or hard is it to resolve multiple spots at different z planes in scattering media/given heat constraints? Exactly how many spots can be read simultaneously in vivo is a critical parameter to assess how viable this is as a practical approach.	We intend to explore the difficulty of resolving multiple spots in different z planes in scattering media in future experiments. Given the sparsity of the labelling of our in-vivo preparation, we found that the average distance between targets was larger than the spread of the fluorescent signal due to scattering (i.e. we observed very little crosstalk during these proof-of-principle experiments). We found that light induced temperature rises were limiting factor for the number of cells during these experiments. We estimated that an upper limit of 10 to 15 cells in layer 2/3 imaged simultaneously (refer to the answer two responses above).

Understanding how and when it fails is key to its use.	
It appears that in L2/3 it can only activate ~10 cells and maybe none in L5? This cuts into its useability.	This is the first manuscript demonstrating scanless 2P voltage imaging in vitro and, in vivo. We acknowledge that reaching L5 with our current scheme presents challenges, our manuscript addresses these limitations in the discussion. We outline strategies for overcoming this obstacle, including employing brighter GEVIs, utilizing de-mixing approaches and adopting multi PMT arrays. Comprehensive testing and validation of all these configurations is beyond the scope of this manuscript. However, we intend to thoroughly explore these possibilities in future studies. Although we have not yet demonstrated multi target voltage imaging in L5, our approach still leaves open a wide array of experiments on upper cortical layers or alternative animal models and experimental paradigms such e.g. zebrafish larvae or mouse retina studies. It should also be noted that, to our knowledge, no approaches have demonstrated capability for high (>10) SNR two-photon multi-target in vivo voltage imaging of L5 neurons.
Fig 6b lower do not appear to be the same cells as above, the legend would imply they don't need to be, but this could be more clear in the presentation.	We have modified the figure caption to clarify that the cells in Fig. 6b lower are not the same cells as above.
Main Point 2. How does heat/power density/long recordings affect cell health. (this question seemed to be asked in a few different ways across several 'concerns')	
The authors provide many lines of evidence evaluating how well individual cells tolerated the experiment, however i feel that the final manuscript would have benefited from a main figure exploring this critical parameter. In particular the change in the action potential width could indicate a variety of problems in cell health, or simply changes in local temperature. Either way, any user of this technique would need to be aware of the potential confounds that this would have.	Supplementary figures 8 and 12 show the required analysis on the effect of illumination on action potential width; these findings reveal that these changes are negligible when using power below 100 mW/cell (Supplementary Figure 8; using high repetition lasers), or 10 mW/cell (Supplementary figure 12; using low repetition lasers) which is sufficient in both cases to detect spikes with high SNR (>10). We thus concluded that working below this threshold represents a safe condition to maintain cell health. This is clearly mentioned in the discussion so we do not think it will generate confusions. Because of the high number of panels describing these experiments, we felt that these experiments were more suitable for Supplementary Figures.

The changes in the latency to respond to depolarization is more concerning as its much larger in amplitude. This indicates that either cells are being chronically depolarized, some leak current is changing, or ion channel conductance/kinetics has changed. Any of these could be sufficient to disrupt a biological experiment. It didn't seem apparent to me if the authors had fully explored how long of an experiment they could perform without damage, but perhaps i missed it.	We do not understand to which data the Reviewer is referring here, we have no figure showing changes in latency.
For any user, information about the limitations and side effects of an approach need to be very apparent and shouldn't be 'hidden' in a supplementary figure. But caveats are not in themselves a reason to reject the paper. All techniques have caveats, and the key to doing science well is to know and account for these side effects.	Given the limited number of allowed main figures, we had to prioritize what to show in the main figures. We have never considered data in supplementary material to be hidden results. On the contrary, we normally read very carefully the supplementary material of published manuscripts, as it is where many important characterizations are reported. We hope this will be the case for the readers of this manuscript as well. These caveats have also been highlighted in the results (lines 341 – 386, 396 - 397, 461 – 473) and discussion (595-600) sections.
Similarly, the low number of neurons that can be imaged in vivo simultaneously seriously constrains possible experiments - but is also just one caveat to be considered.	We are not aware of any techniques that have demonstrated two-photon in vivo multi-target voltage imaging with comparable signal-to-noise ratio (SNR>10) and reaching a higher number of cells. So, we still think that our approach marks an important milestone in multitarget 2P voltage imaging. We agree with the Referee that the number of cells imaged is a feature to be improved in future and are confident that this will be possible with the development of efficient red-shifted indicators and/or the implementation of multi-detectors array or ad hoc light shaping approaches.
Main Point 3.	
At the risk of being blunt, Reviewer 2 seems to believe that a lot of the figures and plots were somewhat sloppy. I	We thank the Reviewer for this positive consideration on the cleanness of our manuscript.

think the revised manuscript is sufficiently 'clean' - but this is a subjective decision. I am very much a fan of presenting raw data, and single trial data, as is featured in several of the main figures. The authors seems to have addressed all the issues raised, and nothing strikes me as 'incorrect'.	
Overall Opinion:	
I remain in favor of publication of this manuscript. I think the approach they present is not perfect, there may be heat induced changes in cell biology, and it can neither image as many cells or as deeply as other approaches can. However, it still manages to hit a niche: imaging moderate numbers of cells with good temporal resolution. It may be a steppingstone to future more revolutionary approaches. So, despite some shortcomings I think this manuscript aids the scientific discourse, and could be beneficial to future scientists.	We thank the Reviewer for remaining in favor of the publication of the manuscript and for recognizing that this is but the first study on 2P scanless voltage imaging, and will act as a stepping stone to further innovation.